# AXIOM: Learning to Play Games in Minutes with Expanding Object-Centric Models

## Abstract

Current deep reinforcement learning (DRL) approaches achieve state-of-the-art performance in various domains, but struggle with data efficiency compared to human learning, which leverages core priors about objects and their interactions. Active inference offers a principled framework for integrating sensory information with prior knowledge to learn a world model and quantify the uncertainty of its own beliefs and predictions. However, active inference models are usually crafted for a single task with bespoke knowledge, so they lack the domain flexibility typical of DRL approaches. To bridge this gap, we propose a novel architecture that integrates a minimal yet expressive set of *core priors* about object-centric dynamics and interactions to accelerate learning in low-data regimes. The resulting approach, which we call AXIOM, combines the usual data efficiency and interpretability of Bayesian approaches with the across-task generalization usually associated with DRL. AXIOM represents scenes as compositions of objects, whose dynamics are modeled as piecewise linear trajectories that capture sparse object-object inter-actions. The structure of the generative model is expanded online by growing and learning mixture models from single events and periodically refined through Bayesian model reduction to induce generalization. AXIOM learns to play various games within only 10,000 interaction steps, with both a small number of parame-ters compared to DRL, and without the computational expense of gradient-based optimization.

## 1 Introduction

Reinforcement learning (RL) has achieved remarkable success as a flexible framework for mastering complex tasks. However, current methods have several drawbacks: they require large amounts of training data, depend on large replay buffers, and focus on maximizing cumulative reward without structured exploration (Li, 2017). This contrasts with human learning, which relies on core priors to quickly generalize to novel tasks (Spelke & Kinzler, 2007; Lake et al., 2017). Core priors represent fundamental organizational principles - or hyperpriors - that shape perception and learning, providing the scaffolding upon which more complex knowledge structures are built. For example, such priors allow humans to intuitively understand that objects follow smooth trajectories unless external forces intervene, and shape our causal reasoning, helping us to grasp action-consequence relationships (Téglás et al., 2011; Spelke et al., 1995). Describing visual scenes as factorized into objects has shown promise in sample efficiency, generalization, and robustness on various tasks (Wiedemer et al., 2023; Agnew & Domingos, 2018; Kipf et al., 2018). These challenges are naturally addressed by Bayesian formulations of model-based reinforcement learning (Ghavamzadeh et al., 2015), that provide a principled framework for incorporating prior knowledge into models and increase sample efficiency. Fully Bayesian variants of model-based RL (Kang et al., 2024; Parr et al., 2022) have been limited to small-scale, hand-engineered generative models, but due to their use of explicit probability distributions support continual adaptation without catastrophic forgetting. It has been argued that this approach aligns closely with biological cognition (Knill & Pouget, 2004; Friston, 2010), where beliefs are updated incrementally as new evidence emerges. Yet, despite these theoretical advantages, applications of active inference have typically been confined to small-scale tasks with carefully designed priors, failing to achieve the versatility that makes deep RL so powerful across diverse domains.

To bridge this gap, we propose a novel active inference architecture that integrates a minimal yet expressive set of core priors about objects and their interactions (Locatello et al., 2020a; Kipf

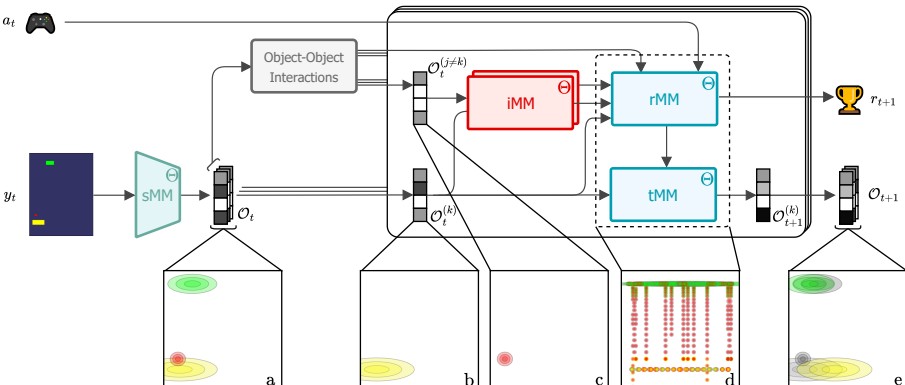

Figure 1: **Inference and prediction flow using AXIOM**: The sMM extracts object-centric representations from pixel inputs. For each object latent and its closest interacting counterpart, a discrete identity token is inferred using the iMM and passed to the rMM, along with the distance and the action, to predict the next reward and the tMM switch. The object latents are then updated using the tMM and the predicted switch to generate the next state for all objects. (a) Projection of the object latents into image space. (b) Projection of the $k^{\text{th}}$ latent whose dynamics are being predicted and (c) of its interaction partner. (d) Projection of the rMM in image space; each of the visualized clusters corresponds to a particular linear dynamical system from the tMM. (e) Projection of the predicted latents. The past latents at time $t$ are shown in gray.

et al., 2018). Specifically, we present AXIOM (**A**ctive e**X**panding **I**nference with **O**bject-centric **M**odels), which employs a object-centric state space model with three key components: (1) a Gaussian mixture model that parses visual input into object-centric representations and automatically expands to accommodate new objects; (2) a transition mixture model that discovers motion prototypes (e.g., falling, sliding, bouncing) and (3) a sparse relational mixture model over multi-object latent features, learning causally relevant interactions as jointly driven by object states, actions, rewards, and dynamical modes. AXIOM's learning algorithm offers three kinds of efficiency: first, it learns sequentially one frame at a time with variational Bayesian updating. This eliminates the need for replay buffers or gradient computations, and enables online adaptation to changes in the data distribution, a feature that has not been demonstrated in previous model-based RL approaches (Karl et al., 2016; Zhang et al., 2019; Hafner et al., 2025). Second, its mixture architecture facilitates fast structure learning by both adding new mixture components when existing ones cannot explain new data, and merging redundant ones to reduce model complexity (Ueda et al., 2000; Friston et al., 2016; 2024a). Finally, by maintaining posteriors over parameters, AXIOM can augment policy selection with information-seeking objectives and thus uncertainty-aware exploration (Dearden et al., 2013; Parr et al., 2022).

To empirically validate our model, we introduce the `Gameworld 10k` benchmark, a new set of environments designed to evaluate how efficiently an agent can play different pixel-based games in 10k interactions. Many existing RL benchmarks, such as the Arcade Learning Environment (ALE) (Bellemare et al., 2013) or MuJoCo (Todorov et al., 2012) domains, emphasize long-horizon credit assignment, complex physics, or visual complexity. These factors often obscure core challenges in fast learning and generalization, especially under structured dynamics. To this end, each of the games in `Gameworld 10k` follows a similar, object-focused pattern: multiple objects populating a visual scene, a player object that can be controlled to score points, and objects following continuous trajectories with sparse interaction mechanics. We formulate a set of 10 games with deliberately simplified visual elements (single color sprites of different shapes and sizes) to focus the current work on the representational mechanisms used for modeling dynamics and control, rather than learning an overly-expressive model for object segmentation. The Gameworld environments also enable precise control of game features and dynamics, which allows testing how systems adapt to sparse interventions to the causal or visual structure of the game, e.g., the shape and color of game objects. On this benchmark, our agent outperforms popular reinforcement learning models in the low-data regime (10,000 interaction steps) without relying on any kind of gradient-based optimization. To conclude, although we have not deployed AXIOM at the scale of complicated control tasks typical of the RL literature, our results represent a meaningful step toward building agents capable of building

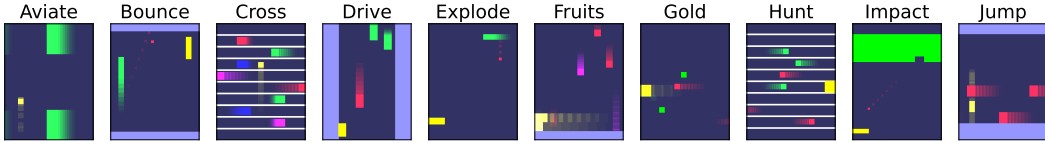

Figure 2: **Gameworld10k**: Visual impression of the 10 games in the `Gameworld 10k` suite. Sequences of ten frames are overlayed with increasing opacity to showcase the game dynamics.

compact, interpretable world models and exploiting them for rapid decision-making across different domains. Our main contributions are the following:

- We introduce AXIOM, a novel object-centric active inference agent that is learned online, interpretable, sample efficient, adaptable and computationally cheap.
- To demonstrate the efficacy of AXIOM, we introduce a new, modifiable benchmark suite targeting sample-efficient learning in environments with objects and sparse interactions.
- We show that our gradient-free method can outperform state-of-the-art deep learning methods both in terms of sample efficiency and absolute performance, and show that our online learning scheme is at least as robust as strong deep RL baselines under structured visual and mechanics perturbations in `Gameworld 10k`, while additionally allowing interpretable, white-box interventions on the world model.

## 2 RELATED WORK

**World Models.** Initial breakthroughs in deep reinforcement learning leveraged deep Q networks to play Atari games (Mnih et al., 2013) and were sample-inefficient; they required training on millions of images to reach human-level performance. Recent work using model-based reinforcement learning (Stadie et al., 2015; Ghavamzadeh et al., 2015), trains an explicit world model to bolster sample efficiency (Ye et al., 2021; Wang et al., 2024). A notable example is the Dreamer class of architectures, which uses a mix of recurrent continuous and discrete states to model the dynamics of the environment (Hafner et al., 2019; 2020; 2025). AXIOM is also an example of a model-based RL architecture, but notably its world model is explicitly probabilistic and interpretable. Unlike previous world models (even probabilistic ones like Karl et al. (2016); Zhang et al. (2019)), AXIOM models the latent world state using explicit object-centric representations that have directly interpretable features like position, velocity, color, and shape, and does so using only closed form variational updating.

**Object-Centric Representations** Unsupervised object segmentation from pixels has recently gained traction in the deep learning community thanks to the introduction of models like IODINE (Greff et al., 2019) and Slot Attention (Locatello et al., 2020b); the latter leverages the strengths and efficiency of self-attention to enforce competition between slots in explaining pixels. This directly inspired AXIOM, which infers the presence and properties of objects in 2-D images using its slot mixture model. Other relevant object-centric approaches include FOCUS, which targets generalization in the low data regime for robot manipulation (Ferraro et al., 2025), and OC-STORM, which uses object-centric state variables (learned with supervision) to predict environment dynamics and rewards (Zhang et al., 2025).

**Bayesian Inference and Learning.** Inference and learning AXIOM are driven by variational Bayesian inference for the states and parameters of mixture models (Wainwright et al., 2008; Van de Maele et al., 2024) and planning is achieved using active inference, which balances reward maximization with information gain by minimizing expected free energy (Parr et al., 2022; Friston, 2010). To learn the structure of the environment, we drawn inspiration from fast structure learning methods (Friston et al., 2024b), that first add mixture components to the model (Rasmussen, 1999; Song & Wang, 2004) and then prunes them using Bayesian model reduction (Friston et al., 2016). An important distinction between AXIOM's model and the original fast structure learning approach is that AXIOM uses a structured, object-centric state space and uses continuous mixture model likelihoods, rather than purely discrete ones. AXIOM's transition model is a type of truncated infinite switching linear dynamical system (SLDS) (Ghahramani & Hinton, 1996; 2000; Geadah et al., 2024). In particular, we rely on a recent formulation called the *recurrent* SLDS (Linderman et al., 2016), that introduces dependence of the switch state on the continuous state, to address two key limitations of the standard

SLDS: state-independent transitions and context-blind dynamics. Our innovation is in how we handle the recurrent connection of the rSLDS: we do this using a *generative*, as opposed to *discriminative*, model for the switching states. This allows for more flexible conditioning of the switch state on various information sources (both continuous and discrete).

## 3 METHODS

AXIOM is formulated in the context of a partially observable Markov decision process (POMDP). At each time step $t$, the hidden state $h_t$ evolves according to $h_t \sim P(h_t \mid h_{t-1}, a_{t-1})$, where $a_t$ is the action taken at time $t$. The agent does not observe $h_t$ directly but instead receives an observation $y_t \sim P(y_t \mid h_t)$, and a reward $r_t \sim P(r_t \mid h_t, a_t)$. AXIOM learns an *object-centric state space model* by maximizing the Bayesian model evidence—equivalently, minimizing (expected) free energy—through active interaction with the environment. The model factorizes perception and dynamics into separate generative blocks: (i) in perception, pixels are explained by object-centric latent variables $\mathcal{O}_t$, associating each of $N$ pixels to one of $K$ slots using the assignment variable $\mathbf{s}_{t,\text{smm}}$[1]; (ii) dynamics are modeled per-object using their object-centric latent descriptions as inputs to a recurrent switching state space model (similar to an rSLDS (Linderman et al., 2016)). Each object-centric slot latent $\mathcal{O}_t^{(k)}$ consists of continuous ($\mathbf{x}_t^{(k)}$), discrete ($\mathbf{z}_t^{(k)}$), and switching ($\mathbf{s}_t^{(k)}$) latent variables[2]. The continuous latents represent properties of an object, such as its position, color, and shape. The discrete latents capture categorical attributes of the slot (e.g., object type), and the sandwiching latents determine the slot's instantaneous trajectory. The relevant variables for the generative model are summarized below:

$$\mathbf{y}_t : \text{pixels} \qquad r_t : \text{reward} \qquad a_t : \text{action} \qquad \mathbf{s}_{t,\text{smm}} : \text{slot assignment}$$
$$\mathbf{x}_t^{(k)} : \text{continuous latent} \qquad \mathbf{z}_t^{(k)} : \text{discrete latent} \qquad \mathbf{s}_t^{(k)} : \text{trajectory switch}$$

For notational brevity, we introduce the following variables that group the object-centric latents $\mathcal{O}_t$, model parameters $\Theta$, and latent sequence $\mathcal{Z}_{0:T}$:

$$\mathcal{O}_t = \{\mathcal{O}_t^{(k)}\}_{k=0}^K \qquad\qquad \tilde{\Theta} = \{\Theta_{\text{smm}}, \Theta_{\text{imm}}, \Theta_{\text{rmm}}, \Theta_{\text{tmm}}\}$$
$$= \{\mathbf{x}_t^{(k)}, \mathbf{z}_t^{(k)}, \mathbf{s}_t^{(k)}\}_{k=0}^K \qquad\qquad \mathcal{Z}_{0:T} = \{\mathcal{O}_t, \mathbf{s}_{t,\text{smm}}\}_{t=0}^T$$

The joint distribution over input sequences $\mathbf{y}_{0:T}$, latent state sequences $\mathcal{Z}_{0:T}$, actions $a_{0:T-1}$, rewards $r_{1:T}$ and parameters $\tilde{\Theta}$ can be expressed as a hidden Markov model:

$$p(\mathbf{y}_{0:T}, \mathcal{Z}_{0:T}, a_{0:T-1}, r_{1:T}, \tilde{\Theta}) = p(y_0, \mathcal{Z}_0)p(\Theta_{\text{smm}})p(\Theta_{\text{imm}})p(\Theta_{\text{rmm}})p(\Theta_{\text{tmm}})$$
$$\prod_{t=1}^T \underbrace{p(\mathbf{x}_{t-1}, \mid \mathbf{z}_t, \Theta_{\text{imm}})}_{\text{Identity mixture model}} \underbrace{p(\mathbf{x}_{t-1}, \mathbf{z}_t, \mathbf{s}_t, a_{t-1}, r_t \mid \Theta_{\text{rmm}})}_{\text{Recurrent mixture model}} \qquad (1)$$
$$\prod_{k=1}^K \underbrace{p(\mathbf{y}_t \mid \mathbf{x}_t^{(k)}, \mathbf{s}_{t,\text{smm}}, \Theta_{\text{smm}})}_{\text{Slot mixture model}} \underbrace{p(\mathbf{x}_t^{(k)} \mid \mathbf{x}_{t-1}^{(k)}, \mathbf{s}_t^{(k)}, \Theta_{\text{tmm}})}_{\text{Transition mixture model}},$$

The slot mixture model (sMM) $p(\mathbf{y}_t \mid \mathbf{x}_t, \mathbf{s}_{t,\text{smm}}, \Theta_{\text{smm}})$ is a likelihood model that explains pixel data using mixtures of slot-specific latent states (see schematic in Figure 1). The identity mixture model (imm) $p(\mathbf{x}_{t-1} \mid \mathbf{z}_t, \Theta_{\text{imm}})$ is a likelihood model that assigns each object-centric latent to one of a set of discrete object types. The transition mixture model (tMM) $p(\mathbf{x}_t^{(k)} \mid \mathbf{x}_{t-1}^{(k)}, \mathbf{s}_t^{(k)}, \Theta_{\text{tmm}})$ describes each object's latent dynamics as a piecewise linear function of its own state. Finally, the recurrent mixture model (rMM) $p(\mathbf{x}_{t-1}, \mathbf{z}_t, \mathbf{s}_t, a_{t-1}, r_t \mid \Theta_{\text{rmm}})$ models the dependencies between multi-object latent states (like the switch states of the transition mixture), other global game states like reward $r$, action $a$, and the continuous and discrete features of each object. This module is what allows AXIOM to model sparse interactions between objects (e.g., collisions), while still treating each slot's dynamics as conditionally-independent given the switch states $\mathbf{s}_t^{(k)}$.

**Slot Mixture Model (sMM).** AXIOM processes sequences of RGB images one frame at a time. Each image is composed of $H \times W$ pixels and is reshaped into $N = HW$ tokens $\{\mathbf{y}_t^{(n)}\}_{n=1}^N$. Each token

---

[1] We use $s$ for denoting mixture switching/assignment variables

[2] We use the superscript index $k$ as in $q^{(k)}$ to select only the subset of $q \equiv q^{(1:K)}$ relevant to the $k^{\text{th}}$ slot.

$\mathbf{y}_t^{(n)}$ is a vector containing the $n^{\text{th}}$ pixel's color in RGB and its image coordinates (normalized to $[-1, +1]$). AXIOM models these tokens at a given time as explained by a mixture of the continuous slot latents (see far left side of Figure 1). The $K$ components of this mixture model are Gaussian distributions whose parameters are directly given by the continuous features of each slot latent $\mathbf{x}_t^{(1:K)}$. Associated to this Gaussian mixture is a binary assignment variable $s_{t,k,\text{smm}}^{(n)} \in \{0, 1\}$ indicating whether pixel $n$ at time $t$ is driven by slot $k$, with the constraint that $\sum_k s_{t,k,\text{smm}}^{(n)} = 1$. The mean of each Gaussian component is given a fixed linear projection $A$ of each object latent, which selects only its position and color features: $A\mathbf{x}^{(k)} = \left[\mathbf{p}^{(k)}, \mathbf{c}^{(k)}\right]$. The covariance of each component is a diagonal matrix whose diagonal is a projection of the 2-D shape of the object latent $B\mathbf{x}^{(k)} = \mathbf{e}^{(k)}$ (its spatial extent in the X and Y directions), stacked on top of a fixed variance for each color dimension $\sigma_{\mathbf{c}}^{(k)}$, which are given independent Gamma priors. The latent variables $\mathbf{p}^{(k)}, \mathbf{c}^{(k)}, \mathbf{e}^{(k)}$ are subsets of slot $k$'s full continuous features $\mathbf{x}_t^{(k)}$, and the projection matrices $A$, $B$ are fixed, not learned, parameters. The sMM's likelihood model for a single pixel and timepoint $\mathbf{y}_t^{(n)}$ can then be expressed as follows (dropping the $t$ subscript for notational clarity):

$$p(\mathbf{y}^{(n)} \mid \mathbf{x}^{(k)}, \sigma_{\mathbf{c}}^{(k)}, s_{k,\text{smm}}^{(n)}) = \prod_{k=1}^{K} \mathcal{N}(A\mathbf{x}^{(k)}, \text{diag}(\left[B\mathbf{x}^{(k)}, \sigma_{\mathbf{c}}^{(k)}\right]^{\top}))^{s_{k,\text{smm}}^{(n)}}. \qquad (2)$$

Each token's slot indicator $\mathbf{s}_{\text{smm}}^n$ is drawn from a Categorical distribution $\mathbf{s}_{\text{smm}}^{(n)} \mid \boldsymbol{\pi}_{\text{smm}} \sim \text{Cat}(\boldsymbol{\pi}_{\text{smm}})$ with mixing weights $\boldsymbol{\pi}_{\text{smm}}$. We place a truncated stick-breaking (finite GEM) prior on these weights, which is equivalent to a $K$-dimensional Dirichlet with concentration vector $(1, \ldots, 1, \alpha_{0,\text{smm}})$, where the first $K - 1$ pseudocounts are 1 and the final pseudocount $\alpha_{0,\text{smm}}$ reflects the propensity to add new slots (see Appendix A.2 for a more detailed description). All mixture models in AXIOM are equipped with the same sort of truncated stick-breaking priors on the mixing weights (Ishwaran & James, 2001).

**Identity Mixture Model (iMM).** A discrete identity code $\mathbf{z}_{\text{type}}^{(k)}$ is inferred for each object based on its continuous features. These identity codes are used to condition the inference of the recurrent mixture model used for dynamics prediction. Conditioning the dynamics on identity-codes in this way, rather than learning a separate dynamics model for each slot, allows AXIOM to use the same dynamics model across slots. This also enables the model to learn the same dynamics in a *type*-specific, rather than *instance*-specific, manner (Beck & Ramstead, 2025), and to remap identities when, e.g., the environment is perturbed, and colors change. Concretely, the iMM models the 5-D colors and shapes $\{\mathbf{c}^{(k)}, \mathbf{e}^{(k)}\}_{k=1}^{K}$ across slots as a mixture of up to $V$ Gaussian components (object types). The slot-level assignment variable $\mathbf{z}_{t,\text{type}}^{(k)}$ indicates which identity is assigned to the $k^{\text{th}}$ slot. The generative model for the iMM is (omitting the $t - 1$ subscript from latent variables):

$$p(\left[\mathbf{c}^{(k)}, \mathbf{e}^{(k)}\right]^{\top} \mid \mathbf{z}_{\text{type}}^{(k)}, \boldsymbol{\mu}_{1:V,\text{type}}, \boldsymbol{\Sigma}_{1:V,\text{type}}) = \prod_{j=1}^{V} \mathcal{N}(\boldsymbol{\mu}_{j,\text{type}}, \boldsymbol{\Sigma}_{j,\text{type}})^{z_{j,\text{type}}^{(k)}} \qquad (3)$$

$$p(\boldsymbol{\mu}_{j,\text{type}}, \boldsymbol{\Sigma}_{j,\text{type}}^{-1}) = \text{NIW}(\mathbf{m}_{j,\text{type}}, \kappa_{j,\text{type}}, \mathbf{U}_{j,\text{type}}, n_{j,\text{type}}) \qquad (4)$$

The same type of Categorical likelihood for the type assignments $\mathbf{z}_{\text{type}}^{(k)} \mid \boldsymbol{\pi}_{\text{type}} \sim \text{Cat}(\boldsymbol{\pi}_{\text{type}})$ and truncated stick-breaking prior $\text{Dir}(1, \ldots, 1, \alpha_{0,\text{type}})$ over the mixture weights is used to allow an arbitrary (up to a maximum of $V$) number of types to be used to explain the continuous slot features. We equip the prior over the component likelihood parameters with conjugate Normal Inverse Wishart (NIW) priors.

**Transition Mixture Model (tMM).** The dynamics of each slot are modelled as a mixture of linear functions of the slot's own previous state. The tMM's switch variable $\mathbf{s}_{t,\text{tmm}}^{(k)}$ selects a set of linear parameters $\boldsymbol{D}_l, \boldsymbol{b}_l$ to describe the $k^{\text{th}}$ slot's trajectory from $t$ to $t + 1$. Each linear system captures a distinct rigid motion pattern for a particular object (e.g., "ball in free flight", "paddle moving left").

$$p(\mathbf{x}_t^{(k)} \mid \mathbf{x}_{t-1}^{(k)}, \mathbf{s}_{t,\text{tmm}}^{(k)}, \boldsymbol{D}_{1:L}, \boldsymbol{b}_{1:L}) = \prod_{l=1}^{L} \mathcal{N}(\boldsymbol{D}_l \mathbf{x}_t^{(k)} + \boldsymbol{b}_l, 2I)^{s_{t,l,\text{tmm}}^{(k)}} \qquad (5)$$

We fix the covariance of all $L$ components to be $2I$, and all mixture likelihoods $\boldsymbol{D}_{1:L}, \boldsymbol{b}_{1:L}$ to have uniform priors. The mixing weights $\boldsymbol{\pi}_{\text{tmm}}$ for $\mathbf{s}_{t,\text{tmm}}^{(k)}$ have a truncated stick-breaking prior $\text{Dir}\big(1, \ldots, 1, \alpha_{0,\text{tmm}}\big)$ enabling the number of linear modes $L$ to be dynamically adjusted to the data by growing the model with propensity $\alpha_{0,\text{tmm}}$. The $L$ transition components of the tMM are not slot-dependent, but are shared and learned across all $K$ slot latents. The tMM can thus predict the motion of different objects using a shared, expanding set of dynamical motifs. As we will see in the next section, interactions between objects are captured by conditioning $\mathbf{s}_{t,\text{tmm}}^{(k)}$ on the states of other objects.

**Recurrent Mixture Model (rMM).** The switch states of the transition model are inferred directly from current slot-level features. This dependence of switch states on continuous features is the same construct used in the recurrent switching linear dynamical system or rSLDS (Linderman et al., 2016) [3]. Concretely, the rMM models the distribution of continuous and discrete variables as a mixture model driven by another per-slot latent assignment variable $\mathbf{s}_{\text{rmm}}^{(k)}$. The rMM defines a mixture likelihood over continuous and discrete slot-specific information: $(f_{t-1}^{(k)}, d_{t-1}^{(k)})$. The continuous slot features $f_{t-1}^{(k)}$ are a tuple of of both a subset of the $k^{\text{th}}$ slot's continuous state state $\mathbf{x}_{t-1}^{(k)}$, selected by projection matrix $C$, as well as a function $g(\mathbf{x}_{t-1}^{(1:K)})$ of the other slots states, which computes slot-to-slot interaction features, such as the displacement to the nearest object, associated identity codes, and other features detailed in Appendix A. The discrete features include categorical slot features like the identity of the closest object, the tMM switch state, and the reward and previous action:

$$f_{t-1}^{(k)} = \Big( C\mathbf{x}_{t-1}^{(k)}, g(\mathbf{x}_{t-1}^{(1:K)}) \Big), \quad d_{t-1}^{(k)} = \Big( \mathbf{z}_{t-1}^{(k)}, \mathbf{s}_{t,\text{tmm}}^{(k)}, a_{t-1}, r_t \Big) \tag{6}$$

We explored an ablation of the rMM (`fixed_distance`) where the displacement to the nearest object is not returned by $g(\mathbf{x}_{t-1}^{(1:K)})$; rather, the distance that triggers detection of the nearest interacting object is a fixed hyperparameter of the function $g$. This hyperparameter can be tuned to attain higher reward in most environments than the standard model, where the rMM learns the distance online. However, it comes at the cost of having to tune this hyperparamter in an environment-specific fashion (see Figure 3 and Table 1 for the effect of the `fixed_distance` ablation on performance).

The rMM assignment variable associated to a given slot is a binary vector $\mathbf{s}_{t,\text{rmm}}^{(k)}$ whose $m^{\text{th}}$ entry $s_{t,m,\text{rmm}}^{(k)} \in \{0,1\}$ indicates whether component $m$ explains the current tuple of mixed continuous-discrete data. Each component likelihood selected by $\mathbf{s}_{t,\text{rmm}}^{(k)}$ factorizes into a product of continuous (Gaussian) and discrete (Categorical) likelihoods:

$$p(f_{t-1}^{(k)}, d_{t-1}^{(k)} \mid \mathbf{s}_{t,\text{rmm}}^{(k)}) = \prod_{m=1}^{M} \left[ \mathcal{N}\big(f_{t-1}^{(k)}; \boldsymbol{\mu}_{m,\text{rmm}}, \boldsymbol{\Sigma}_{m,\text{rmm}}\big) \prod_i \text{Cat}\big(d_{t-1,i}^{(k)}; \mathbf{b}_{m,i}\big) \right]^{s_{t,m,\text{rmm}}^{(k)}} \tag{7}$$

As with all the other modules of AXIOM, we equip the mixing weights for $\mathbf{s}_{t,\text{rmm}}^{(k)}$ with a truncated stick-breaking prior whose final $M^{\text{th}}$ pseudocount parameter tunes the propensity to add new rMM components.

**Variational inference.** AXIOM uses variational inference to perform state inference and parameter learning. Briefly, this requires updating an approximate posterior distribution $q(\mathcal{Z}_{0:T}, \tilde{\boldsymbol{\Theta}})$ over latent variables and parameters to minimize the variational free energy $\mathcal{F}$, an upper bound on negative log evidence $\mathcal{F} \geq -\log p(\mathbf{y}_{0:T})$. In doing so, the variational posterior approximates the true posterior $p(\mathcal{Z}_{0:T}, \tilde{\boldsymbol{\Theta}} \mid \mathbf{y}_{0:T})$ from exact but intractable Bayesian inference . We enforce independence assumptions in the variational posterior over several factors: across states and parameters, across the $K$ slot latents, and over time $T$. This is known as the mean-field approximation:

$$q(\mathcal{Z}_{0:T}, \tilde{\boldsymbol{\Theta}}) = q(\tilde{\boldsymbol{\Theta}}) \prod_{t=0}^{T} \left( \prod_{n=1}^{N} q(\mathbf{s}_{t,\text{smm}}^n) \right) \left( \prod_{k=1}^{K} q(\mathcal{O}_t^{(k)}) \right) \tag{8}$$

---

[3] Unlike the rSLDS, which uses a discriminative mapping to infer the switch state from the continuous state, the rMM recovers this dependence *generatively* using a mixture model over mixed continuous–discrete slot states (Bishop & Lasserre, 2007).

$$q(\mathcal{O}_t^{(k)}) = q(\mathbf{x}_t^{(k)})q(\mathbf{z}_t^{(k)})q(\mathbf{s}_t^{(k)}), \quad q(\tilde{\mathbf{\Theta}}) = q(\mathbf{\Theta}_{\text{smm}})q(\mathbf{\Theta}_{\text{imm}})q(\mathbf{\Theta}_{\text{tmm}})q(\mathbf{\Theta}_{\text{rmm}}) \tag{9}$$

Note that the mixture variable of the sMM $\mathbf{s}_{t,\text{smm}}$ is independent from the other object-centric latents in both the generative model and the posterior because unlike the other discrete latents $\mathbf{z}_t^{(k)}$, it is not independent across $K$ slots.

We update the posterior over latent states $q(\mathcal{Z}_{0:T})$ (i.e., the variational E-step) using a simple form of forward-only filtering and update parameters using coordinate ascent variational inference (CAVI) that operates on single image frames (i.e., minibatches of $N = H \times W$ pixels). We use using a single iteration of CAVI for both the latent updates (E-step) and parameter updates (M-step) per timestep, resulting in a fast, streaming form of inference and learning (Wainwright et al., 2008). The simple, gradient-free form of these updates inherits from the exponential-family form of all the mixture models used in AXIOM.

### 3.1 GROWING AND PRUNING THE MODEL

**Fast structure learning.** In the spirit of *fast structure learning* (Friston et al., 2024a), AXIOM dynamically *expands* all four mixture modules (sMM, iMM, tMM, rMM) using an online growing heuristic: process each new datapoint sequentially, decide whether it is best explained by an existing component or whether a new component should be created, and then update the selected component's parameters. We fix a maximum number of components $\mathcal{C}_{\max}$ for each mixture model and let $\mathcal{C}_{t-1} \leq \mathcal{C}_{\max}$ be the number currently in use. For each component $c$ we store its variational parameters $\mathbf{\Theta}_c$, where for a particular model this might be a set of Normal Inverse Wishart parameters.

Upon observing a new input $y_t$, we compute for each component $c = 1, \ldots, \mathcal{C}_{t-1}$ the variational posterior predictive log density $\ell_{t,c} = \mathbb{E}_{q(\mathbf{\Theta}_c)}\big[\log p(y_t \mid \mathbf{\Theta}_c)\big]$. The truncated stick–breaking prior $\boldsymbol{\pi} \sim \text{Dir}(1, \ldots, 1, \alpha)$ then defines a "new–component" threshold $\tau_t = \log p_0(y_t) + \log \alpha$ where $p_0$ is the prior predictive density under an empty component. We select the component with highest score, $c^* = \arg\max_{c \leq \mathcal{C}_{t-1}} \ell_{t,c}$ and hard-assign $y_t$ to $c^*$ if $\ell_{t,c^*} \geq \tau_t$; otherwise—provided $\mathcal{C}_{t-1} < \mathcal{C}_{\max}$—we instantiate a new component and assign $y_t$ to it. Finally, given the hard assignment $z_t$, we update the chosen component's parameters via a variational M-step (coordinate ascent). The last weight in the Dirichlet absorbs any remaining mass, so $\sum_{c=1}^{\mathcal{C}_{\max}} \pi_c = 1$.) This algorithm is a deterministic, *maximum a posteriori* version of the CRP assignment rule (see Equation (8) of Ishwaran & James (2001)). The expansion threshold $\tau_t$ plays the role of the Dirichlet Process concentration $\alpha$. When $\alpha$ is small the model prefers explaining data with existing slots; larger $\alpha$ makes growth more likely. The procedure is identical for the sMM, iMM, tMM and rMM—only the form of $p(\cdot \mid \mathbf{\Theta}_c)$ and the model-specific caps on components $\mathcal{C}_{\max}$ and expansion thresholds $\tau_t$ differ.

**Bayesian Model Reduction (BMR).** Every $\Delta T_{\text{BMR}} = 500$ frames we sample up to $n_{\text{pair}} = 2000$ used rMM components, score their mutual expected log-likelihoods with respect to data generated from the model through ancestral sampling, and greedily test merge candidates. A merge is accepted if it *decreases* the expected free energy of the multinomial distributions over reward and next tMM switch, conditioned on the sampled data for the remaining variables; otherwise it is rolled back. BMR enables AXIOM to generalize dynamics from single events, for example learning that negative reward is obtained when a ball hits the bottom of the screen, by merging multiple single event clusters (see Section 4, Figure 4a).

### 3.2 PLANNING

AXIOM uses active inference for planning (Friston et al., 2017); it rolls out future trajectories conditioned on different policies (sequences of actions) and then does inference about policies using the expected free energy, where the chosen policy $\pi^*$ is that which minimizes the expected free energy:

$$\pi^* = \arg\min_\pi \sum_{\tau=0}^{H} \mathbb{E}_{q(r_\tau, \mathcal{O}_\tau | \pi)}[\underbrace{-\log \tilde{p}(r_\tau)}_{\text{Utility}} - \underbrace{D_{KL}(q(\boldsymbol{\alpha}_{\text{rmm}} | \mathcal{O}_\tau, \pi) \parallel q(\boldsymbol{\alpha}_{\text{rmm}}))}_{\text{Information gain (IG)}}] \tag{10}$$

The expected per-timestep utility $\mathbb{E}_{q(r_\tau, \mathcal{O}_\tau | \pi)}[\log \tilde{p}(r_\tau)]$ is computed using the predictive distribution over object-centric states under a policy $\pi$, which is calculated using rollouts from the learned model, i.e., $q(r_\tau, \mathcal{O}_\tau \mid \pi) \approx \mathbb{E}_{q(r_\tau, \mathcal{O}_{\tau-1} | \pi)}[p(r_\tau, \mathcal{O}_\tau \mid \mathcal{O}_{\tau-1}, a_{\tau-1}, \mathbf{\Theta})]$. The tilde notation $\tilde{p}(r_\tau)$ indicates

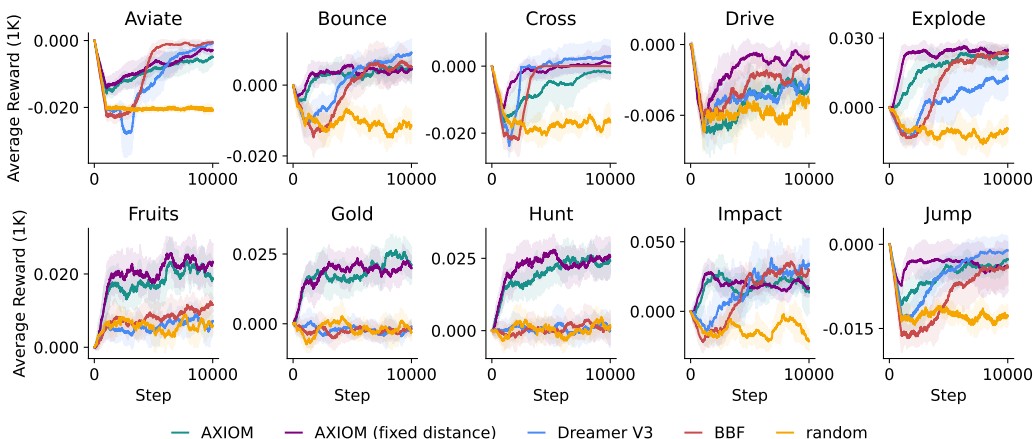

Figure 3: **Online learning performance**. Moving average (1k steps) reward per step during training for AXIOM, BBF, DreamerV3 and a random action baseline on `Gameworld 10k` environments. Mean and standard deviation range over 10 parameter seeds are shown per model, per step, and per environment.

an 'optimistically-biased' prior over for reward observations; this importantly is a fixed prior which differs from the reward contingencies learned by updating the rMM and is the critical ingredient that enables goal-directed behavior under active inference (Parr & Friston, 2019). In this context, the biased prior can be directly analogized to a utility function used in traditional model-based RL.[4] The expected information gain (second term on RHS of Equation (10)) is computed using the posterior Dirichlet counts of the rMM and scores how much information about rMM switch states would be gained by taking the policy under consideration. More details on planning are given in Appendix A.11.

## 4 RESULTS

To evaluate AXIOM, we compare its performance on Gameworld against two state-of-the-art baselines on sample-efficient, pixel-based deep reinforcement learning: BBF and DreamerV3.

**Benchmark.** The Gameworld environments are designed to be solvable by human learners within minutes, ensuring that learning does not hinge on brittle exploration or complex credit assignment. The suite includes 10 diverse games generated with the aid of a large language model, drawing inspiration from ALE and classic video games, while maintaining a lightweight and structured design. Figure 2 illustrates the variety and visual simplicity of the included games. To evaluate robustness, Gameworld 10k supports controlled interventions such as changes in object color or shape, testing an agent's ability to generalize across superficial domain shifts.

**Baselines.** BBF (Schwarzer et al., 2023) builds on SR-SPR (D'Oro et al., 2022) and represents one of the most sample-efficient model-free approaches. We adapt its preprocessing for the `Gameworld 10k` suite by replacing frame-skip with max-pooling over two consecutive frames, and chose the best performing variant of BBF on 10k interactions with Gameworld, which was the published version (width scale 4) with a replay ratio of 2. Second, DreamerV3 (Hafner et al., 2025) is a world-model-based agent with strong performance on games and control tasks with only pixel inputs; we use the published settings but set the train ratio to 1024 at batch size 16 (effective training ratio of 64:1) and use the 420 million parameter variant. Both these choices maximize Dreamer V3's performance in the small data regime (Hafner et al., 2025). We chose these baselines because they represent state of the art in sample-efficient learning from raw pixels. Note that for BBF and DreamerV3, we rescale the frames to $84 \times 84$ and $96 \times 96$ pixels, respectively (following the published implementations), whereas AXIOM operates on full $210 \times 160$ frames of Gameworld.

**Reward.** Figure 3 shows the 1000-step moving average of per-step reward from steps 0 to 10000 on the Gameworld 10k suite (mean ± 1 standard deviation over 10 seeds). Table 1 shows the cumulative

---

[4]We compute the expected utility by averaging the (unbiased) reward expected under the policy-conditioned predictive distribution with the log probabilities of the biased prior over the reward outcome: $\log \tilde{p}(r)^\top \mathbb{E}_{q(\mathcal{O}_\tau|\pi)}[p(r_\tau|\mathcal{O}_\tau)]$.

Table 1: **Cumulative reward over 10k steps for `Gameworld 10k` environments.** Cumulative reward is reported as mean $\pm$ std over 10 model seeds. Italic means AXIOM is better than BBF and Dreamer, bold is the overall best.

| Game | AXIOM | BBF | DreamerV3 | AXIOM (fixed dist.) | | AXIOM (no BMR) | | AXIOM (no IG) | |
|---|---|---|---|---|---|---|---|---|---|
| Aviate | $-90 \pm 19$ | $-90 \pm 05$ | $-114 \pm 20$ | $-76 \pm 13$ | | $-87 \pm 12$ | | $\mathbf{-71 \pm 16}$ | |
| Bounce | $27 \pm 13$ | $-1 \pm 15$ | $14 \pm 16$ | $\mathbf{34 \pm 12}$ | | $8 \pm 03$ | $\downarrow$ | $8 \pm 19$ | |
| Cross | $-68 \pm 36$ | $-48 \pm 07$ | $-27 \pm 08$ | $-18 \pm 21$ | $\uparrow$ | $-34 \pm 25$ | | $\mathbf{-7 \pm 03}$ | $\uparrow$ |
| Drive | $-49 \pm 04$ | $-37 \pm 06$ | $-45 \pm 06$ | $\mathbf{-22 \pm 04}$ | $\uparrow$ | $-67 \pm 03$ | $\downarrow$ | $-32 \pm 02$ | $\uparrow$ |
| Explode | $180 \pm 30$ | $101 \pm 13$ | $35 \pm 59$ | $\mathbf{234 \pm 16}$ | $\uparrow$ | $165 \pm 14$ | | $190 \pm 16$ | |
| Fruits | $182 \pm 21$ | $86 \pm 15$ | $60 \pm 07$ | $\mathbf{209 \pm 19}$ | | $141 \pm 19$ | $\downarrow$ | $200 \pm 20$ | |
| Gold | $190 \pm 18$ | $-26 \pm 12$ | $-21 \pm 10$ | $189 \pm 16$ | | $45 \pm 15$ | $\downarrow$ | $\mathbf{207 \pm 17}$ | |
| Hunt | $206 \pm 20$ | $4 \pm 12$ | $6 \pm 09$ | $\mathbf{231 \pm 28}$ | | $48 \pm 13$ | $\downarrow$ | $216 \pm 11$ | |
| Impact | $189 \pm 45$ | $122 \pm 20$ | $168 \pm 83$ | $192 \pm 09$ | | $\mathbf{197 \pm 21}$ | | $181 \pm 72$ | |
| Jump | $-55 \pm 09$ | $-96 \pm 17$ | $-55 \pm 17$ | $\mathbf{-38 \pm 25}$ | | $-45 \pm 05$ | $\uparrow$ | $-43 \pm 26$ | |

reward attained at the end of the 10k interaction steps for AXIOM, BBF and DreamerV3. AXIOM attains higher, or on par, mean cumulative reward than BBF and DreamerV3 in every Gameworld environment. Notably, AXIOM not only achieves higher peak scores on several games, but also converges much faster, often reaching most of its final reward within the first 5k steps, whereas BBF and DreamerV3 need nearly the full 10k. For those games where BBF and Dreamer seemed to show no-better-than-random performance at 10k, we confirmed that their performance does eventually improve, ruling out that the games themselves are intrinsically too difficult for these architectures (see Appendix E.2). Taken together, this demonstrates that AXIOM's object-centric world model, in tandem with its fast, online structure learning and inference algorithms, can reduce the number of interactions required to achieve high performance in pixel-based control. Fixing the interaction distance yields higher cumulative reward as the agent doesn't need to spend actions learning it, but doing so requires tuning the interaction distance for each game. This illustrates how having extra knowledge about the domain at hand can be incorporated into a Bayesian model like AXIOM to further improve sample efficiency. Including the information gain term from Equation (10) allows the agent to obtain reward faster in some games (e.g., Bounce), but actually results in a slower increase of the average reward for others (e.g., Gold), as encourages visitation of information-rich but negatively-rewarding states. BMR is crucial for games that need spatial generalization (like Gold and Hunt), but actually hurts performance on Cross, as merging clusters early on discounts the information gain term and discourages exploration. See Appendix E.3 for a more detailed discussion.

**Computational costs.** Table 2 compares model sizes and per-step training timing (model update and planning) measured on a single A100 GPU. While AXIOM incurs planning overhead due to the use of many model-based rollouts, its model update is substantially more efficient than BBF, yielding favorable trade-offs in wall-clock time per sample. The expanding object-centric model of AXIOM converges to a sufficient complexity given the environment, in contrast to the fixed (and much larger) model sizes of BBF and DreamerV3.

**Interpretability.** Unlike conventional deep RL methods, AXIOM has a structured, object-centric model whose latent variables and parameters can be directly interpreted in human-readable terms (e.g., shape, color, position). AXIOM's transition mixture model also decomposes complex trajectories into simpler linear sub-sequences. Figure 4a shows imagined trajectories and reward-conditioned clusters of the rMM for the Impact game. The imagined trajectories in latent space (middle panel of Figure 4a) are directly readable in terms of the colors and positions of the corresponding object. Because the recurrent mixture model (rMM) conditions switch states on various game- and object-relevant features, we can condition these switch variables on different game features and visualize them to show the rMM's learned associations (e.g., between reward and space). The right-most panel of

Table 2: **Training time per environment step on `Gameworld 10k`.** Parameter count for AXIOM varies for each environment. Planning time for AXIOM shows the range for 64 to 512 planning rollouts.

| Model | Parameters (M) | Model update (ms/step) | Planning (ms/step) |
|---|---|---|---|
| BBF | 6.47 | $135 \pm 36$ | N/A |
| DreamerV3 | 420 | $221 \pm 37$ | $823 \pm 93$ |
| **AXIOM** | 0.3 - 1.6 | $18 \pm 3$ | 252 - 534 |

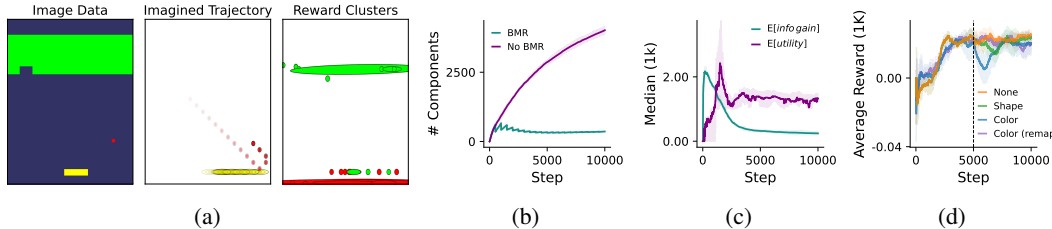

(a)                  (b)              (c)             (d)

Figure 4: **Tracking AXIOM's Behavior.** (a) Example frame from Impact at time $t$ (left); imagined trajectory in latent space conditioned on the observation at time $t$ and 32 timesteps into the future, conditioned on an action sequence with high predicted reward (middle); and rMM clusters shown in 2-D space and colored by expected reward (green positive reward, red negative reward aka punishment) (right). (b) Expanding rMM components are pruned over training using Bayesian Model Reduction (BMR) in Explode. (c) Information gain decreases while expected utility increases during training, showing an exploration-exploitation trade-off in Explode. (d) Performance following perturbation at 5k steps shows robustness to changes in game mechanics in Explode.

Figure 4a show the rMM clusters associated with reward (green) and punishment (red) plotted in space. The distribution of these clusters explains AXIOM's beliefs about where in space it expects to encounter rewards, e.g., expecting a punishment when the player misses the ball (red cluster at bottom of the right panel of Figure 4a). More rMM plots (showing the relationship between object identity and the identity of its interaction partner) can be found in the Appendix in Figure 8a.

Figure 4b shows the sharp decline in active rMM components during training. By actively merging clusters to minimize the expected free energy associated with the reduced model, Bayesian model reduction (BMR) improves computational efficiency while maintaining or improving performance (see Table 1). The resulting merged components enable interpolation beyond the training data, enhancing generalization. This automatic simplification reveals the minimal set of dynamics necessary for optimal performance, making AXIOM's decision process transparent and robust. Figure 4c demonstrates that, as training progresses, per-step information gain decreases while expected utility rises, reflecting a shift from exploration to exploitation as the world model becomes reliable.

**Perturbation Robustness.** Finally, we test AXIOM under systematic perturbations of game mechanics. Here, we perform a perturbation to the color or shape of each object at step 5000. Figure 4d shows that AXIOM is resilient to shape perturbations, as it still correctly infers the object type with the iMM. In response to a color perturbation, AXIOM adds new identity types and needs to re-learn their dynamics, resulting in a slight drop in performance and subsequent recovery. Due to the interpretable structure of AXIOM's world model, we can prime it with knowledge about possible color perturbations, and then only use the shape information in the iMM inference step, before remapping the perturbed slots based on shape and rescue performance. For more details, see Appendix E.4.

## 5 CONCLUSION

In this work, we introduced AXIOM, a novel and fully Bayesian object-centric agent that learns how to play simple games from raw pixels with improved sample efficiency compared to both model-based and model-free deep RL baselines. Importantly, it does so without relying on neural networks, gradient-based optimization, or replay buffers. By employing mixture models that automatically expand to accommodate environmental complexity, our method demonstrates strong performance within a strict $10,000$-step interaction budget on the `Gameworld 10k` benchmark. Furthermore, AXIOM builds interpretable world models with an order of magnitude fewer parameters than standard models while maintaining competitive performance. Our results suggest that Bayesian methods with structured priors about objects and their interactions have the potential to bridge the gap between the expressiveness of deep RL techniques and the data-efficiency of Bayesian methods with explicit models, suggesting a valuable direction for research.

**Limitations and future work.** Our work is limited by the fact that the core priors are themselves engineered rather than discovered autonomously. Future work will focus on developing methods to automatically infer such core priors from data, which should allow our approach to be applied to more complex domains like Atari or Minecraft (Hafner et al., 2025), where the underlying generative processes are less transparent but still governed by similar causal principles. We believe this direction represents a crucial step toward building adaptive agents that can rapidly construct structural models of novel environments without explicit engineering of domain-specific knowledge.

## REPRODUCIBILITY STATEMENT

We provide anonymized source code in the attached supplementary material that can be used to reproduce the main results for AXIOM in Figure 3 of the main text, with instructions for how to run the code available in the README.md of the source code. All hyperparameter configurations used to train AXIOM for the main results can be found in Table 3 of the Supplementary Appendix. Unless otherwise stated in the **Baselines** section of the main text, for two baselines models we used the hyperparameter settings and code repositories associated with the default variants of those architectures.

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

# A    FULL MODEL DETAILS

AXIOM's world model is a hidden Markov model with an object-centric latent state space. The model itself has two main components: 1) an object-centric, slot-attention-like (Locatello et al., 2020a) likelihood model; and 2) a recurrent switching state space model (Linderman et al., 2016). The recurrent switching state space model is applied to each object or slot identified by the likelihood model, and models the dynamics of each object with piecewise-linear trajectories. Unlike most other latent state-space models, including other object-centric ones, AXIOM is further distinguished by its adaptable complexity – it grows and prunes its model online through iterative expansion routines (see Algorithm 1) and reduction (see Algorithm 2) to match the structure of the world it's interacting with. This includes automatically inferring the number of objects in the scene as well as the number of dynamical modes needed to describe the motion of all objects. This is inspired by the recent *fast structure learning* approach (Friston et al., 2024a) developed to automatically learn a hierarchical generative model of a dataset from scratch.

**Preface on notation**    Capital bold symbols denote collections of matrix- or vector-valued random variables and lowercase bold symbols denote multivariate variables.

## A.1    GENERATIVE MODEL

The model factorizes perception and dynamics into separate generative blocks: (i) In perception, a slot Mixture Model (sMM) models pixels $\mathbf{y}_t$ as competitively explained by continuous latent variables factorized across slots or objects: $\mathbf{x}_t = \{\mathbf{x}_t^{(1)}, \ldots, \mathbf{x}_t^{(K)}\}$. Each pixel is assigned to one of (up to) $K$ slots using the assignment variable $\mathbf{s}_{t,\text{smm}}$; (ii) dynamics are modeled per-object using their object-centric latent descriptions as inputs to a recurrent switching state space model (similar to an rSLDS (Linderman et al., 2016)). We define the full latent sequence as $\mathcal{Z}_{0:T} = \{\mathcal{O}_t, \mathbf{s}_{t,\text{smm}}\}_{t=0}^T$. Each slot latent $\mathcal{O}_t^{(k)}$ consists of both continuous $\mathbf{x}_t^{(k)}$ and discrete latent variables. The continuous latents represent continuous properties of an object, such as its position, color and shape. The discrete latents are themselves split into two subtypes: $\mathbf{z}_t^{(k)}$ and $\mathbf{s}_t^{(k)}$. We use $\mathbf{z}_t^{(k)}$ to denote four latent descriptors that capture categorical attributes of the slot (e.g., object type), and $\mathbf{s}_t^{(k)}$ to denote a pair of switch states determining the slot's instantaneous trajectory.[5] Model parameters $\tilde{\mathbf{\Theta}}$ are split into module-specific subsets (e.g., $\mathbf{\Theta}_{\text{smm}}, \mathbf{\Theta}_{\text{tmm}}$). The joint distribution over input sequences $\mathbf{y}_{0:T}$, latent state sequences $\mathcal{Z}_{0:T}$, rewards $r_{0:T}$, actions $a_{0:T}$ and parameters $\tilde{\mathbf{\Theta}}$ can be expressed as a hidden Markov model:

$$p(\mathbf{y}_{0:T}, \mathcal{Z}_{0:T}, r_{0:T}, a_{0:T}, \tilde{\mathbf{\Theta}}) = p(y_0, \mathcal{Z}_0)p(\tilde{\mathbf{\Theta}}) \prod_{t=1}^T \underbrace{p(\mathbf{x}_{t-1}, \mid \mathbf{z}_t, \mathbf{\Theta}_{\text{imm}})}_{\text{Identity mixture model}} \underbrace{p(\mathbf{x}_{t-1}, \mathbf{z}_t, \mathbf{s}_t, a_{t-1}, r_t \mid \mathbf{\Theta}_{\text{rmm}})}_{\text{Recurrent mixture model}}$$

$$\prod_{k=1}^K \underbrace{p(\mathbf{y}_t | \mathbf{x}_t^{(k)}, \mathbf{s}_{t,\text{smm}}, \mathbf{\Theta}_{\text{smm}})}_{\text{Slot mixture model}} \underbrace{p(\mathbf{x}_t^{(k)} \mid \mathbf{x}_{t-1}^{(k)}, \mathbf{s}_t^{(k)}, \mathbf{\Theta}_{\text{tmm}})}_{\text{Transition mixture model}}, \quad (11)$$

We intentionally exclude the slot mixture assignment variable $\mathbf{s}_{t,\text{smm}}$ from the other object-centric discrete latents $\{\mathbf{z}_t^{(k)}\}_{k=1}^K$ because the sMM assignment variable is importantly not factorized over slots, since it is a categorical distribution over $K$-dimensional one-hot vectors, $\mathbf{s}_{t,\text{smm}} \in \{0,1\}^K$.

**Latent object states.**    At a given time–step $t \in 0, \ldots, T$ each object $k \in 1, \ldots, K$ is described by sets of both continuous and discrete variables (we reserve $k$ for 'slot index' everywhere):

$$\mathcal{O}_t = \{\mathbf{x}_t^{(k)}, \mathbf{z}_t^{(k)}, \mathbf{s}_t^{(k)}\}_{k=1}^K,$$

The continuous state $\mathbf{x}_t^{(k)}$ summarize latent features or descriptors associated with the $k^{\text{th}}$ object, including its 2-D position $\mathbf{p}_t^{(k)} = \{p_{t,x}^{(k)}, p_{t,y}^{(k)}\}$, a corresponding 2-D velocity $\mathbf{v}_t^{(k)} = \{v_{t,x}^{(k)}, v_{t,y}^{(k)}\}$,

---

[5]We use the superscript index $k$ as in $q^{(k)}$ to select only the subset of $q \equiv q^{(1:K)}$ relevant to the $k^{\text{th}}$ slot.

its color $\mathbf{c}_t^{(k)} = \{c_{t,r}^{(k)}, c_{t,g}^{(k)}, c_{t,b}^{(k)}\}$ its 2-D shape encoded as its extent along the X and Y directions $\mathbf{e}_t^{(k)} = \{e_{t,x}^{(k)}, e_{t,y}^{(k)}\}$, and an 'unused counter' $u_t^{(k)}$, which tracks how long the $k^{\text{th}}$ object has gone undetected:

$$\mathbf{x}_t^{(k)} = \left[\mathbf{p}_t^{(k)}, \mathbf{c}_t^{(k)}, \mathbf{v}_t^{(k)}, u_t^{(k)}, \mathbf{e}_t^{(k)}\right]^\top \in \mathbb{R}^{10}.$$

In addition to the continuous latents, each object is also characterized by two sets of discrete variables: $\mathbf{z}_t^{(k)}$ and $\mathbf{s}_t^{(k)}$. The first set $\mathbf{z}_t^{(k)}$ captures categorical information about the object that is relevant to predicting its instantaneous dynamics. This includes a latent object 'type' $\mathbf{z}_{t,\text{type}}^{(k)}$ (used to identify dynamics across object instances based on their shared continuous properties, e.g., objects that have the same shape and color are expected to behave similarly); the object type index of the nearest other object that slot $k$ is interacting with (or if it isn't interacting with anything) $\mathbf{z}_{t,\text{interacting}}^{(k)}$, its presence/absence in the current frame $\mathbf{z}_{t,\text{presence}}^{(k)}$, and whether the object is moving or not $\mathbf{z}_{t,\text{moving}}^{(k)}$. We define each of these discrete variables in 'one-hot' vector format, so as vectors whose entries $z_{t,m,\text{name}}^{(k)}$ are either 0 or 1, with the constraint that $\sum_m z_{t,m,\text{name}}^{(k)} = 1$. We can thus write the full discrete $\mathbf{z}_t^{(k)}$ latent as follows:

$$\mathbf{z}_t^{(k)} = \left[\mathbf{z}_{t,\text{type}}^{(k)}, \mathbf{z}_{t,\text{interacting}}^{(k)}, \mathbf{z}_{t,\text{presence}}^{(k)}, \mathbf{z}_{t,\text{moving}}^{(k)}\right]$$

$$\in \{0,1\}^{\mathcal{C}_{\text{type}} + \mathcal{C}_{\text{interacting}} + \mathcal{C}_{\text{presence}} + \mathcal{C}_{\text{moving}}}$$

$$\text{with} \quad \begin{cases} \mathcal{C}_{\text{type}} = V \le V_{\max} \\ \mathcal{C}_{\text{interacting}} = V + 1 \\ \mathcal{C}_{\text{presence}} = 2 \\ \mathcal{C}_{\text{moving}} = 2 \end{cases}$$

where $V$ is the number of object types inferred by the identity mixture model or iMM, subject to a maximum value of $V_{\max}$ (see Appendix A.6). Note that $\mathcal{C}_{\text{interacting}}$ has maximum index $V + 1$ because it includes an extra index for the state of 'not interacting with any other object.'

The second set of discrete variables $\mathbf{s}_t^{(k)}$ form a pair of concurrent switch states that jointly select one of an expanding set of linear dynamical systems to predict the object's future motion:

$$\mathbf{s}_t^{(k)} = \left[\mathbf{s}_{t,\text{tmm}}^{(k)}, \mathbf{s}_{t,\text{rmm}}^{(k)}\right]$$

$$\in \{0,1\}^{\mathcal{S}_{\text{tmm}} + \mathcal{S}_{\text{rmm}}}$$

$$\text{with} \quad \begin{cases} \mathcal{S}_{\text{tmm}} \le L \\ \mathcal{S}_{\text{rmm}} \le M \end{cases}$$

The first variable $\mathbf{s}_{t,\text{tmm}}^{(k)}$ is the switching variable of the transition mixture model or tMM, whereas the second switch variable $\mathbf{s}_{t,\text{rmm}}^{(k)}$ is the assignment variable of another mixture model – the recurrent mixture model or rMM – which furnishes a likelihood over $\mathbf{s}_{\text{tmm}}$ as well as other continuous and discrete features of each object.

In the sections that follow we will detail each of the components in the full AXIOM world model and give their descriptions in terms of generative models. These descriptions will show how the latent variables $\mathcal{Z}_{0:T} = \{\mathcal{O}_t, \mathbf{s}_{t,\text{smm}}\}_{t=0}^T$ and component-specific parameters (e.g, $\mathbf{\Theta}_{\text{rMM}}$ relate to observations and other latent variables.

### A.2 SLOT MIXTURE MODEL (SMM)

AXIOM processes sequences of RGB images one frame at a time. Each image is composed of $H \times W$ pixels and is reshaped into $N = HW$ tokens $\{\mathbf{y}_t^n\}_{n=1}^N$. Each token $\mathbf{y}_t^n$ is a vector containing the $n^{\text{th}}$ pixel's color in RGB and its image coordinates (normalized to $[-1, +1]$). AXIOM models these tokens at a given time as explained by a mixture of the continuous slot latents; we term this likelihood

construction the *Slot Mixture Model* (sMM, see far left side of Figure 1). The $K$ components of this mixture model are Gaussian distributions whose parameters are directly given by the continuous features of each slot latent $\mathbf{x}_t^{(1:K)}$. Associated to this Gaussian mixture is a binary assignment variable $s_{t,k,\text{smm}}^{(n)} \in \{0,1\}$ indicating whether pixel $n$ at time $t$ is driven by slot $k$, with the constraint that $\sum_k s_{t,k,\text{smm}}^{(n)} = 1$. The sMM's likelihood model for a single pixel and timepoint $\mathbf{y}_t^n$ can be expressed as follows (dropping the $t$ subscript for notational clarity):

$$p(\mathbf{y}^{(n)} \mid \mathbf{x}^{(1:K)}, \sigma_{\mathbf{c}}^{(1:K)}, \mathbf{s}_{\text{smm}}^{(n)}) = \prod_{k=1}^{K} \mathcal{N}(A\mathbf{x}^{(k)}, \text{diag}([B\mathbf{x}^{(k)}, \sigma_{\mathbf{c}}^{(k)}]^{\top}))^{s_{k,\text{smm}}^{(n)}} \tag{12}$$

$$\hat{A} = \begin{bmatrix} I_5 & \mathbf{0}_{5\times5} \end{bmatrix}, \quad \hat{B} = \begin{bmatrix} \mathbf{0}_{2\times8} & I_2 \end{bmatrix} \tag{13}$$

$$p(A) = \delta\left(A - \hat{A}\right) \tag{14}$$

$$p(B) = \delta\left(B - \hat{B}\right) \tag{15}$$

$$p(\mathbf{s}_{\text{smm}}^{(n)} \mid \boldsymbol{\pi}_{\text{smm}}) = \text{Cat}(\boldsymbol{\pi}_{\text{smm}}), \; p(\boldsymbol{\pi}_{\text{smm}}) = \text{Dir}\big(\underbrace{1, \ldots, 1}_{K-1 \text{ times}}, \alpha_{0,\text{smm}}\big) \tag{16}$$

$$p(\sigma_{\mathbf{c}}^{(k)}) = \prod_{j \in \text{R, G, B}} \Gamma(\gamma_{0,j}, 1) \tag{17}$$

$$p(\boldsymbol{\Theta}_{\text{smm}}) = p(A)p(B)p(\boldsymbol{\pi}_{\text{smm}})p(\sigma_{\mathbf{c}}^{(k)}) \tag{18}$$

The mean of each Gaussian component is given a linear projection $A$ (with fixed Dirac delta prior $\hat{A}$) of each object latent, which selects only its position and color features: $A\mathbf{x}^{(k)} = [\mathbf{p}^{(k)}, \mathbf{c}^{(k)}]$. The covariance of each component is a diagonal matrix whose diagonal is a projection of the 2-D shape of the object latent $B\mathbf{x}^{(k)} = \mathbf{e}^{(k)}$ (its spatial extent in the X and Y directions), stacked on top of a fixed variance for each color dimension $\sigma_{\mathbf{c}}^{(k)}$, which are given independent Gamma priors. The $B$ matrix is also fixed operationally to a fixed projection via an infinitely precise prior with mean $\hat{B}$, which has the effect of slicing out the shape variables $\mathbf{e}^{(k)}$. The latent variables $\mathbf{p}^{(k)}, \mathbf{c}^{(k)}, \mathbf{e}^{(k)}$ are subsets of slot $k$'s full continuous features $\mathbf{x}_t^{(k)}$, and the projection matrices $A$, $B$ remain fixed and unlearnable. Each token's slot indicator $\mathbf{s}_{\text{smm}}^n$ is drawn from a Categorical distribution with mixing weights $\boldsymbol{\pi}_{\text{smm}}$. We place a truncated stick-breaking (finite GEM) prior on these weights, which is equivalent to a $K$-dimensional Dirichlet with concentration vector $(1, \ldots, 1, \alpha_{0,\text{smm}})$, where the first $K - 1$ pseudocounts are 1 and the final pseudocount $\alpha_{0,\text{smm}}$ reflects the propensity to add new slots. All subsequent mixture models in AXIOM are equipped with the same sort of truncated stick-breaking priors on the mixing weights (Ishwaran & James, 2001).

## A.3 MOVING AND PRESENCE LATENT VARIABLES

Within the discrete variables that describe each slot $\mathbf{z}_t^{(k)}, \mathbf{s}_t^{(k)}$, there are two one-hot-encoded Bernoulli variables: $\mathbf{z}_{t,\text{moving}}^{(k)} \in \{0,1\}^2$ and $\mathbf{z}_{t,\text{present}}^{(k)} \in \{0,1\}^2$. These represent whether the object associated to the $k^{\text{th}}$ slot is moving and present on the screen, respectively. These variables have a particular relationship to the generative model, particularly the dynamics model, which allows inference over them to act as a 'pre-processing step' or filter for slot latents, before passing their continuous features further down the inference chain for use by the identity model, recurrent mixture model, and transition model. The way this gating is functionally defined is detailed in the subsection **Gated dynamics learning** below.

The latent state sequences $\mathbf{z}_{0:T,\text{presence}}^{(k)}$ and $\mathbf{z}_{0:T,\text{moving}}^{(k)}$ variables are modelled as evolving according to discrete, object-factorized markov chains which concurrently emit observations via a proxy presence variable $o_t^{(k)}$ (see below for details on how this is computed). This conditional hidden Markov model can be written as follows for the two variables:

$$p(\mathbf{z}_{0:T,\text{presence}}^{(k)}) = \prod_{t=0}^{T} p(o_t^{(k)} \mid \mathbf{z}_{t,\text{presence}}^{(k)}) p(\mathbf{z}_{t,\text{presence}}^{(k)}) \mid \mathbf{z}_{t-1,\text{presence}}^{(k)}, \theta_{\text{presence}})$$

$$p(\mathbf{z}_{0:T,\text{moving}}^{(k)}) = \prod_{t=1}^{T} p(\mathbf{z}_{t,\text{moving}}^{(k)} \mid \mathbf{z}_{t-1,\text{moving}}^{(k)}, o_t^{(k)}, \mathbf{v}_{t-1}^{(k)}, \theta_{\text{moving}}) \tag{19}$$

**Presence latent $\mathbf{z}_{t,\text{presence}}$** The presence chain uses a *time-homogeneous* $2 \times 2$ transition matrix parametrised by

$$\theta_{\text{presence}} = \{\phi_{\text{NP}\to\text{P}}, \phi_{\text{P}\to\text{NP}}\}, \qquad 0 \le \phi_{\text{NP}\to\text{P}}, \phi_{\text{P}\to\text{NP}} \le 1.$$

where the subscripts for the 'not present' and 'present' states of $\mathbf{z}_{t,\text{presence}}^{(k)}$ are abbreviated as NP and P, respectively. Writing $\phi_{i\to\text{not present}} = 1 - \phi_{i\to\text{present}}$, the transition matrix is

$$T_{\text{presence}} = \begin{pmatrix} \phi_{\text{NP}\to\text{NP}} & \phi_{\text{NP}\to\text{P}} \\ \phi_{1\to\text{NP}} & \phi_{\text{P}\to\text{P}} \end{pmatrix} = \begin{pmatrix} 1 - \phi_{\text{NP}\to\text{P}} & \phi_{\text{NP}\to\text{P}} \\ \phi_{\text{P}\to\text{NP}} & 1 - \phi_{\text{P}\to\text{NP}} \end{pmatrix}.$$

For all experiments we *fix* $\phi_{0\to1} = 0$, $\phi_{1\to0} = 0.01$, encoding the prior that an absent slot cannot spontaneously re-appear while a present one "dies out" with probability 0.01 each frame.

**Proxy observation.** We define the assignment-count indicator $o_t^{(k)}$ as a variable that indicates whether any pixels $1, 2, \ldots, N$ were assigned to slot $k$ at time $t$. This can be expressed as an element-wise product of all pixel-specific entries of the row of sMM assignment variable corresponding to slot $k$'s assignments: $\mathbf{z}_{t,k,\text{smm}}^{(1:N)}$:

$$o_t^{(k)} = \prod_{n=1}^{N} z_{t,k,\text{smm}}^{n}$$

Finally, the relationship between the count-assignments-indicator $o_t^{(k)}$ and the presence variable $\mathbf{z}_{t,\text{presence}}$ is a Bernoulli likelihood with the following form:

$$p(o_t|\mathbf{z}_{t,\text{presence}}) = o_t^{z_{t,P,\text{presence}}}(1 - o_t)^{z_{t,A,\text{presence}}} \tag{20}$$

which means that presence and the assignment count $o_t^{(k)}$ are expected to be 'on' simultaneously.

The subscripts P and A refer to the indices of $\mathbf{z}_{t,\text{presence}}^{(k)}$ that correspond to the 'present' and 'absent' indicators, respectively.

**Moving latent $\mathbf{z}_{t,\text{moving}}$** We define the speed of slot $k$ as follows (leaving out the $k$ superscript to avoid overloaded superscripts):

$$\psi_t = \sqrt{v_{t,x}^2 + v_{t,y}^2} \tag{21}$$

The transition likelihood for the moving latent $\mathbf{z}_{t,\text{moving}}^{(k)}$ depends on the presence indicator $o_t^{(k)}$ and the slot speed $\psi_{t-1}$; this forces inference of the moving indicator $\mathbf{z}_{t,\text{moving}}^{(k)}$ to be driven by the inferred speed and presence of the $k$-th slot. The form of this dependence is encoded in the $2 \times 2$ transition matrix $T_{\text{moving}}$ with parameters $\theta_{\text{moving}}$ as follows:

$$\theta_{\text{moving}} = \{\lambda, \beta\}, \qquad 0 \le \lambda \le 1, \ 0 \le \beta \le 1, \ \lambda + \beta \le 1. \tag{22}$$

The parameters $\lambda$ and $\beta$ determine the two conditional probabilities $\phi_{i\to\text{M}}(o_t^{(k)}, \psi_{t-1})$ and $\phi_{i\to\text{NM}}(o_t^{(k)}, \psi_{t-1})$. We use the subscripts NM, M to indicate the 'not-moving' and 'moving' states of $\mathbf{z}_{t,\text{moving}}^{(k)}$, respectively. The dependence of the conditional probabilities on the $\lambda, \beta$ hyperparameters can be written as follows:

$$\phi_{i\to\text{M}}(o_t^{(k)}, \psi_{t-1}) = \begin{cases} \lambda\, i + \beta\, \psi_{t-1}, & o_t^{(k)} = 1, \\ i, & o_t^{(k)} = 0, \end{cases} \qquad \phi_{i\to\text{NM}}(o_t^{(k)}, \psi_{t-1}) = 1 - \phi_{i\to 1}(o_t^{(k)}, \psi_{t-1}).$$

The full transition matrix $T_{\text{moving}}$ can be written:

$$T_{\text{moving}}\big(o_t^{(k)}, \psi_{t-1}; \theta_{\text{moving}}\big) = \begin{pmatrix} \phi_{\text{NM}\to\text{NM}} & \phi_{\text{NM}\to\text{M}} \\ \phi_{\text{M}\to\text{NM}} & \phi_{\text{M}\to\text{M}} \end{pmatrix}. \tag{23}$$

This sort of parameterization results in the following interpretation: if the slot is inferred to be absent (i.e., no pixels are assigned to it and $o_t^{(k)} = 0$), $\mathbf{z}_{t,\text{moving}}^{(k)}$ stays in its previous state. However, if the slot is inferred to be present ($o_t^{(k)} = 1$), then the previous "moving" probability is shrunk by $\lambda$ and nudged upward by $\beta\psi_{t-1}$.

In all experiments we set $\lambda = 0.99$, $\beta = 0.01$, but they remain exposed hyperparameters.

**Gated dynamics learning**   The presence and moving latents $\mathbf{z}_{t,\text{presence}}, \mathbf{z}_{t,\text{moving}}$ exist in order to filter which slots get fit by the rMM. In order to achieve this selective routing of only active, moving slots, we introduce an auxiliary gate variable that is connected to the moving- and presence-latents via a multiplication factor that parameterizes a Bernoulli likelihood over the two values ('ON' and 'OFF') of the gate variable $\mathcal{G}_t^{(k)}$:

$$p(\mathcal{G}_t^{(k)} \mid \mathbf{z}_{t,\text{moving}}^{(k)} \mathbf{z}_{t,\text{present}}^{(k)}) = \text{Bernoulli}\big(p_{\text{gate}}\big) \tag{24}$$
$$\text{where} \quad p_{\text{gate}} = z_{t,\text{M,moving}}^{(k)} z_{t,\text{P,present}}^{(k)}$$

This binary gate variable then modulates the input precision of the various likelihoods associated with the identity model (iMM), transition mixture model (tMM), and recurrent mixture model (rMM) to effectively 'mask' the learning of these models on untracked or absent slots. The end effect is that slots which are inferred to be *moving* and *present* keep full precision, while any other combination deflates the slot-specific input covariance to $0$, removing the influence of their sufficient statistics from parameter learning.

A.4   INTERACTION VARIABLE

We also associate each object with a discrete latent variable $\mathbf{z}_{t,\text{interacting}}^{(k)}$ which indicates the type of the closest object interacting with the focal object (i.e., that indexed by $k$). In practice, we infer this interaction variable by finding the object whose position variable is closest to the focal slot, i.e. $\arg\min_{j\in\text{nearest}} \|\mathbf{p}_j - \mathbf{p}_l\|$, within some constrained set of 'nearest' objects whose position latents are within a predefined interaction radius of the focal object (determined by a fixed parameter $r_{\text{min}}$). This interaction radius can be tuned on a game specific basis – see the main text results in Section 4 for how fixing this parameter affects the results. We then perform inference on the identity model using the continuous features of that nearest-interacting slot: $\big[\mathbf{c}_t^{(j)}, \mathbf{e}_t^{(j)}\big]^\top$. The inferred identity of the resulting $j^{\text{th}}$ slot is then converted into a one-hot vector representing the type of the 'interacting' latent $\mathbf{z}_{t,\text{interacting}}^{(k)}$, which can then be fed as input into the recurrent mixture model or rMM as described in the following section.

A.5   UNUSED COUNTER

To keep track of how long a slot has remained inactive we introduce a non-negative-integer latent that is treated as a continuous variable in $\mathbb{R}$: $u_t^{(k)}$ or the 'unused counter'. This allows the model to predict the respawning of objects after they go off-screen. We couple the unused counter again to the proxy assignment-count variable $o_t^{(k)}$ using an exponentially–decaying Bernoulli likelihood (identical in spirit to the EP–style presence likelihood):

$$P\big(o_t^{(k)} = 1 \mid u_t^{(k)}\big) = 1 - \exp\big\{-\xi\, e^{-\gamma_u\, u_t^{(k)}}\big\},$$

$$P\big(o_t^{(k)} = 0 \mid u_t^{(k)}\big) = \exp\big\{-\xi\, e^{-\gamma_u\, u_t^{(k)}}\big\}, \qquad \xi \in (0,1],\ \gamma_u > 0. \tag{25}$$

When $u_t^{(k)} = 0$ the slot is present with probability $1 - e^{-\xi} \simeq \xi$ (for typical $\xi \gtrsim 0.8$). Each increment by $\nu_u$ multiplies that probability by $\exp\big(-\xi\,\gamma_u\,\nu_u\big)$, i.e. it decays roughly one $e$-fold per unused step when $\gamma_u \simeq \nu_u^{-1}$.

## A.6 IDENTITY MIXTURE MODEL

AXIOM uses an *identity mixture model* (iMM) to infer a discrete identity code $\mathbf{z}_{\text{type}}^{(k)}$ for each object based on its continuous features. These identity codes are used to condition the inference of the recurrent mixture model used for dynamics prediction. Conditioning the dynamics on identity-codes in this way, rather than learning a separate dynamics model for each slot, allows AXIOM to use the same dynamics model across slots. This also enables the model to learn the same dynamics in a *type*-specific, rather than *instance*-specific, manner, and to remap identities when e.g., the environment is perturbed and colors change. Concretely, the iMM models the 5-D colors and shapes $\{\mathbf{c}^{(k)}, \mathbf{e}^{(k)}\}_{k=1}^{K}$ across slots as a mixture of up to $V$ Gaussian components (object types). The slot-level assignment variable $\mathbf{z}_{t,\text{type}}^{(k)}$ indicates which identity is assigned to the $k^{\text{th}}$ slot. The generative model for the iMM is (omitting the $t-1$ subscript from object latents):

$$p\big(\big[\mathbf{c}^{(k)}, \mathbf{e}^{(k)}\big]^{\top} \mid \mathbf{z}_{\text{type}}^{(k)}, \boldsymbol{\mu}_{1:V,\text{type}}, \boldsymbol{\Sigma}_{1:V,\text{type}}\big) = \prod_{j=1}^{V} \mathcal{N}\big(\boldsymbol{\mu}_{j,\text{type}}, \big(\mathcal{G}_t^{(k)}\big)^{-1} \boldsymbol{\Sigma}_{j,\text{type}}\big)^{z_{j,\text{type}}^{(k)}} \tag{26}$$

$$p\big(\boldsymbol{\mu}_{j,\text{type}}, \boldsymbol{\Sigma}_{j,\text{type}}^{-1}\big) = \text{NIW}(\mathbf{m}_{0,j,\text{type}}, \kappa_{0,j,\text{type}}, \mathbf{U}_{0,j,\text{type}}, n_{0,j,\text{type}}) \tag{27}$$

$$p\big(\mathbf{z}_{\text{type}}^{(k)} \mid \boldsymbol{\pi}_{\text{type}}\big) = \text{Cat}(\boldsymbol{\pi}_{\text{type}}),\ p(\boldsymbol{\pi}_{\text{type}}) = \text{Dir}\big(\underbrace{1, \ldots, 1}_{V-1\ \text{times}}, \alpha_{0,\text{type}}\big) \tag{28}$$

$$p(\boldsymbol{\Theta}_{\text{imm}}) = p(\boldsymbol{\mu}_{1:V,\text{types}}, \boldsymbol{\Sigma}_{1:V,\text{type}}) p(\boldsymbol{\pi}_{\text{type}}) \tag{29}$$

The $\big(\mathcal{G}_t^{(k)}\big)^{-1} \mathbf{X}$ notation represents element-wise broadcasting of the reciprocal of $\mathcal{G}_t^{(k)}$ across all elements of the matrix or vector $\mathbf{X}$. When applied as a mask to a covariance matrix, for instance, the effect is that slots that are inferred to be moving and present do not have their covariance affected $\big(\mathcal{G}_t^{(k)}\big)^{-1} = 1$, whereas those slots that are inferred to be either not present or not moving will 'inflate' the covariance of the mixture model, due to $\big(\mathcal{G}_t^{(k)}\big)^{-1} \to \infty$. The same type of Categorical likelihood for the type assignments and truncated stick-breaking prior over the mixture weights is used to allow an arbitrary (up to a maximum of $V$) number of types to be used to explain the continuous slot features. We equip the prior over the component likelihood parameters with conjugate Normal Inverse Wishart (NIW) priors.

## A.7 TRANSITION MIXTURE MODEL

The dynamics of each slot are modelled as a mixture of linear functions of the slot's own previous state. To stress the homology between this model and the other modules of AXIOM, we refer to this module as the *transition mixture model* or tMM, but this formulation is more commonly also known as a switching linear dynamical system or SLDS (Ghahramani & Hinton, 1996). The tMM's switch variable $\mathbf{s}_{t,\text{tmm}}^{(k)}$ selects a set of linear parameters $\boldsymbol{D}_l, \boldsymbol{b}_l$ to describe the $k^{\text{th}}$ slot's trajectory from $t$ to $t+1$. Each linear system captures a distinct rigid motion pattern for a particular object:

$$p(\mathbf{x}_t^{(k)} \mid \mathbf{x}_{t-1}^{(k)}, \mathbf{s}_{t,\text{tmm}}^{(k)}, \boldsymbol{D}_{1:L}, \boldsymbol{b}_{1:L}) = \prod_{l=1}^{L} \mathcal{N}\big(\boldsymbol{D}_l \mathbf{x}_t^{(k)} + \boldsymbol{b}_l, \big(\mathcal{G}_t^{(k)}\big)^{-1}, \boldsymbol{\Sigma}_{1:L}\big)^{s_{t,l,\text{tmm}}^{(k)}} \tag{30}$$

$$p(\mathbf{s}_{t,\text{tmm}}^{(k)} \mid \boldsymbol{\pi}_{\text{tmm}}) = \text{Cat}(\boldsymbol{\pi}_{\text{tmm}}), \tag{31}$$

$$p(\boldsymbol{\pi}_{\text{tmm}}) = \text{Dir}\big(\underbrace{1, \ldots, 1}_{L-1 \text{ times}}, \alpha_{0,\text{tmm}}\big), \tag{32}$$

$$p(\boldsymbol{D}_{1:L}, \boldsymbol{b}_{1:L}) = \prod_{l=1}^{L} p(\boldsymbol{D}_l)\, p(\boldsymbol{b}_l), \tag{33}$$

$$p(\boldsymbol{D}_l) \propto 1, \qquad p(\boldsymbol{b}_l) \propto 1, \tag{34}$$

$$p(\boldsymbol{\Sigma}_{1:L}) = \prod_{l=1}^{L} \delta(\boldsymbol{\Sigma}_l - 2\mathbf{I}), \tag{35}$$

$$p(\boldsymbol{\Theta}_{\text{tmm}}) = p(\boldsymbol{\pi}_{\text{tmm}}) p(\boldsymbol{D}_{1:L}, \boldsymbol{b}_{1:L}) p(\boldsymbol{\Sigma}_{1:L}) \tag{36}$$

where we fix the covariance of all $L$ components to be $2I$, and all mixture likelihoods $\boldsymbol{D}_{1:L}, \boldsymbol{b}_{1:L}$ to have uniform priors. Note that we gate the covariance once again using the reciprocal of $\mathcal{G}_t^{(k)}$ to filter the tMM to only model objects that are moving and present. The truncated stick-breaking prior $\text{Dir}\big(1, \ldots, 1, \alpha_{0,\text{tmm}}\big)$ over the component mixing weights enables the number of linear modes $L$ to be dynamically adjusted to the data by growing the model with propensity $\alpha_{0,\text{tmm}}$. Importantly, the $L$ transition components of the tMM are not slot-dependent, but are shared and thus learned using the data from all $K$ slot latents. The tMM can thus explain and predict the motion of different objects using a shared, expanding set of (up to $L$ distinct) linear dynamical systems.

## A.8 RECURRENT MIXTURE MODEL

The *recurrent mixture model* (rMM) is used to infer the switch states of the transition model directly from current slot-level features. This dependence of switch states on continuous features is the same construct used in the recurrent switching linear dynamical system or rSLDS (Linderman et al., 2016). However, in contrast to the rSLDS, which uses a discriminative mapping to infer the switch state from the continuous state (usually a softmax or stick-breaking parameterization thereof), the rMM recovers this dependence *generatively* using a mixture model over mixed continuous–discrete slot states (Bishop & Lasserre, 2007). In this way, 'selecting' the switch state used for conditioning the tMM actually emerges from *inference* over discrete latent variables, which have a particular conditional relationship (in this context, a joint mixture likelihood relationship) to other latent and observed variables. Concretely, the rMM models the distribution of continuous and discrete variables as a mixture model driven by another per-slot latent assignment variable $\mathbf{s}_{\text{rmm}}^{(k)}$. The rMM defines a mixture likelihood over a tuple of continuous and discrete slot-specific information. We use the notation $f_{t-1}^{(k)}$ to collect the continuous features that the rMM parameterizes a density over: they include a subset of the $k^{\text{th}}$ slot's own continuous state $\mathbf{x}_{t-1}^{(k)}$, as well as a nonlinear transformation $g$ applied to other slots' features, that computes the 2-D distance vector pointing from the inferred position of the focal object (i.e., slot $k$) to the nearest interacting slot $j$. We detail how these features are computed below:

**Continuous features for the rMM** The continuous latent dimensions of $\mathbf{x}_{t-1}^{(k)}$ used for the rMM include the following: $\mathbf{p}_{t-1}^{(k)}, \mathbf{v}_{t-1}^{(k)}, u_{t-1}^{(k)}$. We represent extracting this subset as a sparse linear projection applied to the full continuous latents $C\mathbf{x}_{t-1}^{(k)}$. In addition, the rMM models the distribution of the 2-D vectors pointing from the focal object's position (the slot $k$ in consideration) to the position of the nearest interacting object, i.e. $\Delta\mathbf{p}_{t-1}^{(k)} \equiv \mathbf{p}_{t-1}^{(j)} - \mathbf{p}_{t-1}^{(k)}$. The nearest interacting object with index $j$ is the one whose inferred position is the closest to the focal object's, while only considering neighbors within some interaction zone with radius $r_{\text{min}}$. This is the same interaction distance used to compute nearest-neighbors when populating the $\mathbf{z}_{\text{interacting}}^{(k)}$ latent, see Appendix A.4 for details. If no object is within the interaction radius, then we set $\Delta\mathbf{p}_{t-1}^{(k)}$ to a random sample from a 2-D

uniform distribution with mean $[1.2, 1.2]$ and lower/upper bounds of $[1.198, 1.198]$, $[1.202, 1.202]$. We summarize this computation with a generic nonlinear function applied to the latent states of all slots $g(\mathbf{x}_{t-1}^{(1:K)})$, to intimate possible generalizations where different sorts of (possibly learnable) neighborhood relationships could be inserted here.

**Discrete features for the rMM**  The discrete features include the following: $\mathbf{z}_{t-1,\text{type}}^{(k)}$, $\mathbf{z}_{t-1,\text{interacting}}^{(k)}$, the assignment-count indicator variable $o_{t-1}^{(k)}$, the switch state of the transition mixture model $\mathbf{s}_{t,\text{tmm}}^{(k)}$, the action at timestep $t-1$ and reward at the current timestep $r_t$. We refer to this discrete collection as $d_{t-1}^{(k)}$. The inclusion of $\mathbf{s}_{t,\text{tmm}}^{(k)}$ in the discrete inputs to the rMM is a critical ingredient of this recurrent dynamics formulation – it allows inference on this the switching state of the tMM to be driven by high-dimensional configurations of continuous and discrete variables relevant for predicting motion.

**rMM description**  The rMM assignment variable associated to a given slot is a binary vector $\mathbf{s}_{t,\text{rmm}}^{(k)}$ whose $m^{\text{th}}$ entry $s_{t,m,\text{rmm}}^{(k)} \in \{0,1\}$ indicates whether component $m$ explains the current tuple of mixed continuous-discrete data. Each component likelihood selected by $\mathbf{s}_{t,\text{rmm}}^{(k)}$ factorizes into a product of continuous (Gaussian) and discrete (Categorical) likelihoods.

$$f_{t-1}^{(k)} = \left(C\mathbf{x}_{t-1}^{(k)}, g(\mathbf{x}_{t-1}^{(1:K)})\right), \quad d_{t-1}^{(k)} = \left(\mathbf{z}_{t-1,\text{type}}^{(k)}, \mathbf{z}_{t-1,\text{interacting}}^{(k)}, o_{t-1}^{(k)}, \mathbf{s}_{t,\text{tmm}}^{(k)}, a_{t-1}, r_t\right) \tag{37}$$

$$p(f_{t-1}^{(k)}, d_{t-1}^{(k)} \mid \mathbf{s}_{t,\text{rmm}}^{(k)}) = \prod_{m=1}^{M} \left[\mathcal{N}\left(f_{t-1}^{(k)}; \boldsymbol{\mu}_{m,\text{rmm}}, \left(\mathcal{G}_t^{(k)}\right)^{-1}\boldsymbol{\Sigma}_{m,\text{rmm}}\right) \prod_i \text{Cat}\left(d_{t-1,i}; \mathcal{G}_t^{(k)}\mathbf{b}_{m,i}\right)\right]^{s_{t,m,\text{rmm}}} \tag{38}$$

$$p(\boldsymbol{\mu}_{m,\text{rmm}}, \boldsymbol{\Sigma}_{m,\text{rmm}}^{-1}) = \text{NIW}(\mathbf{m}_{0,m,\text{rmm}}, \kappa_{0,m,\text{rmm}}, \mathbf{U}_{0,m,\text{rmm}}, n_{0,m,\text{rmm}}), \; p(\mathbf{b}_{m,i}) = \text{Dir}(\boldsymbol{\beta}_{0,m,i}) \tag{39}$$

$$p(\mathbf{s}_{t,\text{rmm}}^{(k)} \mid \boldsymbol{\pi}_{\text{rmm}}) = \text{Cat}(\boldsymbol{\pi}_{\text{rmm}}), \; p(\boldsymbol{\pi}_{\text{rmm}}) = \text{Dir}(\underbrace{1, \ldots, 1}_{M-1 \text{ times}}, \alpha_{0,\text{rmm}}) \tag{40}$$

$$p(\boldsymbol{\Theta}_{\text{rmm}}) = p(\boldsymbol{\mu}_{1:M,\text{rmm}}, \boldsymbol{\Sigma}_{1:M,\text{rmm}}^{-1})p(\mathbf{b}_{1:M})p(\boldsymbol{\pi}_{\text{rmm}}) \tag{41}$$

The parameters of the multivariate normal components are equipped with NIW priors and those of the discrete Categorical likelihoods with Dirichlet priors. As with all the other modules of AXIOM, we equip the mixing weights for $\mathbf{s}_{t,\text{rmm}}^{(k)}$ with a truncated stick-breaking prior whose final $M^{\text{th}}$ pseudocount parameter tunes the propensity to add new rMM components. Note also the use of the gate variable $\mathcal{G}_t^{(k)}$ to filter slots for dynamics learning by inflating the covariance associated with any slot inputs not inferred moving and present.

**Fixed distance variant.**  We explored a variant of the rMM (`fixed_distance`) where the displacement vector $\Delta\mathbf{p}_{t-1}^{(k)}$ is not returned by $g(\mathbf{x}_{t-1}^{(1:K)})$ and therefore not included as one of the continuous input features for the rMM. In this case, the entry of $\mathbf{z}_{t-1,\text{interacting}}^{(k)}$ that corresponds to the nearest interacting object is still determined by $r_{\min}$, however. In this case, the choice of $r_{\min}$ matters more for performance because the rMM cannot learn to nuance its dynamic predictions based on precise distances. In general, this means that $r_{\min}$ requires more game-specific tuning. See Section 4 for the results of tuning $r_{\min}$ compared to learning it directly by providing $\Delta\mathbf{p}_{t-1}^{(k)}$ as input to the rMM.

## A.9 VARIATIONAL INFERENCE AND LEARNING

To perform inference and learning within our proposed model, we employ a variational Bayesian approach. The core idea is to approximate the true posterior distribution over latent variables and parameters with a more tractable factorized distribution, $q(\mathcal{Z}_{0:T}, \tilde{\boldsymbol{\Theta}})$. This is achieved by optimizing the variational free-energy functional, $\mathcal{F}(q)$, which establishes an upper-bound on the negative log-marginal likelihood of the observed data $\mathbf{y}_{0:T}$:

$$\mathcal{F}(q) = \mathbb{E}_q\Big[\log q(\mathcal{Z}_{0:T}, \tilde{\boldsymbol{\Theta}}) - \log p(\mathbf{y}_{0:T}, \mathcal{Z}_{0:T}, \tilde{\boldsymbol{\Theta}})\Big], \quad \text{such that} \quad \mathcal{F}(q) \geq -\log p(\mathbf{y}_{0:T}). \tag{42}$$

Minimizing this functional $\mathcal{F}(q)$ with respect to $q$ is equivalent to maximizing the Evidence Lower Bound (ELBO), thereby driving the approximate posterior $q(\mathcal{Z}_{0:T}, \tilde{\boldsymbol{\Theta}})$ to more closely resemble the true posterior $p(\mathcal{Z}_{0:T}, \tilde{\boldsymbol{\Theta}} \mid \mathbf{y}_{0:T})$.

We assume a mean-field factorization for the approximate posterior, which decomposes over the global parameters $\tilde{\boldsymbol{\Theta}}$ and the sequence of latent variables $\mathcal{Z}_{0:T}$:

$$q(\mathcal{Z}_{0:T}, \tilde{\boldsymbol{\Theta}}) = q(\tilde{\boldsymbol{\Theta}}) \prod_{t=0}^{T} q(\mathcal{Z}_t), \tag{43}$$

where $q(\mathcal{Z}_t)$ further factorizes across individual latent variables for frame $t$:

$$q(\mathcal{Z}_t) = \left( \prod_{n=1}^{N} q(\mathbf{z}_{t,\text{smm}}^n) \right) \prod_{k=1}^{K} \left( q(\mathbf{x}_t^{(k)}) \, q(\mathbf{z}_t^{(k)}) \, q(\mathbf{s}_t^{(k)}) \right). \tag{44}$$

The variational distribution over the global parameters $\tilde{\boldsymbol{\Theta}}$ is also assumed to factorize according to the distinct components of our model:

$$q(\tilde{\boldsymbol{\Theta}}) = q(\boldsymbol{\Theta}_{\text{smm}}) \, q(\boldsymbol{\Theta}_{\text{imm}}) \, q(\boldsymbol{\Theta}_{\text{tmm}}) \, q(\boldsymbol{\Theta}_{\text{rmm}}). \tag{45}$$

Note that the pixel-to-slot assignment variables $\mathbf{z}_{t,\text{smm}}^n$ are specific to each pixel $n$ but are not factorized across slots for a given pixel, as they represent a single categorical choice from $K$ slots. Other latent variables, such as the continuous state $\mathbf{x}_t^{(1:K)}$ and discrete attributes $\mathbf{z}_t^{(1:K)}, \mathbf{s}_t^{(1:K)}$, are factorized per slot $k$.

The inference and learning procedure for each new frame $t$ involves an iterative alternation between an E-step, where local latent variable posteriors are updated, and an M-step, where global parameter posteriors are refined.

**E-step** In the Expectation-step (E-step), we hold the variational posteriors of the global parameters $q(\boldsymbol{\Theta})$ fixed. We then update the variational posteriors for each local latent variable factor within $q(\mathcal{Z}_t)$, such as $q(\mathbf{z}_{t,\text{smm}}^n)$, $q(\mathbf{x}_t^{(k)})$, $q(\mathbf{z}_t^{(k)})$, and $q(\mathbf{s}_t^{(k)})$. These updates are derived by optimizing the ELBO with respect to each factor in turn and often result in closed-form coordinate-ascent updates due to conjugacy between the likelihood terms and the chosen forms of the variational distributions.

**M-step** In the Maximization-step (M-step), we update the variational posteriors for the global parameters $q(\boldsymbol{\Theta}_\mu)$ associated with each model component $\mu \in \{\text{sMM}, \text{iMM}, \text{tMM}, \text{rMM}\}$. Each $q(\boldsymbol{\Theta}_\mu)$ is assumed to belong to the exponential family, characterized by natural parameters $\eta_\mu$:

$$q(\boldsymbol{\Theta}_\mu) = h(\boldsymbol{\Theta}_\mu) \exp\{\eta_\mu^\top T(\boldsymbol{\Theta}_\mu) - A(\eta_\mu)\}, \tag{46}$$

where $T(\boldsymbol{\Theta}_\mu)$ represents the sufficient statistics for $\boldsymbol{\Theta}_\mu$, and $A(\eta_\mu)$ is the log-partition function (or log-normalizer). The M-step update proceeds in two stages for each component:

1. First, we compute the *expected sufficient statistics* $\widehat{T}_\mu$ using the current posteriors over the latent variables $q(\mathcal{Z}_t)$ obtained from the $t$-th E-step:

$$\widehat{T}_\mu = \mathbb{E}_{q(\mathcal{Z}_t)}\big[T(\boldsymbol{\Theta}_\mu, \mathcal{Z}_t)\big]. \tag{47}$$

   These expected statistics are then combined with prior natural parameters $\eta_{\mu,0}$ to form the target natural parameters for the update: $\widehat{\eta}_\mu = \widehat{T}_\mu + \eta_{\mu,0}$.

2. Second, we update the current natural parameters $\eta_\mu^{(t-1)}$ using a *natural-gradient* step, which acts as a stochastic update blending the previous parameters with the new target parameters, controlled by a learning rate schedule $\rho_t$:

$$\eta_\mu^{(t)} \leftarrow (1 - \rho_t)\, \eta_\mu^{(t-1)} + \rho_t\, \widehat{\eta}_\mu, \qquad \text{where } 0 < \rho_t \leq 1. \tag{48}$$

SLOT MIXTURE MODEL (sMM)

The Slot Mixture Model (sMM) provides a likelihood for the observed pixel data $\mathbf{y}_t$ by modeling each pixel as originating from one of $K$ object slots. The variational approximation involves posteriors over pixel-to-slot assignments $\mathbf{z}_{t,\text{smm}}^n$, slot mixing weights $\boldsymbol{\pi}_{\text{smm}}$, slot-specific color variances $\sigma_{c,j}^{(k)}$, and the continuous latent states of slots $\mathbf{x}_t^{(k)}$. Specifically, for each pixel $n$ at time $t$, the posterior probability that it belongs to slot $k$ is $q(z_{t,k,\text{smm}}^n = 1) = r_{t,k}^n$, ensuring that $\sum_{k=1}^K r_{t,k}^n = 1$. The posterior over the slot-mixing probabilities $\boldsymbol{\pi}_{\text{smm}}$ is a Dirichlet distribution: $q(\boldsymbol{\pi}_{\text{smm}}) = \text{Dir}(\boldsymbol{\pi}_{\text{smm}} \mid \alpha_{1,\text{smm}}, \ldots, \alpha_{K,\text{smm}})$, parameterized by concentrations $\alpha_{k,\text{smm}} > 0$. For each slot $k$ and color channel $j \in \{r, g, b\}$, the posterior over the color variance $\sigma_{c,j}^{(k)}$ is a Gamma distribution: $q(\sigma_{c,j}^{(k)}) = \text{Gamma}(\sigma_{c,j}^{(k)} \mid \gamma_{k,j}, b_{k,j})$, with shape $\gamma_{k,j}$ and rate $b_{k,j}$. Finally, each slot's continuous latent state $\mathbf{x}_t^{(k)} \in \mathbb{R}^{10}$ (encompassing position, color, velocity, shape, and the unused counter) is modeled by a Gaussian distribution: $q(\mathbf{x}_t^{(k)}) = \mathcal{N}(\mathbf{x}_t^{(k)} \mid \mu_t^{(k)}, \Sigma_t^{(k)})$. The precision matrix is denoted $\Lambda = \Sigma^{-1}$, and the precision-weighted mean is $h_t^{(k)} = \Lambda_t^{(k)} \mu_t^{(k)}$.

**E-step Updates for sMM**  During the E-step for the sMM at frame $t$, the variational distributions for local latent variables $\mathbf{z}_{t,\text{smm}}^n$ (represented by the responsibilities $r_{t,k}^n$) and $\mathbf{x}_t^{(k)}$ (represented by $\mu_t^{(k)}, \Sigma_t^{(k)}$) are updated, while the global sMM parameters $q(\boldsymbol{\Theta}_{\text{smm}})$ are held fixed.

1. The pixel responsibilities $r_{t,k}^n$, representing $q(z_{t,k,\text{smm}}^n = 1)$, are updated using the standard mixture model update:

$$r_{t,k}^n = \frac{\exp\big(\mathbb{E}_q[\log \pi_{k,\text{smm}}] + \mathbb{E}_q[\log \mathcal{N}(\mathbf{y}_t^n; A\mathbf{x}_t^{(k)}, \boldsymbol{\Sigma}^{(k)})]\big)}{\sum_{j=1}^K \exp\big(\mathbb{E}_q[\log \pi_{j,\text{smm}}] + \mathbb{E}_q[\log \mathcal{N}(\mathbf{y}_t^n; A\mathbf{x}_t^{(j)}, \boldsymbol{\Sigma}^{(j)})]\big)}. \tag{49}$$

The per-slot observation covariance $\boldsymbol{\Sigma}^{(k)}$ is constructed as $\boldsymbol{\Sigma}^{(k)} = \text{diag}\big(B\,\mathbb{E}_q[\mathbf{x}_t^{(k)}], \ \mathbb{E}_q[\sigma_c^{(k)}]\big)$, consistent with the generative model where $B\mathbf{x}^{(k)}$ provides variance related to shape and $\sigma_c^{(k)}$ provides color channel variances (see Equation (12) for the sMM likelihood equations).

2. The parameters of the Gaussian posterior $q(\mathbf{x}_t^{(k)})$ are updated by incorporating evidence from the pixels assigned to slot $k$. This involves updating its natural parameters (precision $\Lambda_t^{(k)}$ and precision-adjusted mean $h_t^{(k)}$):

$$\Lambda_t^{(k)} = \Lambda_{t|t-1}^{(k)} + \sum_{n=1}^N r_{t,k}^n\, A^\top \big(\Sigma^{(k)}\big)^{-1} A,$$

$$h_t^{(k)} = h_{t|t-1}^{(k)} + \sum_{n=1}^N r_{t,k}^n\, A^\top \big(\Sigma^{(k)}\big)^{-1} \mathbf{y}_t^n. \tag{50}$$

The terms $\Lambda_{t|t-1}^{(k)}$ and $h_{t|t-1}^{(k)}$ are the natural parameters of the predictive distribution for $\mathbf{x}_t^{(k)}$.

The standard parameters are then $\Sigma_t^{(k)} = \big(\Lambda_t^{(k)}\big)^{-1}$ and $\mu_t^{(k)} = \Sigma_t^{(k)} h_t^{(k)}$.

**M-step Updates for sMM**  In the M-step, the global parameters of the sMM, which are the Dirichlet parameters $\alpha_{k,\text{smm}}$ for mixing weights and the Gamma parameters $(\gamma_{k,j}, b_{k,j})$ for color variances, are updated. This begins by accumulating the expected sufficient statistics from the E-step:

$$N_{t,k} = \sum_{n=1}^N r_{t,k}^n,$$

$$S_{1,t,k}^{\mathbf{y}} = \sum_{n=1}^N r_{t,k}^n\, \mathbf{y}_t^n, \tag{51}$$

$$S_{2,t,k}^{\mathbf{y}} = \sum_{n=1}^N r_{t,k}^n\, \mathbf{y}_t^n\, (\mathbf{y}_t^n)^\top.$$

The Dirichlet concentration parameters are updated as (now using the $t$ index to represent the current vs. last settings of the posterior parameters):

$$
\begin{aligned}
\alpha_{t,k,\mathrm{smm}} &= (1 - \rho_t)\,\alpha_{t-1,k,\mathrm{smm}} + \rho_t\,(\alpha_{0,k,\mathrm{smm}} + N_{t,k}), \\
\gamma_{t,k,j} &= (1 - \rho_t)\,\gamma_{t-1,k,j} + \rho_t\left(\gamma_{0,j} + \frac{N_{t,k}}{2}\right), \\
b_{t,k,j} &= (1 - \rho_t)\,b_{t-1,k,j} + \rho_t\left(1 + \frac{1}{2}\sum_{n=1}^{N} r_{t,k}^n (\mathbf{y}_{t,\mathrm{color}\,j}^n - (A\mathbb{E}_q[\mathbf{x}_t^{(k)}])_{\mathrm{color}\,j})^2\right).
\end{aligned}
\tag{52}
$$

Here, $\alpha_{0,k,\mathrm{smm}}$ represents the prior concentration for the Dirichlet distribution (e.g., 1 for the first $K-1$ components and $\alpha_{0,\mathrm{smm}}$ for the $K$-th, if using a truncated stick-breaking prior). For the Gamma parameters, $\gamma_{0,j}$ is the prior shape and 1 is the prior rate (or related prior parameters). The projection matrices $A$ and $B$ are considered fixed and are not learned.

PRESENCE, MOTION, AND UNUSED COUNTER DYNAMICS

The model includes latent variables for each slot $k$ that track its presence $\mathbf{z}_{t,\mathrm{presence}}^{(k)}$, motion $\mathbf{z}_{t,\mathrm{moving}}^{(k)}$, and an unused counter $u_t^{(k)}$.

**Inference over the presence latent**   The presence state is informed by an 'assignment-count-indicator' $o_t^{(k)}$. This indicator is set to 1 if the slot is actively explaining pixels (e.g., if the sum of its responsibilities $\sum_n r_{t,k}^n$ exceeds a small threshold $\epsilon_{\mathrm{active}}$), and 0 otherwise.

Recall the Bernoulli likelihood over $o_t^{(k)}$ that links it to the $\mathbf{z}_{t,\mathrm{presence}}^{(k)}$ latent as follows (cf. Equation (20)):

$$
p(o_t|\mathbf{z}_{t,\mathrm{presence}}) = o_t^{z_{t,P,\mathrm{presence}}}(1 - o_t)^{z_{t,A,\mathrm{presence}}}
\tag{53}
$$

There is an implied superscript $k$ on both $o_t^{(k)}$ and the presence variable $z_{t,P,\mathrm{presence}}^{(k)}$, which are left out to avoid visual clutter. This linkage is incorporated into $q(\mathbf{z}_{t,\mathrm{presence}}^{(k)})$ using an Expectation Propagation (EP) style update via a pseudo-likelihood (Minka, 2013):

$$
\tilde{\ell}\big(\mathbf{z}_{t,\mathrm{presence}}^{(k)}\big) = \left[(o_t^{(k)})^{z_{t,P,\mathrm{presence}}^{(k)}}\big(1 - o_t^{(k)}\big)^{z_{t,A,\mathrm{presence}}^{(k)}}\right]^{\zeta},
\tag{54}
$$

where $\zeta$ is a damping factor. This update increases posterior evidence for presence if $o_t^{(k)} = 1$, and for absence if $o_t^{(k)} = 0$. The first-order effect of this pseudo-likelihood when updating the posterior over $\mathbf{z}_{t,\mathrm{presence}}^{(k)}$ is

$$
q(z_{t,\mathrm{P,presence}}^{(k)}) \approx (1 - \zeta)q(z_{t-1,\mathrm{P,presence}}^{(k)}) + \zeta\,o_t^{(k)}.
\tag{55}
$$

Recall that the $A, P$ subscripts refer to the indices of $\mathbf{z}_{t-1,\mathrm{presence}}^{(k)}$ that signify the 'is-absent' and 'is-present' states, respectively.

**Inference over the moving latent**   Similar EP updates apply for inferring $q(z_{t,M,\mathrm{moving}}^{(k)})$ based on its specific likelihoods involving velocity and $o_t^{(k)}$. The gate $\mathcal{G}_t^{(k)} = q(z_{t,P,\mathrm{present}}^{(k)} = 1)\cdot q(z_{t,M,\mathrm{moving}}^{(k)} = 1)$ is then formed from these inferred probabilities.

**Inference over the unused counter**   The 'unused counter' $u_t^{(k)}$ tracks how long slot $k$ has been inactive. Appendix A.5 of the full model details describes a generative likelihood $P(o_t^{(k)} = 1 \mid u_t^{(k)}) = 1 - \exp(-\xi e^{-\gamma_u u_t^{(k)}})$, for which damped EP updates for $q(u_t^{(k)})$ can be derived and would remain in closed form. However, a specific case, by choosing hyperparameters such that a hard constraint $o_t^{(k)} = 1 \iff u_t^{(k)} = 0$ is effectively enforced (e.g., by taking $\gamma_u \to \infty$ and $\xi = 1$

in the generative likelihood), leads to a simplified, deterministic update for the posterior mean $\mu_{t,u}^{(k)} = \mathbb{E}_q[u_t^{(k)}]$:

$$\mu_{t,u}^{(k)} = \left(1 - o_t^{(k)}\right)\left(\mu_{t-1,u}^{(k)} + \nu_u\right), \qquad \nu_u = 0.05. \tag{56}$$

In this simplified regime, the counter is reset to 0 if $o_t^{(k)} = 1$; otherwise, it increments by a fixed amount $\nu_u$.

IDENTITY MIXTURE MODEL (iMM)

The Identity Mixture Model (iMM) assigns one of $V$ possible discrete identities to each active slot, based on its continuous features. This allows for shared characteristics and dynamics across instances of the same object type. The variational approximation targets posteriors over these slot-to-identity assignments $\mathbf{z}_{t,\text{type}}^{(k)}$ (which is a component of $\mathbf{z}_t^{(k)}$), identity mixing weights $\boldsymbol{\pi}_{\text{type}}$, and the parameters $(\boldsymbol{\mu}_{j,\text{type}}, \boldsymbol{\Sigma}_{j,\text{type}})$ for each identity's feature distribution. The features $\mathbf{y}_{t,\text{imm}}^{(k)}$ utilized by the iMM for slot $k$ are its color $\mathbf{c}_t^{(k)}$ and shape $\mathbf{e}_t^{(k)}$, thus $\mathbf{y}_{t,\text{imm}}^{(k)} = [\mathbf{c}_t^{(k)}, \mathbf{e}_t^{(k)}]^\top$. For each slot $k$ at time $t$ where the activity gate $\mathcal{G}_t^{(k)} \approx 1$, the posterior probability that it belongs to identity $v$ is $q(z_{t,v,\text{imm}}^{(k)} = 1) = \gamma_{t,v}^{(k)}$, satisfying $\sum_{v=1}^V \gamma_{t,v}^{(k)} = 1$. For slots where $\mathcal{G}_t^{(k)} \approx 0$, these responsibilities are effectively null or uniform, contributing negligibly to parameter updates. The posterior over identity-mixing probabilities $\pi_{1:V,\text{imm}}$ is a Dirichlet distribution: $q(\boldsymbol{\pi}_{1:V,\text{type}}) = \text{Dir}(\boldsymbol{\pi}_{1:V,\text{type}} \mid \alpha_{1,\text{type}}, \ldots, \alpha_{V,\text{type}})$, with $\alpha_{v,\text{type}} > 0$. Each identity $v$ is characterized by a mean $\boldsymbol{\mu}_{v,\text{type}}$ and covariance $\boldsymbol{\Sigma}_{v,\text{type}}$. The variational posterior over these parameters is a Normal–Inverse-Wishart (NIW) distribution: $q(\boldsymbol{\mu}_{v,\text{type}}, \boldsymbol{\Sigma}_{v,\text{type}}) = \text{NIW}(\boldsymbol{\mu}_{v,\text{type}}, \boldsymbol{\Sigma}_{v,\text{type}} \mid \boldsymbol{m}_{v,\text{type}}, \kappa_{v,\text{type}}, \mathbf{U}_{v,\text{type}}, n_{v,\text{type}})$.

**E-step Updates for iMM** In the E-step for the iMM, the local assignment probabilities $\gamma_{t,v}^{(k)}$ for each slot $k$ are updated. This update is primarily driven by slots where $\mathcal{G}_t^{(k)} \approx 1$:

$$\gamma_{t,v}^{(k)} \propto \exp\left(\mathbb{E}_q[\log \pi_{v,\text{type}}]\right) \times \exp\left(\mathbb{E}_q[\log \mathcal{N}(\mathbf{y}_{t,\text{imm}}^{(k)}; \boldsymbol{\mu}_{v,\text{type}}, \boldsymbol{\Sigma}_{v,\text{type}})]\right). \tag{57}$$

These responsibilities are normalized such that $\sum_{v=1}^V \gamma_{t,v}^{(k)} = 1$ for each active slot.

**M-step Updates for iMM** During the M-step, the global parameters of the iMM are updated. Sufficient statistics are accumulated, weighted by the gate $\mathcal{G}_t^{(k)}$ to ensure that only actively moving slots contribute significantly to the updates:

$$N_{t,v} = \sum_{k=1}^K \mathcal{G}_t^{(k)} \gamma_{t,v}^{(k)},$$

$$S_{1,t,v} = \sum_{k=1}^K \mathcal{G}_t^{(k)} \gamma_{t,v}^{(k)} \mathbb{E}_q[\mathbf{y}_{t,\text{imm}}^{(k)}], \tag{58}$$

$$S_{2,t,v} = \sum_{k=1}^K \mathcal{G}_t^{(k)} \gamma_{t,v}^{(k)} \mathbb{E}_q[\mathbf{y}_{t,\text{imm}}^{(k)} (\mathbf{y}_{t,\text{imm}}^{(k)})^\top].$$

The Dirichlet parameters $\alpha_{t,v,\text{type}}$ are updated using $N_{t,v}$ and prior parameters $\alpha_{0,v,\text{type}}$ (which, due to the stick-breaking priors are all 1's except for the final count $\alpha_{0,V,\text{type}}$):

$$\alpha_{t,v,\text{type}} = (1 - \rho_t)\, \alpha_{t-1,v,\text{type}} + \rho_t \left(\alpha_{0,v,\text{type}} + N_{t,v}\right). \tag{59}$$

The NIW parameters $(\mathbf{m}_{t,v,\text{type}}, \kappa_{t,v,\text{type}}, \mathbf{U}_{t,v,\text{type}}, n_{t,v,\text{type}})$ are updated by blending their natural parameter representations. The target natural parameters $\widehat{\eta}_{t,v}^{\text{NIW}}$ are derived from the current sufficient statistics $\{N_{t,v}, S_{1,t,v}, S_{2,t,v}\}$ and the NIW prior parameters:

$$(\text{NatParams}_{t,v}^{\text{NIW}}) \leftarrow (1 - \rho_t)(\text{NatParams}_{t-1,v}^{\text{NIW}}) + \rho_t\, \widehat{\eta}_{t,v}^{\text{NIW}}. \tag{60}$$

RECURRENT MIXTURE MODEL (RMM)

The Recurrent Mixture Model (rMM) provides a generative model for a collection of slot-specific features, and importantly, it furnishes the distribution over the switch state $\mathbf{s}_{t,\mathrm{tmm}}^{(k)}$ that governs the Transition Mixture Model (tMM). The rMM itself is a mixture model with $M$ components, and its own slot-specific assignment variable is $\mathbf{s}_{t,\mathrm{rmm}}^{(k)}$. The variational factors include the posterior probability of assignment to rMM component $m$, $q(s_{t,m,\mathrm{rmm}}^{(k)} = 1) = \rho_{t,m}^{(k)}$ (which sums to one over $m$), a Dirichlet posterior $q(\boldsymbol{\pi}_{\mathrm{rmm}}) = \mathrm{Dir}(\boldsymbol{\pi}_{\mathrm{rmm}} \mid \alpha_{1,\mathrm{rmm}}, \ldots, \alpha_{M,\mathrm{rmm}})$ for its mixing weights, NIW posteriors $q(\boldsymbol{\mu}_m, \boldsymbol{\Sigma}_m)$ for continuous features $f_{t-1}^{(k)}$ modeled by each component $m$, and Dirichlet posteriors $q(\boldsymbol{\beta}_{i,m})$ for the parameters of categorical distributions over various discrete features $d_i^{(k)}$ also modeled by component $m$. These discrete features $d_i^{(k)}$ encompass inputs like $\mathbf{z}_{t-1,\mathrm{type}}^{(k)}$, $\mathbf{z}_{t-1,\mathrm{interacting}}^{(k)}$, as well as the tMM switch state $\mathbf{s}_{t,\mathrm{tmm}}^{(k)}$ which the rMM models generatively.

**E-step Updates for rMM**  In the rMM E-step, for each slot $k$ considered active (i.e., $\mathcal{G}_t^{(k)} \approx 1$), the responsibilities $\rho_{t,m}^{(k)}$ for its $M$ components are updated. These updates depend on the likelihood of the slot's input features under each rMM component. The input features include continuous aspects $f_{t-1}^{(k)}$ (a subset of $\mathbf{x}_{t-1}^{(k)}$ such as position and velocity, and interaction features like $\Delta \mathbf{p}_{t-1}^{(k)}$) and a set of discrete features $d_{t-1,\mathrm{inputs}}^{(k)}$ (e.g., type from the previous step $\mathbf{z}_{t-1,\mathrm{type}}^{(k)}$).

$$
\begin{aligned}
\rho_{t,m}^{(k)} \propto{}& \exp\!\big(\mathbb{E}_q[\log \pi_{m,\mathrm{rmm}}]\big) \; \times \; \exp\!\big(\mathbb{E}_q[\log \mathcal{N}(f_{t-1}^{(k)}; \boldsymbol{\mu}_{m,\mathrm{rmm}}, \boldsymbol{\Sigma}_{m,\mathrm{rmm}})]\big) \\
& \times \prod_{i \in \text{input discrete features}} \exp\!\big(\mathbb{E}_q[\log \mathrm{Cat}(d_{t-1,i}^{(k)}; \mathbf{b}_{i,m})]\big).
\end{aligned}
\tag{61}
$$

These responsibilities are normalized to sum to one for each active slot $k$. During rollouts used in planning, the predicted posterior distribution for the tMM switch state $\mathbf{s}_{t,\mathrm{tmm}}^{(k)}$ is then determined as a mixture of the output distributions from the rMM components, weighted by $\rho_{t,m}^{(k)}$:

$$
q\big(s_{t,l,\mathrm{tmm}}^{(k)} = 1\big) = \sum_{m=1}^{M} \rho_{t,m}^{(k)} \, \mathbb{E}_q\big[\boldsymbol{\beta}_{\mathrm{tmm\_switch},m,l}\big],
$$

where $\boldsymbol{\beta}_{\mathrm{tmm\_switch},m,l}$ are the Dirichlet counts associating tMM switch state $l$ to discrete rMM component $m$.

**M-step Updates for rMM**  In the rMM M-step, its global parameters are updated, with contributions from slots weighted by the gate $\mathcal{G}_t^{(k)}$. The expected sufficient statistics are accumulated. For the mixing weights:

$$
N_{t,m} = \sum_{k=1}^{K} \mathcal{G}_t^{(k)} \, \rho_{t,m}^{(k)}.
\tag{62}
$$

For the continuous feature distributions (NIW parameters):

$$
\begin{aligned}
S_{1,t,m}^f &= \sum_{k=1}^{K} \mathcal{G}_t^{(k)} \, \rho_{t,m}^{(k)} \, \mathbb{E}_q[f_{t-1}^{(k)}], \\
S_{2,t,m}^f &= \sum_{k=1}^{K} \mathcal{G}_t^{(k)} \, \rho_{t,m}^{(k)} \, \mathbb{E}_q[f_{t-1}^{(k)} (f_{t-1}^{(k)})^\top].
\end{aligned}
\tag{63}
$$

For each discrete feature $d_i$ modeled by the rMM (this includes input features $d_{t-1,i}^{(k)}$ and the output tMM switch state $s_{t,\mathrm{tmm}}^{(k)}$), and its category $\ell$:

$$
N_{t,m,i,\ell} = \sum_{k=1}^{K} \mathcal{G}_t^{(k)} \, \rho_{t,m}^{(k)} \begin{cases} \mathbb{I}[d_{t-1,i}^{(k)} = \ell] & \text{if } d_i \text{ is an input from } t-1 \\ q(s_{t,\mathrm{tmm}}^{(k)} = \ell) & \text{if } d_i \text{ is } s_{t,\mathrm{tmm}}^{(k)} \end{cases}.
\tag{64}
$$

The parameters are then updated using these statistics. Dirichlet parameters for the categorical distribution over the rMM discrete state $\alpha_{t,m,\text{rmm}}$ (from $N_{t,m}$), NIW parameters for continuous features $(\mathbf{m}_{t,m,\text{rmm}}, \kappa_{t,m,\text{rmm}}, \mathbf{U}_{t,m,\text{rmm}}, n_{t,m,\text{rmm}})$ (from $N_{t,m}, S_{1,t,m}^f, S_{2,t,m}^f$), and Dirichlet parameters $\boldsymbol{\beta}_{t,i,m,\ell}$ for all discrete features (from $N_{t,m,i,\ell}$) are updated via natural gradient blending:

$$\alpha_{t,m,\text{rmm}} = (1 - \rho_t)\,\alpha_{t-1,m,\text{rmm}} + \rho_t\left(\alpha_{0,m,\text{rmm}} + N_{t,m}\right),$$
$$(\text{NatParams}_{t,m}^{\text{NIW}}) \leftarrow (1 - \rho_t)\left(\text{NatParams}_{t-1,m}^{\text{NIW}}\right) + \rho_t\,\widehat{\eta}_{t,m}^{\text{NIW}}, \tag{65}$$
$$\boldsymbol{\beta}_{t,i,m,\ell} = (1 - \rho_t)\,\boldsymbol{\beta}_{t-1,i,m,\ell} + \rho_t\left(\boldsymbol{\beta}_{0,i,m,\ell} + N_{t,m,i,\ell}\right).$$

TRANSITION MIXTURE MODEL (TMM)

The Transition Mixture Model (tMM) describes the dynamics of slot states $\mathbf{x}_t^{(k)}$ using a mixture of $L$ linear transitions. We approximate posteriors over transition assignments and mixing weights with variational factors. Transition responsibilities for slot $k$ using transition $l$ are $q(s_{t,l,\text{tmm}}^{(k)} = 1) = \xi_{t,l}^{(k)}$, satisfying $\sum_{l=1}^L \xi_{t,l}^{(k)} = 1$. The mixing-weight distribution over the $L$ transitions $\boldsymbol{\pi}_{\text{tmm}}$ is $q(\boldsymbol{\pi}_{\text{tmm}}) = \text{Dir}(\boldsymbol{\pi}_{\text{tmm}} \mid \alpha_{1,\text{tmm}}, \ldots, \alpha_{L,\text{tmm}})$.

**E-step Updates**   In the tMM E-step, for each slot $k$, the responsibilities $\xi_{t,l}^{(k)}$ for each transition $l$ are updated based on how well that transition explains the observed change from $\mathbf{x}_{t-1}^{(k)}$ to $\mathbf{x}_t^{(k)}$:

$$\xi_{t,l}^{(k)} = \frac{\exp\left(\mathbb{E}[\log \pi_l]\right)\mathcal{N}\left(\mathbf{x}_t^{(k)};\, D_l\,\mathbf{x}_{t-1}^{(k)} + b_l,\, \mathcal{G}_{t-1}^{(k)}\,2I\right)}{\sum_{u=1}^L \exp\left(\mathbb{E}[\log \pi_u]\right)\mathcal{N}\left(\mathbf{x}_t^{(k)};\, D_u\,\mathbf{x}_{t-1}^{(k)} + b_u,\, \mathcal{G}_{t-1}^{(k)}\,2I\right)}$$

for $l = 1, \ldots, L$. The term $\mathcal{G}_{t-1}^{(k)}$ indicates if the slot was active at $t - 1$, and $2I$ is the process noise covariance, assumed fixed or shared.

**M-step Updates**   The tMM does not use an explicit M-step. Instead, parameters are fixed to their initial values identified during the expansion algorithm. In other words, once we identify a new dynamics mode in the expansion algorithm, these parameters are added as a new component for the tMM and remain fixed.

A.10   BAYESIAN MODEL REDUCTION

Growing new clusters ensures plasticity, but left unchecked it leads to over-parameterisation and over-fitting. To enable generalization, every $\Delta T_{\text{BMR}} = 500$ frames we therefore run *Bayesian model reduction* on the rMM, merging pairs of components whenever doing so *increases* the expected evidence lower bound (ELBO) of the multinomial distributions over the next reward and SLDS switch. The ELBO is computed with respect to generated data from the model through ancestral sampling. Given two candidate components $k_1, k_2$ with posterior-sufficient statistics $(\eta_{k_1}, \eta_{k_2})$, their merged statistics are $\eta_{k_1 \cup k_2} = \eta_{k_1} + \eta_{k_2} - \eta_{k_2}^{\text{prior}}$, ensuring that prior mass is not double-counted.

Candidate pairs are proposed by a fast heuristic that (i) samples up to $n_{\text{pair}} = 2000$ used clusters, (ii) computes their mutual expected log-likelihood under the other's parameters, and (iii) retains the highest-scoring pairs. Each proposal is accepted *iff* the merged ELBO (line 7) is not smaller than the current ELBO (line 2). The procedure is spelled out in Algorithm 2.

A.11   PLANNING WITH ACTIVE INFERENCE

In active inference, policies $\pi = a_{0:H}$ are selected that minimize expected Free Energy (Friston et al., 2017):

$$p(\pi) = \sigma(-G(\pi)), \text{with}$$

$$G(\pi) = \sum_{\tau=0}^H \mathbb{E}_{q(r_\tau, \mathcal{O}_\tau \mid \pi)}\big[-\underbrace{\log \tilde{p}(r_\tau)}_{\text{Utility}} - \underbrace{D_{KL}(q(\boldsymbol{\alpha}_{\text{rmm}} \mid \mathcal{O}_\tau, \pi) \parallel q(\boldsymbol{\alpha}_{\text{rmm}}))}_{\text{Information gain (IG)}}\big], \tag{66}$$

---

**Algorithm 1** Mixture Model Expansion Algorithm

---

| Input | Output | Hyperparameters / Settings |
|---|---|---|
| $\mathbf{Y} \in \mathbb{R}^{N \times d}$: Matrix whose $i^{\text{th}}$ row $(i = 1, \ldots, N)$ is a $d$-dimensional token (e.g., pixel). | $\boldsymbol{\theta}^*_{1:K_{t+1}}$: Updated posterior NIW parameters ($K_{t+1}$ components where $K_{t+1} \geq K_t$) | $\tau$: expansion threshold |
| $\boldsymbol{\theta}^*_{1:K_t} = (\mathbf{m}_{1:K_t}, \kappa_{1:K_t}, \mathbf{U}_{1:K_t}, n_{K_t})$: Initial posterior NIW parameters ($K_t$ components). | | $\mathcal{E}$: maximum expansion steps |

---

1: **Initialise** $K \leftarrow K_t$ and NIW parameters $\boldsymbol{\theta}_k = \left(\mathbf{m}_k, \kappa_k, \mathbf{U}_k, n_k\right)$ for $k = 1 : K$
2: **for** $g = 1$ **to** $\mathcal{E}$ **do**                                        ▷ outer "expand-or-stop" loop
     *// E-step*
3:     **for** $i = 1$ **to** $N$ **do**
4:         $\ell_{ik} \leftarrow \mathbb{E}_{q(\theta_k)}\big[\log p(y_i \mid \theta_k)\big], \; k = 1{:}K$
5:         $r_{ik} \leftarrow \exp(\ell_{ik}) \Big/ \sum_{j=1}^{K} \exp(\ell_{ij})$
6:     $\ell_i^{\max} \leftarrow \max_{k \leq K} \ell_{ik}, \;\; i = 1{:}N$
7:     **if** $\min_i \ell_i^{\max} > \tau$ **then**
8:         **break**                                                               ▷ all tokens well explained
9:     $i^* \leftarrow \arg\min_i \ell_i^{\max}$                                       ▷ worst-explained token
10:    $K \leftarrow K + 1$                                                         ▷ instantiate new component
11:    **Hard-assign** $r_{i^*,k} \leftarrow 0 \, (k < K), r_{i^*,K} \leftarrow 1$

12:    **Initialise component** $K$**:** $\kappa_K \leftarrow 1, \; \nu_K \leftarrow d + 2, \; \mu_K \leftarrow y_{i^*}, \; \Psi_K \leftarrow \Psi_0$
     *// M-step (natural–gradient update)*
13:    **for** $k = 1$ **to** $K$ **do**
14:        $\widehat{\eta}_k \; \leftarrow \; \underbrace{\sum_{i=1}^{N} r_{ik}\, T(y_i)}_{\widehat{T}_k} \; + \; \eta_{k,0}$                ▷ compute target natural parameters
15:        $\eta_k^{(t)} \; \leftarrow \; (1 - \rho_t)\,\eta_k^{(t-1)} \; + \; \rho_t\,\widehat{\eta}_k$                ▷ natural-gradient update with rate $\rho_t$
16:        Unpack $\eta_k^{(t)} \rightarrow (m_k, \kappa_k, U_k, n_k)$                                    ▷ recover NIW hyperparams
17: **return** $\boldsymbol{\theta}^*_{1:K_{t+1}}$
18:

---

with $H$ the planning horizon and $\sigma$ the softmax function. We compute the predicted posterior over future object states and rewards $q(r_\tau, \mathcal{O}_\tau|\pi)$ by rolling out the initial slot posterior at the current timestep $q(\mathcal{O}_t|\pi)$ for $H$ timesteps into the future, using the learned transition model (a combination of the tMM and rMM) to predict future multi-object states and corresponding rewards. The predictive posterior for a given timestep $\tau$ in the planning horizon is therefore given by

$$q(r_\tau, \mathcal{O}_\tau|\pi) = \mathbb{E}_{q(\mathcal{O}_{\tau-1})}[p(r_\tau, \mathcal{O}_\tau \mid \mathcal{O}_{\tau-1}, a_\tau, \boldsymbol{\Theta})] \tag{67}$$

where $a_\tau$ is the action entailed by the policy $\pi$ at timestep $\tau$, and $\boldsymbol{\Theta}$ refers to the parameters of the mixture models relevant to next timestep prediction (the iMM, tMM, and rMM). This operation is applied sequentially to generate an entire trajectory $q(\mathcal{O}_{1:H}|\pi)$. The tilde notation $\tilde{p}(r_\tau)$ indicates the 'biased' reward prior used to compute the expected utility term of $G$. This is a unique feature of active inference, that instead of rewards, planning is accomplished by performing inference over policies given a prior distribution over future observations that is biased towards the agent's preferences (Parr & Friston, 2019). This importantly differs from the unbiased reward model that learns the contingencies between latent states and rewards, which is learned by updates to $\boldsymbol{\Theta}_{\text{rmm}}$. We fix $\tilde{p}(r_\tau)$ *a prior* to be positive for rewarding outcomes and negative for punishments, encoded in terms of its log marginal probabilities $\log \tilde{p}(r_\tau) = [-1, 0, +1]$ for all $\tau$. We compute the expected utility by averaging the reward probabilities expected under the predictive distribution with the log probabilities of the biased prior over the reward: $\log \tilde{p}(r_\tau)^\top \mathbb{E}_{q(\mathcal{O}_\tau|\pi)}[p(r_\tau|\mathcal{O}_\tau, \boldsymbol{\Theta}_{\text{rmm}})]$.

---

**Algorithm 2** Bayesian model reduction for the recurrent mixture model

---

| Input | Output | Hyper-parameters |
|---|---|---|
| $\mathcal{M}$: Posterior rMM | Reduced model $\mathcal{M}'$ | $n_{pair}$, $n_{samples}$ 
 pruning interval $\Delta T_{BMR}$ |

---

1: $\mathcal{D} = \{(\mathbf{c}_i, \mathbf{d}_i)\}_{i=0}^{n_{samples}} \sim \mathcal{M}$      ▷ Draw $n_{samples}$ pairs of continuous and discrete data samples from $\mathcal{M}$
2: $\mathcal{L}^{(0)} \leftarrow \text{ELBO}(\mathcal{M}, \mathcal{D})$      ▷ Compute current ELBO
3: $\mathcal{P} = \{(k_1, k_2)_i\}_{i=0}^{n_{pairs}}$      ▷ Draw up to $n_{pair}$ candidate pairs by heuristic overlap
4: **for** $s = 1$ **to** $|\mathcal{P}|$ **do**
5:      $(k_1, k_2) \leftarrow \mathcal{P}_s$
6:      $\mathcal{M}^{try} \leftarrow \text{MERGE}(\mathcal{M}, k_1, k_2)$
7:      $\mathcal{L}^{try} \leftarrow \text{ELBO}(\mathcal{M}^{try}, \mathcal{D})$
8:      **if** $\mathcal{L}^{try} \geq \mathcal{L}^{(s-1)}$ **then**
9:          $\mathcal{M} \leftarrow \mathcal{M}^{try}$ **and** $\mathcal{L}^{(s)} \leftarrow \mathcal{L}^{try}$      ▷ Set current model to the candidate
10:      **else**
11:          $\mathcal{L}^{(s)} \leftarrow \mathcal{L}^{(s-1)}$
12: **return** $\mathcal{M}$

---

In practice, to generate the predictive posterior $q(\mathcal{O}_\tau|\pi)$, we do not analytically compute the expectation shown in Equation (67), but rather sample from the posterior over the recurrent mixture model switch variable $s_{\tau,m,rmm} \sim q(\mathbf{s}_{\tau,rmm})$, and then use the sampled rMM switch state to sample a transition mixture model switch state $s_{\tau,l,tmm} \sim p(\mathbf{s}_{\tau,tmm}|s_{\tau,m,rmm})$. This is done using the rMM discrete likelihood distribution which describes the probabilities of different tMM switches given rMM switch states. The selected transition switch state is then used to propagate the continuous object states (positions, velocities) through the corresponding transition mixture component analytically (with its learned linear-Gaussian parameters). We then use the next predicted continuous and discrete states (clamping the action input to that entailed by the policy, clamping the identity inputs to the posterior values $q(\mathbf{s}_{t,v,imm})$ at the beginning of the rollout, and omitting the reward input since we are trying to predict it) to generate an expectation over the rMM switch state – a "fictive inference step". This then serves as the rMM posterior over switch states, for the next timestep of the rollout. Finally, to predict the reward at a given timestep, we use the expected reward distribution under the predicted rMM switch state and compute the expected utility by taking the dot-product of the predicted reward distribution with the biased reward prior with logits $= [-1, 0, +1]$.

As the number of possible policies grows exponentially with a larger planning horizon, enumerating all policies at every timestep becomes infeasible. Therefore, we draw inspiration from model predictive control (MPC) solutions such as Cross Entropy Method (CEM) and model predictive path integral (MPPI), which can be cast as approximating an action posterior by moment matching (Okada & Taniguchi, 2019).

In particular, we sample $P$ policies of horizon $H$, and evaluate their expected Free Energy $G$. Instead of sampling actions for each future timestep $\tau$ uniformly, we maintain a horizon-wise categorical proposal $p(a_\tau)$. After every planning step, we keep the top-k samples with minimum $G$, and importance weight with an inverse temperature $\beta = \texttt{temperature}^{-1}$ to get a new probability for each action $p(a_\tau)$ at future timestep $\tau$:

$$p(a_\tau) = \sum_k \frac{\exp(-\beta G(a_\tau^{(k)}))}{\sum_j \exp(-\beta G(a_\tau^{(j)}))} \tag{68}$$

Instead of doing multiple cross-entropy iterations per planning step, we maintain a moving average of $p(a_\tau)$. At every instant, the first action of the current best policy is actually executed by the agent. Our planning loop is hence composed of the following three stages (see Alg. 3).

**Sampling policies.** For each iteration we draw

$$\underbrace{P - R}_{\text{CEM}} + \underbrace{R}_{\text{random}} \quad \text{policies}$$

---

**Algorithm 3** Planning algorithm

---

**Require:** current posterior state $q$, proposal $p(a_\tau)$, best plan $\tilde{\mathbf{a}}$

    *// Sample P candidate policies*

1: $\mathcal{A}_{\text{cem}} \leftarrow \text{Cat}(p(a_\tau))^{\otimes(P-R)}$

2: $\mathcal{A}_{\text{rand}} \leftarrow \text{RANDOM}(R)$

3: $\mathcal{A} \leftarrow [\tilde{\mathbf{a}}, \text{const}_{0:A-1}, \mathcal{A}_{\text{cem}}, \mathcal{A}_{\text{rand}}]$

    *// Evaluate*

4: **for all** $\mathbf{a}^{(p)} \in \mathcal{A}$ **do**

5:     $(\hat{r}_{0:H-1}, \widehat{\text{IG}}_{0:H-1}) \leftarrow \text{Rollout}(q, \mathbf{a}^{(p)})$

6:     $G^{(p)} \leftarrow \sum_\tau \gamma_{\text{discount}}^\tau (\hat{r}_\tau + \lambda_{\text{IG}} \widehat{\text{IG}}_\tau)$

    *// Refit proposal*

7: $\mathcal{K} \leftarrow \text{top-}K$ indices of $-G^{(p)}$

8: **for** $\tau = 0$ **to** $H - 1$ **do**

9:     $\hat{p}(a_\tau) \leftarrow \text{softmax}(\texttt{temperature}^{-1} \text{hist}\{a_\tau^{(p)}\}_{p \in \mathcal{K}})$

10:    $p(a_\tau) \leftarrow \alpha_{\text{smooth}} \hat{p}(a_\tau) + (1 - \alpha_{\text{smooth}}) p(a_\tau)$

11: **return** first action of best plan, updated $p(a_\tau)$, best plan $\tilde{\mathbf{a}}$

---

where the first set is sampled i.i.d. from the proposal $p(a_\tau)$ and the remaining $R$ are "exploratory" sequences generated by a (smoothed) random walk. In addition, the previous best plan is always injected in slot 0 and the $A$ constant action sequences occupy slots $1:A$ to guarantee coverage.

**Evaluating a policy.** Each policy is rolled forward $S$ times through the world-model to obtain per-step predictions of reward $\hat{r}_\tau$ and information gain $\widehat{\text{IG}}_\tau$, averaged over samples. We calculate an expected Free Energy $G$, where in addition, we weigh the information gain term with a scalar $\lambda_{\text{IG}}$ to trade off exploration and exploitation, as well as apply temporal discounting:

$$G = \frac{1}{S} \sum_{s=0}^{S-1} \sum_{\tau=0}^{H-1} \gamma_{\text{discount}}^\tau (\hat{r}_\tau^s + \lambda_{\text{IG}} \widehat{\text{IG}}_\tau^s), \qquad \gamma_{\text{discount}} \in [0, 1), \ \lambda_{\text{IG}} = \texttt{info\_gain} \text{ weight.}$$

**Proposal update.** Let $\mathcal{K}$ be the indices of the top-$K = \lfloor \texttt{topk\_ratio} \cdot P \rfloor$ policies by (negative) expected Free Energy. For every horizon step $\tau$ we form the empirical action histogram of $\{\mathbf{a}_{k,\tau}\}_{k \in \mathcal{K}}$, convert it to probabilities with a tempered softmax (`temperature`) and perform an exponential-moving-average update

$$p(a_\tau)^{\text{new}} = \alpha_{\text{smooth}} p(a_\tau) + (1 - \alpha_{\text{smooth}}) p(a_\tau)^{\text{old}}, \qquad \alpha_{\text{smooth}} = \texttt{alpha}.$$

## B    HYPERPARAMETERS

We list the hyperparameters used for main AXIOM results shown in Figure 3 in Table 3. For the `fixed_distance` ablations, we fixed $r_{\min} = 1.25$ for the games Explode, Bounce, Impact, Hunt, Gold, Fruits, and fixed $r_{\min} = 0.8$ for the games Jump, Drive, Cross, and Aviate.

## C    COMPUTATIONAL RESOURCES, COSTS AND SCALING

AXIOM and baseline models were trained and evaluated on A100 40G GPUs. All models use a single A100 GPU per environment. AXIOM and BBF train a single environment to 10k steps in about 30min, whereas DreamerV3 trains in 2.5h. The corresponding per-step breakdown of average inference and planning times can be seen in Table 2.

### C.1    PLANNING AND INFERENCE TIME

For AXIOM, each time step during training can be broken down into planning and model inference. To quantify the planning time scaling we perform ablations over the number of planning policies (P). Since model inference correlates with the number of mixture model components, we evaluate the

Table 3: Hyperparameters of the AXIOM agent trained to play all 10 games of Gameworld as reported in the main text (see Figure 3).

| Inference Hyperparameters | |
| --- | --- |
| **Parameter** | **Value** |
| $\tau_{\mathrm{smm}}$ (expansion threshold of the sMM) | 5.7 |
| $\tau_{\mathrm{imm}}$ (expansion threshold of the iMM) | $-5 \times 10^2$ |
| $\tau_{\mathrm{rmm}}$ (expansion threshold of the rMM) | $-1 \times 10^1$ |
| $\tau_{\mathrm{tmm}}$ (expansion threshold of the tMM) | $-1 \times 10^{-5}$ |
| $\mathcal{E}$ (maximum number of expansion steps) | 20 |
| $\Delta T_{\mathrm{BMR}}$ (timesteps to BMR) | 500 |
| $\zeta$ (exponent of the damped $\mathbf{z}_{\mathrm{presence}}$ likelihood) | 0.01 |
| $r_{\min}$ | 0.075 |

| Planning Hyperparameters | |
| --- | --- |
| **Parameter** | **Value** |
| $H$ (planning depth) | 32 |
| $P$ (number of rollouts) | 512 |
| $S$ (number of samples per rollout) | 3 |
| $\lambda_{\mathrm{IG}}$ (information gain weight) | 0.1 |
| $\gamma_{\mathrm{discount}}$ (discount factor) | 0.99 |
| top-k ratio | 0.1 |
| random sample ratio | 0.5 |
| temperature ($\frac{1}{\beta}$) | 10.0 |
| $\alpha_{\mathrm{smooth}}$ | 1.0 |

| Generative Model Parameters | |
| --- | --- |
| *Structure* | |
| $K$ (max sMM components) | 32 |
| $V$ (max iMM components) | 32 |
| $L$ (max tMM components) | 500 |
| $M$ (max rMM components) | 5000 |
| $\lambda$ (from $\theta_{\mathrm{moving}}$) | 0.99 |
| $\beta$ (from $\theta_{\mathrm{moving}}$) | 0.01 |
| $\gamma_u$ | $\infty$ |
| $\xi$ | 1.0 |
| $\nu_u$ | 0.05 |
| *Dirichlet/Gamma priors* | |
| $\gamma_{0,\mathrm{R,G,B}}$ | 0.1 |
| $\alpha_{0,\mathrm{smm}}$ | 1 |
| $\alpha_{0,\mathrm{imm}}$ | $1 \times 10^{-4}$ |
| $\alpha_{0,\mathrm{tmm}}$ | 0.1 |
| $\alpha_{0,\mathrm{rmm}}$ | 0.1 |
| *Normal-Inverse–Wishart* $(1{:}V)$ | |
| $\mathbf{m}_{0,1:V,\mathrm{type}}$ | $\mathbf{0}$ |
| $\kappa_{0,1:V,\mathrm{type}}$ | $1 \times 10^{-4}$ |
| $\mathbf{U}_{0,1:V,\mathrm{type}}$ | $\frac{1}{4}I_5$ |
| $n_{0,1:V,\mathrm{type}}$ | 11 |
| *Normal-Inverse–Wishart* $(1{:}M)$ | |
| $\mathbf{m}_{0,1:M,\mathrm{rmm}}$ | $\mathbf{0}$ |
| $\kappa_{0,1:M,\mathrm{rmm}}$ | $1 \times 10^{-4}$ |
| $\mathbf{U}_{0,1:M,\mathrm{rmm}}$ | $625\,I_7$ |
| $n_{0,1:M,\mathrm{rmm}}$ | 15 |
| *Component–wise Dirichlet priors* $(1{:}M)$ | |
| $\boldsymbol{\beta}_{0,1:M,\mathrm{type}}$ | $1 \times 10^{-4}$ |
| $\boldsymbol{\beta}_{0,1:M,\mathrm{interacting}}$ | $1 \times 10^{-4}$ |
| $\boldsymbol{\beta}_{0,1:M,\mathrm{o}}$ | $1 \times 10^{-4}$ |
| $\boldsymbol{\beta}_{0,1:M,\mathrm{tmm}}$ | $1 \times 10^{-4}$ |
| $\boldsymbol{\beta}_{0,1:M,\mathrm{action}}$ | $1 \times 10^{-4}$ |
| $\boldsymbol{\beta}_{0,1:M,\mathrm{reward}}$ | 1.0 |

scaling using the environments Explode (few objects) and Cross (many objects). Figure 5 shows that the planning time scales linearly with the number of policies rolled out (left panel), and how model inference time scales with the number of mixture model components (right panel).

# D  GAMEWORLD 10K ENVIRONMENTS

In this section we provide an informal description of the proposed arcade-style environments, inspired by the Arcade learning environment (Bellemare et al., 2013). To this end, our environments have an observation space of shape $210 \times 160 \times 3$, that corresponds to a RGB pixel arrays game screen. Agents interact via a set of 2 to 5 discrete actions for movement or game-specific interactions. As is standard practice, positive rewards (+1) are awarded for achieving objectives, while negative rewards (-1) are given for failures. Here is a brief description of the games:

**Aviate.**  This environment puts the player in control of a bird, challenging them to navigate through a series of vertical pipes. The bird falls under gravity and can be made to jump by performing a "flap" action. The player's objective is to guide the bird through the narrow horizontal gaps between the pipes without colliding with any part of the pipe structure or the top/bottom edges of the screen. Any collision with a pipe, or going out of screen at the top or bottom results in a negative reward and ends the game.

**Bounce.**  This environment simulates a simplified version of the classic game Pong, where the player controls a paddle to hit a ball against an AI-controlled opponent. The player has three discrete actions: move their paddle up, move it down, or keep it stationary, influencing the ball's vertical trajectory upon contact. The objective is to score points by hitting the ball past the opponent's paddle (reward +1), while preventing the opponent from doing the same (reward -1). The game is episodic, resetting once a point is scored by either side.

**Cross.**  Inspired by the classic Atari game Freeway, this environment tasks the player, represented as a yellow square, with crossing a multi-lane road without being hit by cars. The player has three discrete actions: move up, move down, or stay in place, controlling vertical movement across eight distinct lanes. Cars of varying colors and speeds continuously traverse these lanes horizontally, wrapping around the screen. The objective is to reach the top of the screen for a positive reward; however, colliding with any car resets the player to the bottom of the screen and incurs a negative reward.

**Driver.**  This environment simulates a lane-based driving game where the player controls a car from a top-down perspective, navigating a multi-lane road. The player can choose from three discrete actions: stay in the current position, move left, or move right, allowing for lane changes. The goal is to drive as far as possible, avoiding collisions with opponent cars that appear and move down the lanes at varying speeds. Colliding with another car results in a negative reward and ends the game.

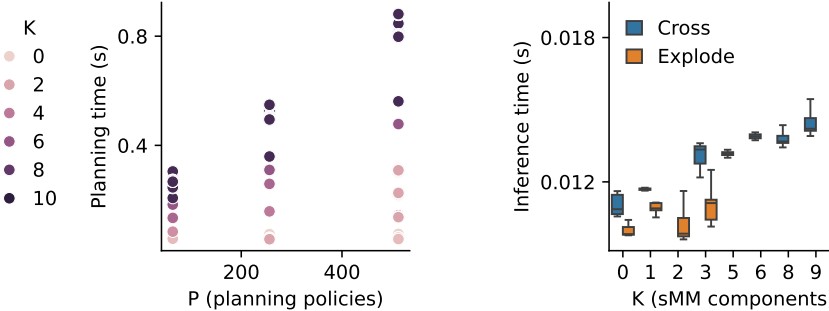

Figure 5: **Computational costs.** Scaling of planning time as a function of the number of policies (left), and model inference time as a function of the number of sMM components (right). All times measured on a single A100 GPU.

**Explode.**  In this game inspired by the arcade classic Kaboom!, the player controls a horizontal bucket at the bottom of the screen, tasked with catching bombs dropped by a moving bomber. The player can choose from three discrete actions: remain stationary, move left, or move right, allowing for precise horizontal positioning to intercept falling projectiles. A bomber continuously traverses the top of the screen, periodically releasing bombs that accelerate as they fall towards the bottom. Successfully catching a bomb in the bucket yields a positive reward, whereas allowing a bomb to fall off-screen results in a negative reward.

**Fruits.**  This game casts the player as a character who must collect falling fruits while dodging dangerous rocks. The player can perform one of three discrete actions: move left, move right, or stay in place, controlling horizontal movement at the bottom of the screen. Fruits of various colors fall from the top, granting a positive reward upon being caught in the player's invisible basket. Conversely, falling rocks, represented as dark grey rectangles, will end the game and incur a negative reward if collected.

**Gold.**  In this game, the player controls a character, represented by a yellow square, from a top-down perspective, moving across a grassy field to collect gold coins and avoid dogs. The player can choose from five discrete actions: stay put, move up, move right, move down, or move left, enabling agile navigation across the screen. Gold coins are static collectibles that grant positive rewards upon contact, while dogs move dynamically across the screen, serving as obstacles that end the game and incur a negative reward if collided with.

**Hunt.**  This game features a character navigating a multi-lane environment, akin to a grid, from a top-down perspective. The player has four discrete actions available: move left, move right, move up, or move down, allowing full two-dimensional movement within the game area. The screen continuously presents a flow of items and obstacles moving horizontally across these lanes. The player's goal is to collect beneficial items to earn positive rewards while deftly maneuvering to avoid contact with detrimental obstacles, which incur negative rewards, encouraging strategic pathfinding.

**Impact.**  This environment simulates the classic arcade game Breakout, where the player controls a horizontal paddle at the bottom of the screen to bounce a ball and destroy a wall of bricks. The player has three discrete actions: move the paddle left, move it right, or keep it stationary. The objective is to eliminate all the bricks by hitting them with the ball, earning a positive reward for each destroyed brick. If the ball goes past the paddle, the player incurs a negative reward and the game resets. The game ends when all bricks are destroyed.

**Jump.**  In this side-scrolling endless runner game, the player controls a character who continuously runs forward, encountering various obstacles. The player has two discrete actions: perform no action or initiate a jump allowing the character to avoid different types of obstacles. Colliding with an obstacle results in a negative reward and immediately resets the game.

# E    ADDITIONAL RESULTS AND ABLATIONS

## E.1    INTERQUARTILE MEAN

Figure 6 depicts the inter-quartile mean (IQM) and interquartile range (Agarwal et al., 2021) of the average collected reward for AXIOM, BBF, DreamerV3, and a random action baseline.

## E.2    BASELINE PERFORMANCE ON 100K

Extending the wall-clock budget to 100 K interaction steps sharpens the contrast between model-based and model-free agents. On Hunt, **DreamerV3** fails to make measurable progress over the entire horizon, remaining near its random-play baseline, whereas BBF and AXIOM attain a similar mean episodic return, on par with the score of AXIOM after 10 K steps (compare to Figure 3 of the main text). In Gold, both baselines do learn to play better than random by 100 K interaction steps, but their asymptotic performance still plateaus below the level AXIOM achieves in both the 100 K-step and the much shorter 10 K-step regime. In table Table 4, we show the final performance for all models at 100 K, with the mean and standard deviation computed from 3 seeds per model and game.

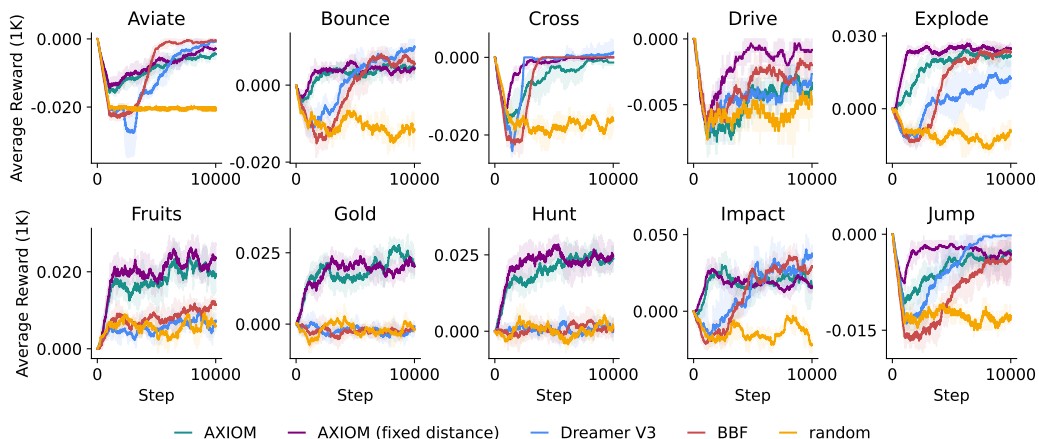

Figure 6: **Online learning performance**. Moving average (1k steps) reward per step during training for AXIOM, BBF, DreamerV3, and a random action baseline on `Gameworld 10k` environments. Interquartile mean (IQM) and interquartile range over 10 parameter seeds are shown per model, per step, and per environment.

### E.3 Ablations

**No information gain.**   When disabling the information gain, we obtain the purple curves in Figure 7. In general, at first glance there appears to be little impact of the information gain on most games. However, this is to be expected, as in Figure 4c we showed that e.g. for Explode, the information gain is only driving performance for the first few hundred steps, after which expected utility takes over. In terms of cumulative rewards, information gain is actually hurting performance on most games where interactions between player and object result in a negative reward. This is because these interaction events will be predicted as information-rich in the beginning, encouraging the agent to experience these multiple times. This is especially apparent in the Cross game, where the no-IG-ablated agent immediately decides not to attempt crossing the road at all after the first few collisions. Figure 8 visualizes the created rMM clusters, which illustrates how no information gain kills exploration in Cross. We hence believe that information gain will play a more important role in hard exploration tasks, which is an interesting direction for future research.

**No Bayesian Model Reduction.**   The orange curves in Figure 7 show the impact of disabling the Bayesian Model Reduction (BMR). BMR clearly has a crucial impact on both Gold and Hunt, which are the games where the player can move freely around the 2D area. In this case, BMR is able to generalize the dynamics and object interactions spatially by merging clusters together. The exception to this is once again Cross, where disabling BMR actually yields the best performing agent. This is again explained by the interplay with information gain. As BMR will merge similar clusters together, moving up without colliding will be assigned to a single, often visited cluster. This will render this cluster less informative from an information gain perspective, and the agent will be more attracted to collide with the different cars first. However, when disabling BMR, reaching each spatial location will get its own cluster, and the agent will be attracted to visit less frequently observed locations, like the top of the screen. This can also be seen qualitatively if we plot the resulting rMM clusters in Figure 8c. This prompts the question of when to best schedule BMR during the course of learning. Clearly, BMR is crucial to generalize observed events to novel situations, but when done too early in learning, it can be detrimental for learning. Further investigating this interplay remains a topic for future work.

Table 4: **Performance at 100k environment steps.** Mean ± standard deviation over seeds.

| Game | AXIOM (mean ± std) | Dreamer V3 (mean ± std) | BBF (mean ± std) |
|------|--------------------|--------------------------|-------------------|
| Hunt | 0.01776 ± 0.00652 | 0.00633 ± 0.00586 | 0.02113 ± 0.00501 |
| Gold | 0.01991 ± 0.00415 | 0.00825 ± 0.00406 | 0.00773 ± 0.00394 |

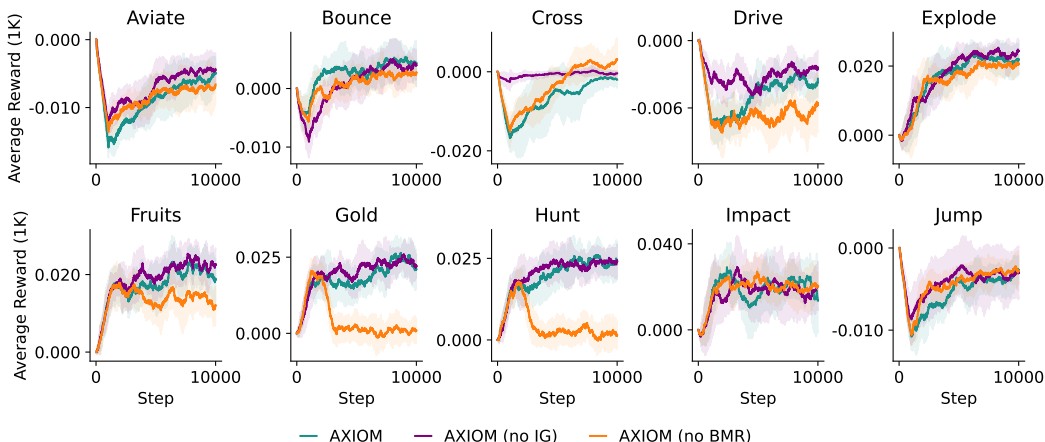

Figure 7: **Performance of AXIOM ablations**. Average reward over the final 1,000 frames across 10 `Gameworld 10K` environments for three AXIOM variants: the full AXIOM model, a version without Bayesian Model Reduction (AXIOM (no BMR)), and a version excluding information gain during planning (AXIOM (no IG)).

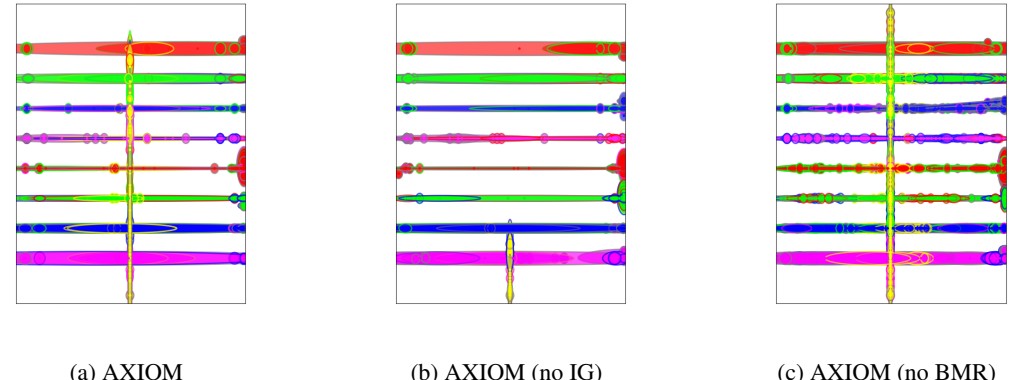

(a) AXIOM                    (b) AXIOM (no IG)                    (c) AXIOM (no BMR)

Figure 8: **Visualizations of the rMM clusters on Cross for information gain and BMR ablations.** Each Gaussian cluster depicts a particular dynamics for a particular object type, colored by the object color, and the edge color of a nearby "interacting" object. (a) AXIOM has various small clusters for the player object (yellow) interacting with the colored cars in the various lanes, and elongated clusters that model the player dynamics of moving up or down. (b) Without information gain, the player collides with the bottom most cars, and then stops exploring because of the negative expected utility. (c) Without BMR, all player positions get small clusters assigned, which in this case helps the player to cross, as visiting these locations is now rendered information gaining.

**Planning rollouts and samples.** As we sample rollouts at each timestep during the planning phase, there is a clear tradeoff between the number of policies and rollout samples to collect in terms of computation time spent (see Figure 5) and the quality of the found plan. We performed a grid search, varying the number of rollouts $[64, 128, 256, 512]$ and number of samples per rollout $[1, 3, 5]$, evaluating 3 seeds each. The results, shown in Figure 9 shows there are no significant performance differences, but more rollouts and drawing more than one sample seem to perform slightly better on average. Therefore for our main evaluations we used 512 policies and 3 samples per policy, but the results in Figure 5 Figure 9 suggest that when compute time is limited, scaling the number of policies down to 128 or 64 is a viable way to increase efficiency without sacrificing performance.

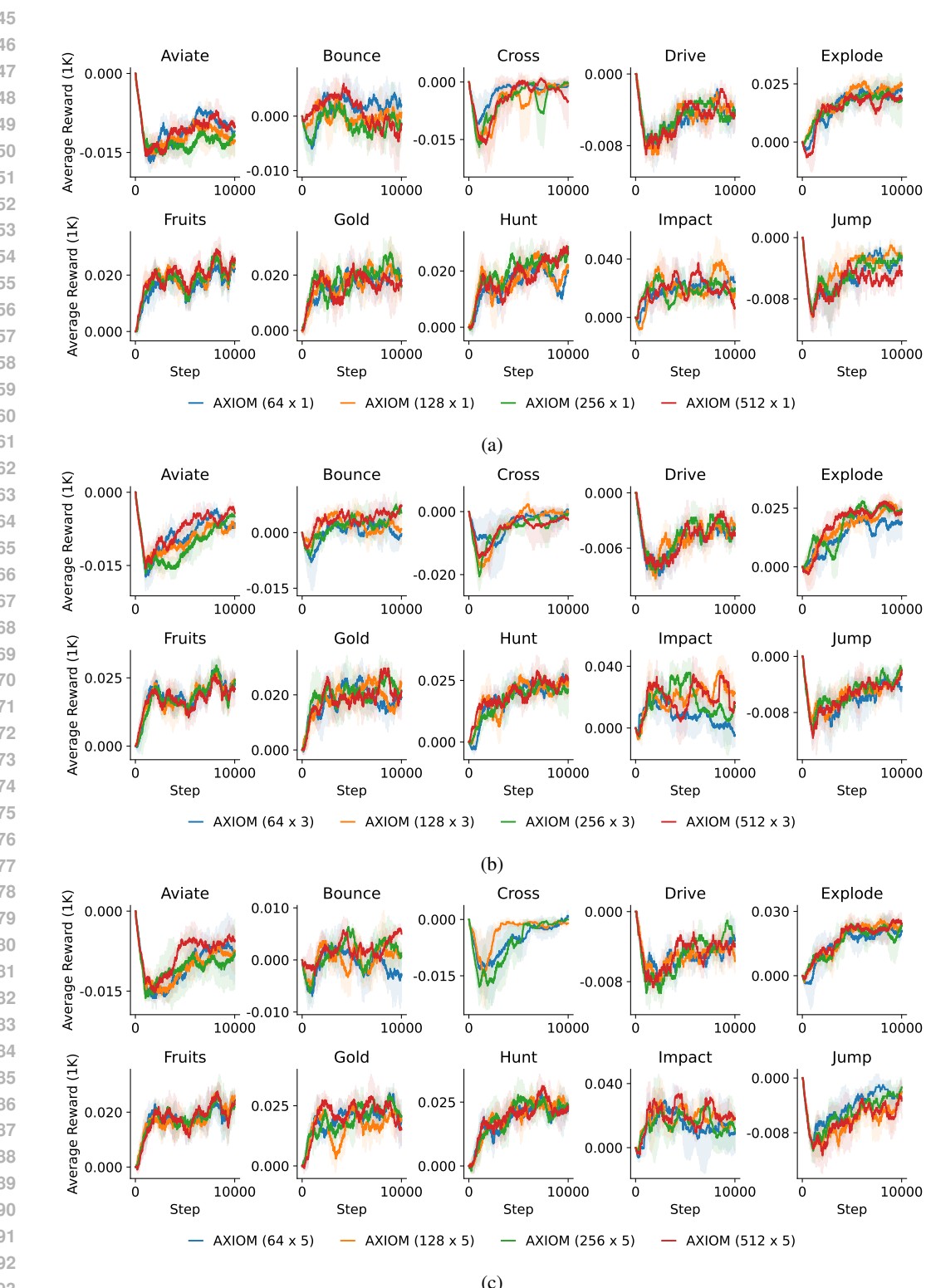

Figure 9: **Ablation on the amount of sampled policies.** The label indicates the number of policies × number of samples for that policy (a) 1 Sample (b) 3 Samples (c) 5 Samples

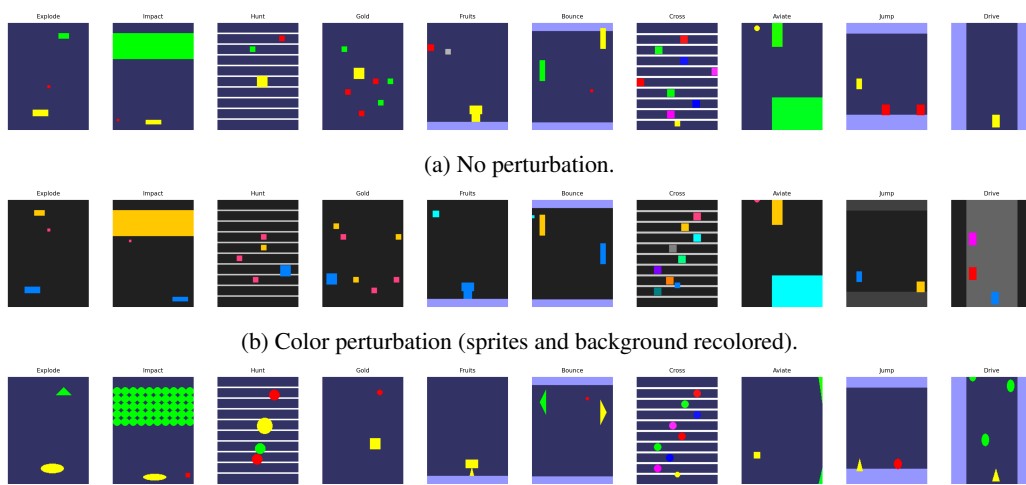

(a) No perturbation.

(b) Color perturbation (sprites and background recolored).

(c) Shape perturbation (primitives replaced with circles and triangles).

Figure 10: **Perturbations.** Sample frames from each of the ten environments under (a) no perturbation, (b) a color perturbation, and (c) a shape perturbation.

### E.4 PERTURBATIONS

**Perturbations.**  One advantage of the `Gameworld 10k` benchmark is its ability to apply homogeneous perturbations across environments, allowing us to quantify how robust different models are to changes in visual features. In our current experiments, we introduce two types of perturbations: a color perturbation, which alters the colors of all sprites and the background (see Figure 10b), and a shape perturbation, which transforms primitives from squares into circles and triangles (see Figure 10c).

To assess model robustness, we apply each perturbation halfway through training (at 5,000 steps) and plot the average reward for Axiom, Dreamer, and BBF across each game in Figure 11. Under the shape perturbation, Axiom demonstrates resilience across games. We attribute this to the identity model (**iMM**), which successfully maps the new shapes onto existing identities despite their altered appearance. Under the color perturbation, however, Axiom's performance often drops - suggesting the identity model initially treats the perturbed sprites as new objects - but then rapidly recovers as it reassigns those new identities to the previously learned dynamics.

Our results also show that BBF and Dreamer are robust to shape changes. For the color perturbation, Dreamer - like Axiom - sometimes experiences a temporary performance decline (for example, in Explode) but then recovers. BBF, by contrast, appears unaffected by either perturbation. We hypothesize that this resilience stems from applying the perturbation early in training - before BBF has converged - so that altering visual features has minimal impact on its learning dynamics.

**Remapped slot identity perturbations**  In this perturbation, shown by the purple line in Figure 11, we performed a special type of perturbation to showcase the 'white-box', interpretable nature of AXIOM's world model.  For this experiment, we performed a standard 'color perturbation' as described above, but after doing so, we encode knowledge about the unreliability of object color into AXIOM's world model.  Specifically, because the latent object features learned by AXIOM are directly interpretable as the colors of the objects in the frame, we can remove the influence of the latent dimensions corresponding to color from the inference step that extracts object identity (namely, the inference step of the iMM), and instead only use shape information to perform object type inference. In practice, what this means is that slots that changed colors don't rapidly get assigned new identities, meaning the same identity-conditioned dynamics (clusters of the rMM) can be used to predict and explain the behavior of the same objects, despite their color having changed.  This explains the absence of an effect of perturbation for some games when using this 'color remapping' trick at the time of perturbation, especially the ones where object identity can easily be inferred from shape, such as Explode. Figure 12 shows the iMM identity slots, with and without the 'remapping

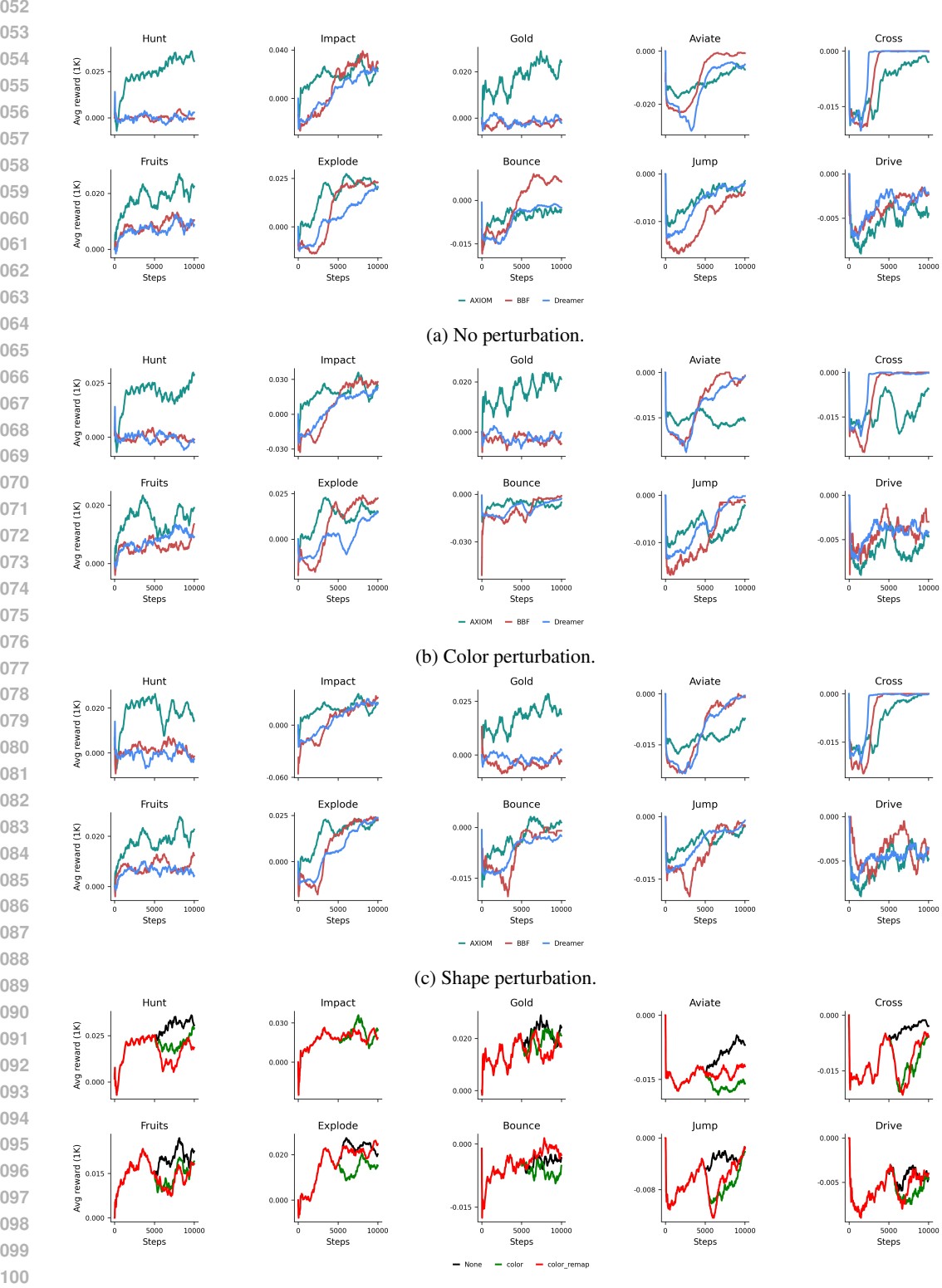

Figure 11: **Impact of perturbations on average reward.** Smoothed 1k-step average rewards for Axiom, BBF, and Dreamer across ten games under (a) no perturbation, (b) color perturbation, (c) shape perturbation, and (d) Axiom's color perturbation with and without remapping.

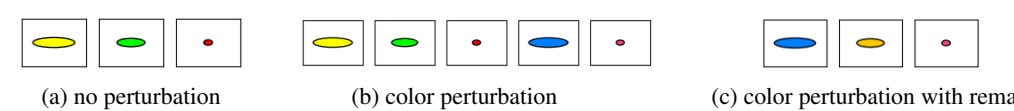

|  (a) no perturbation | (b) color perturbation | (c) color perturbation with remap |

Figure 12: **iMM identity slots on Explode.** (a) On Explode, the iMM constructs a slot for the player, bomber and bomb respectively. (b) When color is perturbed, novel slots are created for the blue (player) and the pink bomb, and the yellow enemy is mapped onto the old player slot. (c) However, with color remapping, the blue player, yellow bomber and pink bomb are correctly remapped to the old player and bomber slots based on their shape.

trick'. Impact on performance for all games is shown in 11d). For games where certain objects have the same shape (e.g., rewards and obstacles in Hunt, or fruits and rocks in Fruits), this remapping trick has no effect because shape information alone is not enough to infer object type and thus condition the dynamics on object types. In such cases, one might use more features to infer the object identity, such as position or dynamics, but extending our model to incorporate these to further improve robustness is left as future work.

## F EXTENDED RELATED WORK

**World Models.** The first breakthroughs in deep reinforcement learning, that leveraged deep Q networks to play Atari games (Mnih et al., 2013), were not model-based, and required training on millions of images to reach human-level performance. To this end, recent works have leveraged model-based reinforcement learning (Stadie et al., 2015; Ghavamzadeh et al., 2015), which learns a world model and can generalize from fewer environment interactions (Ye et al., 2021; Wang et al., 2024). A notable example is the Dreamer class of architectures, which uses a mix of recurrent continuous and discrete states to model the dynamics of the environment (Hafner et al., 2019; 2020; 2025). This class of world models simulates aspects of human cognition, such as intuitive understanding of physics and object tracking (Téglás et al., 2011; Spelke, 1990). AXIOM is also an example of a model-based RL architecture, but notably the world model used for inference, learning, and planning is explicitly probabilistic and interpretable. Unlike previous world models (even probabilistic ones like Karl et al. (2016); Zhang et al. (2019)), AXIOM models the latent world state using explicit object-centric representations that have directly interpretable features like position, velocity, color, and shape. Additionally, all of its probabilistic components are parametric probability distributions from the exponential family; learning these models does require any amortization or use of black-box function approximation.

**Object-centric representations** In recent years, the field of object segmentation has gained momentum thanks to the introduction of models like IODINE (Greff et al., 2019) and Slot Attention (Locatello et al., 2020b), which leverages the strengths and efficiency of self-attention to enforce competition between slot latents in explaining pixels in image data. The form of self-attention used in slot attention is closely-related to the E- and M-steps used to fit Gaussian mixture models (Singh & Buckley, 2023; Kirilenko et al., 2023), which inspired our approach, where AXIOM segments object from images using inference and learning of the Slot Mixture Model. Examples of improvements over this seminal work include Latent Slot Diffusion, which improves upon the original work using diffusion models and SlotSSM (Jiang et al., 2024) which uses object-factorization not only as an inductive bias for image segmentation but also for video prediction. Recent works that have also proposed object-centric, model-based approaches are FOCUS, that confirms how such approaches help towards generalization in the low data regime for robot manipulation (Ferraro et al., 2025), and OC-STORM and SSWM, that use object-centric information to predict environment dynamics and rewards (Zhang et al., 2025; Collu et al., 2024). SPARTAN proposes the use of a large transformer architecture that identifies sparse local causal models that accurately predict future object states (Lei et al., 2024). Unlike OC-STORM, which uses pre-extracted object features using a pre-trained vision foundational model and segmentation masks, AXIOM learns to segment objects online without object-level supervision (although so far we have only tested AXIOM on simple objects like monochromatic polygons). AXIOM also grows and prunes its object-centric state-space online, but like OC-STORM plans using trajectories generated from its world model.

**Bayesian Inference.** Inference, learning, and planning in our model are derived from the active inference framework, that allows us to integrate Bayesian principles with reinforcement learning, balancing reward maximization with information gain by minimizing expected free energy (Parr et al., 2022; Friston, 2010). To learn the structure of the environment, we drawn inspiration from fast structure learning methods (Friston et al., 2024b), that first add mixture components to the model (Rasmussen, 1999) and then prunes them using Bayesian model reduction (Friston et al., 2016; 2024b). Our approach to temporal mixture modeling shares conceptual similarities with recent work on structure-learning Gaussian mixture models that adaptively determine the number of components for perception and transition modeling in reinforcement learning contexts (Champion et al., 2024). An important distinction between AXIOM's model and the original fast structure learning approach (Friston et al., 2024a), is that AXIOM uses more structured priors (in the form of the object-centric factorization of the sMM and the piecewise linear tMM), and uses continuous mixture model likelihoods, rather than purely discrete ones. The transition mixture model we use is a type of truncated infinite switching linear dynamical system (SLDS) (Ghahramani & Hinton, 1996; 2000; Geadah et al., 2024). In particular, we rely on a recent formulation called the recurrent SLDS (Linderman et al., 2016), that introduces dependence of the switch state on the continuous state, to address two key limitations of the standard SLDS: state-independent transitions and context-blind dynamics. Our innovation is in how we handle the recurrent connection of the rSLDS: we do this

using a *generative*, as opposed to *discriminative*, model for the switching states. This allows for more flexible conditioning of the switch state on various information sources (both continuous and discrete), as well as a switch dependence that is quadratic in the continuous features; this overcomes the intrinsic linear separability assumptions made by using a classic softmax likelihood over the switch state, as used in the original rSLDS formulation (Linderman et al., 2015; 2016).

