# OpenReview forum: "AXIOM: Learning to Play Games in Minutes with Expanding Object-Centric Models"
_ICLR.cc/2026/Conference — Submitted to ICLR 2026_

### Official Review · Reviewer_6jLa · 2025-10-29

**Soundness:** 3
**Presentation:** 2
**Contribution:** 3
**Rating:** 6
**Confidence:** 3

**Summary:**

The paper introduces AXIOM, a fully Bayesian, gradient-free reinforcement learning agent that learns from pixels within 10,000 interaction steps. AXIOM models environments as interacting objects through expanding mixture modules for perception, identity, dynamics, and interactions, updating online via variational inference and pruning with Bayesian Model Reduction. Using active inference for action selection, it is evaluated on the proposed Gameworld 10k benchmark of ten object-centric games, where it achieves faster and more data-efficient learning than DreamerV3 and BBF while maintaining interpretable object-level representations.

**Strengths:**

1. Semantically meaningful representations: AXIOM’s vector tokens correspond to interpretable physical quantities such as position, color, velocity, and size, providing a level of semantic transparency rarely achieved in prior slot-based object-centric models.

2. Direct comparison with strong baselines: The paper includes quantitative evaluations against state-of-the-art agents such as BBF and DreamerV3.

3. Introduction of a new benchmark: The authors propose Gameworld 10k, a suite of ten minimalist, object-centric games designed to systematically study sample efficiency, generalization, and object-centric reasoning in low-data regimes.

4. Comprehensive methodological exposition: The paper offers a detailed and well-structured description of AXIOM’s components—including perception, identity, dynamics, and interaction models—supported by ablations and qualitative analyses that clarify each module’s function and contribution.

**Weaknesses:**

1. Limited evaluation scope: The experiments are restricted to the newly introduced Gameworld 10k benchmark, leaving the method’s scalability and effectiveness on more complex or standard environments (e.g., Atari, DMControl, MuJoCo) untested.

2. Potential baseline comparability concerns: Since DreamerV3 and BBF were trained for the new environment, performance differences may partially stem from adaptation or implementation details rather than fundamental algorithmic superiority.

3. Presentation and qualitative analysis limitations: The visualization in Figure 1 could be clearer in illustrating the model’s structure and information flow. And given AXIOM’s strong interpretability claims, the paper would benefit from more extensive qualitative examples demonstrating object tracking, dynamics understanding, and policy reasoning.

**Questions:**

1. Generality of empirical validation: Could the authors extend evaluation to standard benchmarks such as Atari/Procgen/etc to demonstrate AXIOM’s effectiveness and scalability in more complex, visually diverse, and long-horizon environments? I understand the main contribution, but if the method cannot be further scaled, then the value of the claims would be diminished.

2. Necessity of the object-centric assumption: To what extent is the object-centric prior essential for AXIOM’s functioning? Could the same expanding Bayesian framework and active inference mechanism be adapted to environments without explicit object structure or physical dynamics, and if so, what modifications would be required?

---

> ### Author Response · Authors · 2025-11-20
> **Limited evaluation scope; fairness to baselines; qualitative analysis of interpretability**
>
> > Limited evaluation scope: The experiments are restricted to the newly introduced Gameworld 10k benchmark, leaving the method’s scalability and effectiveness on more complex or standard environments (e.g., Atari, DMControl, MuJoCo) untested.
> Potential baseline comparability concerns: Since DreamerV3 and BBF were trained for the new environment, performance differences may partially stem from adaptation or implementation details rather than fundamental algorithmic superiority.
>
> Thanks for your review and comments. We agree that our empirical scope is narrow and focused. As noted above, our primary goal is to demonstrate feasibility and advantages of an expanding Bayesian world model in a controlled, low-data, object-centric regime, rather than to claim broad scalability to all standard benchmarks. We explicitly acknowledged this in the Conclusion and Limitations section of the paper.
>
> To address your concern about the adaptation or implementation details of DreamerV3 and BBF:
> - We expanded the description of the baseline configurations (DreamerV3 and BBF) in the main text, clarifying that we used the official implementations with recommended hyperparameters as a starting point and that we provided DreamerV3 with a large replay ratio and high-capacity model, which should favor it in small data regimes (based on the published findings of the Dreamer V3 paper by Hafner et al. 2025).
> - For BBF we used the paper’s reported model size (width scale 4) and systematically varied the replay ratio to find one that maximizes performance at the 10k interaction budget.
> - All agents share the same observation preprocessing and reward structure.
>
> We added this to the Baselines section for BBF (lines 416-432, new text in bold): “We adapt its preprocessing for the \texttt{Gameworld 10k} suite by replacing frame-skip with max-pooling over two consecutive frames, **and chose the best performing variant of BBF on 10k interactions with Gameworld, which was the published version (width scale 4) with a replay ratio of RR2.**”. And for Dreamer V3: (lines 432-436): “Second, DreamerV3 \citep{hafner2025mastering} is a world-model-based agent with strong performance on games and control tasks with only pixel inputs; we use the published settings but set the train ratio to 1024 at batch size 16 (effective training ratio of 64:1) and **use the 420 million parameter variant. Both these choices maximize Dreamer V3's performance in the small data regime \cite{hafner2025mastering}.**”
>
> > Presentation and qualitative analysis limitations: The visualization in Figure 1 could be clearer in illustrating the model’s structure and information flow. And given AXIOM’s strong interpretability claims, the paper would benefit from more extensive qualitative examples demonstrating object tracking, dynamics understanding, and policy reasoning.
>
> Thank you for this comment and for engaging with the architectural details. We agree that the overall model is complex, but we respectfully believe that the current presentation already provides a fairly clear picture of both the structure and information flow, as well as substantial qualitative evidence for interpretability. Nonetheless, we have attempted to point out the interpretability aspects more explicitly in the main text, by directing attention to more interpretability-relevant plots in the appendix.
>
> - Figure 1 (main text) is intended as a high-level schematic that groups the model into perception, identity, dynamics, and planning modules, and explicitly shows the flow from pixels → object slots → mixture components → action selection. Each arrow corresponds directly to a probabilistic dependency or message-passing step that is spelled out in Section 2 and Appendix A.
>
>
> - Beyond Figure 1, the paper already contains dedicated interpretability figure in the main text: Figure 4(a) visualizes the object-centric state representation, showing that AXIOM’s slots correspond to interpretable physical quantities (positions, colors, etc.), action-conditioned rollouts demonstrate physically-plausible dynamics (predicting a collision between the ball and the wall and between the ball and the paddle in breakout), and the rewarding clusters identified by the rMM correspond exactly to parts of 2-D space where points are accrued or lost.
>
> - In the supplementary material, we provide additional qualitative analyses:
> Figure 8 shows rMM clusters on the Cross environment colored by object-type and the identity of its interaction partner, which shows the learned dynamics of how the player car interacts with the other cars in the different lanes.
> Figure 12 plots the identity components in shape–color space for no perturbation, color perturbation, and color perturbation with remapping, making it visually clear how the iMM reorganizes object identities under distribution shift and how our “remap” intervention acts at the level of explicit, human-understandable features (color and shape).
>
> (continued below)

---

> > ### Author Response · Authors · 2025-11-20
> > **continuation of interpretability demonstration; generality of empirical validation; object-centric assumption**
> >
> > (continued from previous) ...
> > Taken together, we feel these figures give a fairly rich picture of (i) how information flows through AXIOM’s modules, and (ii) how its latent variables and mixture components admit direct semantic interpretation. Nonetheless, to make this more immediately visible to readers, we will explicitly point to Figure 8 in addition to Figure 4(a) to highlight more of the rMM interpretability plots.
> >
> > > Generality of empirical validation: Could the authors extend evaluation to standard benchmarks such as Atari/Procgen/etc to demonstrate AXIOM’s effectiveness and scalability in more complex, visually diverse, and long-horizon environments? I understand the main contribution, but if the method cannot be further scaled, then the value of the claims would be diminished.
> >
> > We agree that demonstrating AXIOM (or its descendants) on larger, more complex suites like Atari or Procgen would significantly strengthen the case for its generality. Due to the points made earlier, we still maintain that this is out of the scope of this paper.
> >
> > > Necessity of the object-centric assumption: To what extent is the object-centric prior essential for AXIOM’s functioning? Could the same expanding Bayesian framework and active inference mechanism be adapted to environments without explicit object structure or physical dynamics, and if so, what modifications would be required?
> >
> > On the necessity of the object-centric prior, we see three levels:
> >
> > 1. Current instantiation (strong object priors).
> >
> > In the current paper, the object-centric prior is essential: the observation pipeline, identity mixture, and transition mixture are all designed around the assumption of discrete objects with approximately locally linear dynamics. Removing that assumption without replacing it with some alternative structure would significantly degrade performance in Gameworld-10k.
> >
> > 2. General expanding Bayesian framework.
> >
> > Mathematically, the expanding mixture machinery and active-inference-style planning do not require objects per se. In fact, the fast structure learning paper (Friston et al. 2024) that inspired our growing mixture model algorithm, is exactly an example of that, where a hierarchical generative model of data is learned one frame at a time, and it doesn’t represent the states in terms of objects, but rather in terms of ‘paths’ or trajectories of activity; in the examples in that paper, the architecture (called RGM: renormalizing generative models) operates directly on pixels and learns a compressed hierarchical representation of high-dimensional data-sequences over time.
> >
> >
> > 3. Intermediate designs (learned objects or parts).
> >
> > A promising middle ground is to retain “entity-centric” structure (slots, parts, or factors) but learn the mapping from pixels to entities via neural encoders, as discussed above. In such a design:
> > - the expanding Bayesian world model stays largely intact;
> > - the object-centric assumption is softened to “entity-like latent structure,” which can be learned rather than explicitly engineered.

---

> > > ### Comment · Reviewer_6jLa · 2025-11-25
> > >
> > > I appreciate the authors' efforts in the rebuttal and the substantial clarifications they have provided. Given that AXIOM achieves performance comparable to Dreamer V3 and BBF while relying on a fundamentally different optimizing paradigm, I believe the work has clear novelty and potential and is worthy of publication. However, without experiments on even a small subset of standard benchmarks, such as Atari Pong or Boxing or basic DMC control tasks, it remains difficult to assess how well the proposed method generalizes beyond the custom Gameworld 10k suite. Additional evidence on such environments would make it much easier to evaluate the practical effectiveness and broader applicability of AXIOM. Therefore I will keep my original score.

---

> ### Author Response · Authors · 2025-11-27
> **Final remarks and justification of Gameworld**
>
> Thank you for your follow-up and for explicitly acknowledging the novelty and potential of AXIOM, even while keeping your original score.
>
> Our choice to focus on Gameworld-10k was driven by two constraints that are hard to satisfy simultaneously in existing benchmarks: (i) tasks that a human can understand and play competently within minutes, and (ii) clean, object-centric 2D physics that align with our explicit priors and a strict 10k-step interaction budget. Many Atari games either require substantially longer horizons and non-obvious mechanics (making “human-in-minutes” and 10k-steps unrealistic), or have visual quirks that complicate an object-centric pixel model (e.g., in Pong the ball often disappears at the boundary or is visually merged with the wall, which the RL literature typically works around via frame stacking). Gameworld-10k therefore abstracts the core dynamics of games like Pong/Breakout and simple control tasks into a small number of interacting sprites with precisely parameterized shapes, colors, and interaction rules, which lets us probe both standard performance and well-defined perturbations, while remaining non-trivial for generic deep RL agents such as DreamerV3 and BBF.
>
> We see our current results as a proof-of-concept that a fully Bayesian, gradient-free world model can compete with strong deep RL baselines in such a low-data, object-centric regime.

---

### Official Review · Reviewer_omuG · 2025-10-31

**Soundness:** 3
**Presentation:** 2
**Contribution:** 3
**Rating:** 6
**Confidence:** 2

**Summary:**

The paper presents AXIOM, an object-centric, Bayesian world model that learns directly from pixels without gradient updates. It decomposes perception and dynamics into modular mixtures for object discovery, type identification, motion prediction, and interaction modeling, with adaptive expansion and pruning through Bayesian model reduction. Using expected free-energy planning, AXIOM achieves fast, sample-efficient learning on the custom Gameworld-10k benchmark, outperforming baselines like DreamerV3 and BBF.

**Strengths:**

**Strengths**

**1. Good Writing:** The paper presents a clear, fully probabilistic framework that decomposes perception, dynamics, and interaction into modular mixture components. AXIOM’s architecture is transparent, where each latent variable has a defined physical or semantic meaning (slot, type, mode, interaction).

**2. High sample-efficiency:** Within only 10k interaction steps, AXIOM achieves competent performance across multiple tasks, often surpassing baselines such as DreamerV3 and BBF. This demonstrates the practical advantage of its object-centric, non-gradient Bayesian updates, which yield faster credit assignment and stable learning in low-data regimes.

**Weaknesses:**

**Weaknesses:**

**1. Generalization to complex tasks:** The Gameworld-10k suite is tailored to object-centric, sparse-interaction dynamics with low visual complexity. While useful for probing the proposed priors, it risks design–method coupling and may inflate relative gains versus deep baselines optimized for high-dimensional, long-horizon settings. Claims of generality are not justified without external baselines. The paper acknowledges not scaling to “complicated control tasks typical of the RL literature,” which limits generalization claims. On the top of that, methods like Dreamer/BBF are known to benefit from larger encoders and replay; compressing inputs (resizing, pooling), altering frame-skip, and a small data budget can handicap them.

**2. Priors encode environment knowledge:** The observation pipeline (pixel tokens with coordinates, nearest-neighbor interaction features, fixed radii) injects strong inductive bias that suits the Gameworld physics.The model’s success appears heavily contingent on its encoded inductive biases like explicit object decomposition, locally linear dynamics, and spatial interaction priors. While these assumptions fit the Gameworld environments almost perfectly, they effectively hard-code much of the task’s physical structure and visual simplicity. Consequently, AXIOM’s reported sample efficiency may reflect prior alignment rather than genuinely better inference or planning. Although the authors briefly acknowledge that AXIOM’s object-centric and locally linear priors restrict its applicability to simple domains, this is treated more as future work than as a central limitation. The paper lacks quantitative evidence on how sensitive performance is to these priors or how gracefully the model degrades when they are violated.

**Questions:**

1. If you remove pixel coordinates and nearest-neighbor features, can a learned CNN and attention recover performance?

---

> ### Author Response · Authors · 2025-11-20
> **Claims of generality, fairness of the baseline parameterization**
>
> > The Gameworld-10k suite is tailored to object-centric, sparse-interaction dynamics with low visual complexity. While useful for probing the proposed priors, it risks design–method coupling and may inflate relative gains versus deep baselines optimized for high-dimensional, long-horizon settings. Claims of generality are not justified without external baselines. The paper acknowledges not scaling to “complicated control tasks typical of the RL literature,” which limits generalization claims. On the top of that, methods like Dreamer/BBF are known to benefit from larger encoders and replay; compressing inputs (resizing, pooling), altering frame-skip, and a small data budget can handicap them.
>
> We largely agree with this assessment, and we appreciate the clear articulation of the concern. Our intention is not to claim that AXIOM, as currently instantiated, is generally superior to deep RL baselines across arbitrary domains. Rather, we see this work as a proof of concept that:
>
> 1. A novel, fully Bayesian, expanding, object-centric world model can achieve strong sample efficiency in a non-trivial pixel environment when its core priors are reasonably aligned with the task; and
> 2. This can be done without neural networks, gradient-based optimization, or replay buffers, using only amortization-free variational updates to mixture models with conjugate priors.
>
> Regarding potential handicapping of DreamerV3/BBF: our goal was to be as fair as possible given the small data budget. The small data budget was intentional, as one of our claims is specifically about sample efficiency (learning to play the games very quickly with few environment interactions).
> To achieve this, we did the following:
>
> - used the official DreamerV3 implementation with the 420M-parameter model, and we gave it a large replay ratio (train ratio 1024 @ batch size 16, effective replay ratio 64), which is at the upper end of what the original paper explores and which favors data efficiency (see Figure 6 of the Nature paper, where scaling size and replay ratio increase sample efficiency).
> -  used the standard published version of BBF (width scale 4) and replay ratio 2, which we found maximized its performance at 10k interaction steps on Gameworld.
> - performed input preprocessing (resizing to 96×96, frame skips) to follow standard practice in model-based Atari/DMControl setups and is not tuned specifically against Dreamer/BBF.
> - gave both baselines the exactly same 10k-step budget and reward signals as AXIOM.
>
> That being said, we did also benchmark a few of the games at 100k interaction steps (Hunt and Gold, with all three methods), and the results are reported in Appendix E.2 and Table 4, which demonstrates that even at 10k interaction steps AXIOM is already achieving the best performance observed across any of the models at 10k steps. BBF does catch up to AXIOM at Hunt by 100k steps, in comparison to its performance at 10k (Figure 3).
>
> (our discussion of your other comments/questions are continued in the next comment)

---

> ### Author Response · Authors · 2025-11-20
> **Dependence on inductive biases; sensitivity of performance to priors; learned CNN + attention alternative**
>
> > The observation pipeline (pixel tokens with coordinates, nearest-neighbor interaction features, fixed radii) injects strong inductive bias that suits the Gameworld physics.The model’s success appears heavily contingent on its encoded inductive biases like explicit object decomposition, locally linear dynamics, and spatial interaction priors. While these assumptions fit the Gameworld environments almost perfectly, they effectively hard-code much of the task’s physical structure and visual simplicity. Consequently, AXIOM’s reported sample efficiency may reflect prior alignment rather than genuinely better inference or planning. Although the authors briefly acknowledge that AXIOM’s object-centric and locally linear priors restrict its applicability to simple domains, this is treated more as future work than as a central limitation. The paper lacks quantitative evidence on how sensitive performance is to these priors or how gracefully the model degrades when they are violated.
>
> We agree that AXIOM’s success in Gameworld-10k is tightly coupled to its inductive biases. In fact, one of the main conceptual messages of the paper is that explicit, human-understandable priors over objects and local interactions can be extremely powerful in low-data regimes, provided they are approximately correct.
> We see two distinct questions here:
>
> 1. Are the priors “hard-coding” the solution?
>
> We do not believe so. While the observation pipeline provides pixel-level features (position and color) for the sMM and local interaction features (nearest-neighbor distances) at the level of the latent states, the agent must still learn:
> - which clusters correspond to semantically meaningful objects (iMM and sMM),
> - which mixtures of local linear dynamics explain the behavior of each object (tMM), and
> - how reward, actions, and interactions are coupled via the recurrent mixture (rMM).
>
> This is evidenced by ablation results where we removed fixed inductive biases like the `fixed_distance` variant of the continuous features provided to the rMM: performance drops, but the agent does not become trivial or fully determined by the priors. The priors structure the hypothesis space; they do not fix a single solution.
>
> 2. Is prior alignment the main driver of sample efficiency?
>
> We think prior alignment and structure learning with fast mixture model expansion/compression is a key driver of sample efficiency in this regime, and we do not intend to downplay that. Deep agents like Dreamer/BBF also rely on strong priors (e.g. convolutional translation invariance, temporal recurrence, replay buffers) that are known to be beneficial in specific domains. Our contribution is to show that _explicitly encoded_ object and interaction priors, combined with structure learning, can yield substantial gains while retaining a fully probabilistic, interpretable model.
>
> > If you remove pixel coordinates and nearest-neighbor features, can a learned CNN and attention recover performance?
>
> This is an excellent question and directly targets the trade-off we would like to explore in future work. We have not run a  variant of AXIOM with a learned front-end replacing the sMM (which operates on pixel-level positions and colors), so we cannot make empirical claims about performance parity. Conceptually, though, we see two plausible directions:
>
> 1. **Hybrid AXIOM with learned object encoder**.
>
> One could attach an object-centric encoder (e.g., Slot Attention) that takes raw pixels and outputs a set of object latents. AXIOM’s expanding mixture stack (iMM/tMM/rMM) would then operate on these latents rather than on cluster-derived tokens. In that setting, coordinates and interaction features would be learned, not hand-engineered. We expect that, with sufficient capacity and training, such an encoder could recover much of the same structure and therefore similar performance, but this comes at the cost of reintroducing neural networks and gradient-based training, and losing interpretability in the latent features output by Slot Attention
>
> 2. **Purely learned perceptual pipelines.**
>
> Alternatively, one could replace the entire observation pipeline with a deep encoder (CNN+attention) and treat its output as “slots” for the Bayesian dynamics stack. This would move the burden of inductive bias from explicit features (coordinates, nearest-neighbor edges) to architectural bias in the encoder.
>
> Exploring these hybrids is a major direction for future work, and we mention this explicitly in the conclusion, when we talk about learning the core priors directly from data. In this paper, we chose to push the opposite extreme: a fully explicit, interpretable pipeline with no learned perception, to isolate the benefits and limitations of hand-crafted core priors. We see our current results as demonstrating what is achievable when those priors are correct, and we view integrating learned perceptual front-ends as a natural next step rather than something we can claim to have solved here.

---

### Official Review · Reviewer_5WXa · 2025-11-01

**Soundness:** 3
**Presentation:** 3
**Contribution:** 3
**Rating:** 4
**Confidence:** 4

**Summary:**

This work proposes to use mixture models (MM) over object-centric representations to learn a dynamics model. The MMs provide a flexible way to expand to additional objects (during testing/training). The authors also propose a benchmark -- Gameworld 10K which has a 10000 steps budget for the agent to achieve the maximum score. Because they use MMs, the number of learnable parameters are significantly smaller when compared to the current state-of-the-art model based RL approaches (like Dreamer). They show fast adaptation to newer object shapes and colors in their experiments.

**Strengths:**

1. A novel way to employ model-based planning (without any neural networks or gradient optimization) that can potentially, in the future be an avenue for fast adaptation.

2. I appreciate that the authors provided anonymized code -- I had a brief look at it.

**Weaknesses:**

1. The core claim of "robustness to environmental perturbations" is not necessarily applicable to AXIOM in particular. As the authors point out, Dreamer and AXIOM are both similarly robust to such perturbations, and BBF instead outperforms both when it comes to robustness. So, I'm not fully convinced of this claim of robustness.

2. There are too many components in the model -- which isn't inherently a bad thing -- however, I wonder if this will scale up to more realistic observations. For instance, can this model generalize and outperform Dreamer on robotic suites like Robosuite / ManiSkill for object manipulation?

3. For a benchmark, to compare different algorithms, I'd like to see IQM [1] metrics since, in certain environments (Figure 3), it is hard to gauge if AXIOM is significantly better than Dreamer/BBF.

----

**References:**

[1] Deep Reinforcement Learning at the Edge of the Statistical Precipice, Rishabh Agarwal et al., NeurIPS 2021

**Questions:**

4. Why is there a need for this benchmark? HackAtari (https://github.com/k4ntz/HackAtari), for instance, enables environment modifications, and since ATARI 100k is an existing and well-known benchmark, creating it for the purpose of showcasing AXIOM work is not a compelling reason.

5. Why is the reward not a function of the next state $\mathcal{O}_{t+1}^k$ and instead predicted from rMM? I'd like the authors to explain their rationale behind this.


6. Which model is being used for DreamerV3? Is it the base model?

---

> ### Author Response · Authors · 2025-11-20
> **Robustness claims; model scaling**
>
> > The core claim of "robustness to environmental perturbations" is not necessarily applicable to AXIOM in particular. As the authors point out, Dreamer and AXIOM are both similarly robust to such perturbations, and BBF instead outperforms both when it comes to robustness. So, I'm not fully convinced of this claim of robustness.
>
> Thank you for raising this. We agree that, as currently phrased, the paper over-emphasizes robustness as a core claim of AXIOM, and that our wording can give the impression that AXIOM is strictly more robust than all baselines.
>
> Our intention with the perturbation experiments was more modest: (i) to show that AXIOM is at least comparable to strong deep RL baselines under structured perturbations in Gameworld-10k (shape and color changes), and (ii) to illustrate how its white-box, object-centric structure allows us to surgically adapt to such perturbations (e.g. the “color remapping” experiment in Appendix E.4), rather than to claim superiority in robustness per se.
>
> Concretely, in the current results:
> - Under shape perturbations, all three agents (AXIOM, DreamerV3, BBF) retain performance or continue the rate of performance increase at their pre-perturbation levels.
>
> - Under color perturbations, AXIOM typically shows a temporary drop followed by recovery as the iMM learns new colors for the affected identities, while DreamerV3 often exhibits a similar pattern and BBF appears less affected, likely because the perturbation occurs early in its training when it has not yet converged (Fig. 10b).
>
> - AXIOM’s distinctive behavior is that, due to its interpretable identity model, we can explicitly modify how identity inference uses color vs. shape (“remapped” condition in Fig. 11d and Fig. 12), effectively telling the agent to ignore color changes and reuse previously learned dynamics, allowing more robust recovery to the performance drops seen on certain games in Fig. 11b. This kind of targeted intervention is not something we can do with DreamerV3 or BBF due to the lack of semantic interpretability of their latent states / internal activations.
>
> To address your concern, we will soften and clarify our robustness claims. In the list of contributions section, we will rephrase the third bullet to avoid the implication that robustness is a unique core advantage of AXIOM, that the alternative methods lack. For example, we have changed ““with our online learning scheme showing robustness to environmental perturbations” to “and show that our online learning scheme is **at least as robust as strong deep RL baselines** under structured visual perturbations in Gameworld 10k, while additionally allowing interpretable, white-box interventions on the world model.”
>
> >There are too many components in the model -- which isn't inherently a bad thing -- however, I wonder if this will scale up to more realistic observations. For instance, can this model generalize and outperform Dreamer on robotic suites like Robosuite / ManiSkill for object manipulation?
>
> We appreciate this question. We want to clarify two things:
> 1. What we are and are not claiming in the current paper.
> 2. Why we believe the expanding-mixture architecture can scale conceptually, and how we envision doing so.
>
> (i) Scope of our claims
>
> We agree that, as its currently stands, AXIOM is tailored to relatively simple, object-centric pixel environments like Gameworld-10k, where the underlying physics is sparse and the visual structure is deliberately minimal. We do not claim that the exact observation pipeline and prior choices we use here would immediately outperform DreamerV3 on much more complex robotic suites such as Robosuite or ManiSkill.
> In fact, we acknowledge this limitation in the conclusion: “Our work is limited by the fact that the core priors are themselves engineered rather than discovered autonomously… Future work will focus on developing methods to automatically infer such core priors from data, which should allow our approach to be applied to more complex domains like Atari or Minecraft…”
>
> (ii) "Too many components” and computational scaling
>
> AXIOM includes four mixture modules, but the effective model size is modest and self-regularizing. Across all Gameworld environments, AXIOM uses between ~0.3M and 1.6M parameters, compared to 6.47M for BBF and 420M for DreamerV3 (Table 2). Per-step model update time is ~18 ms for AXIOM vs. 135 ms (BBF) and 221 ms (DreamerV3), with planning cost scaling linearly in the number of rollouts (Table 2, Fig. 5).
>
> The rMM starts with many components but is aggressively pruned via Bayesian model reduction; as shown in Fig. 4b, the number of active components drops sharply during training, converging to a compact set of “useful” modes.
>
> So while the design exposes large caps (e.g. up to 500 tMM modes, 5000 rMM components), the learned models remain small and environment-adapted, and the inference cost scales linearly in the number of active components (Appendix C.1).
> (comment continued below)

---

> > ### Author Response · Authors · 2025-11-20
> > **Continuation on model scaling; reporting of IQM rather than mean; why this benchmark?**
> >
> > (iii) How we envision scaling to more realistic observations
> >
> > The core mechanisms of AXIOM - expanding mixture models for perception and dynamics, recurrent mixture-based switching, and Bayesian model reduction - are not tied to hand-engineered pixel tokens. What is domain-specific is the current observation pipeline (pixel+coordinates as inputs to the sMM and manually designed interaction features like the distance-to-nearest-object feature).
> > In a robotics setting, we envision two natural extensions:
> > 1. Plug AXIOM’s dynamics stack on top of learned object-centric encoders.
> > Instead of feeding raw pixels into the sMM, one could use Slot Attention, FOCUS, or other object-centric encoders to produce a small set of object latents (including robot state), and apply the iMM/tMM/rMM stack to those latents. This would decouple “perception from raw images” from “Bayesian structure learning over object dynamics,” and should scale more naturally to complex scenes.
> >
> >
> > 2. Hybrid observation models.
> > For manipulation tasks, it is natural to incorporate low-dimensional proprioceptive and action spaces alongside visual tokens. The rMM already mixes continuous and discrete features and is structurally compatible with such hybrid inputs. Extending the observation model to include joint angles/velocities and contact information is conceptually straightforward within our framework.
> >
> > We see the present work as establishing that this expanding Bayesian machinery can support fast, gradient-free control in a non-trivial pixel domain. Demonstrating competitive performance on large-scale robotic suites (Robosuite/ManiSkill) is an exciting but substantial engineering effort, and we agree that it lies beyond the scope of this submission.
> >
> > > For a benchmark, to compare different algorithms, I'd like to see IQM [1] metrics since, in certain environments (Figure 3), it is hard to gauge if AXIOM is significantly better than Dreamer/BBF.
> >
> > We have now included an additional variant of Figure 3 (Figure 6 in Appendix E.1) that uses IQM + inter-quartile range as defined in the paper you cited, instead of the mean + std. The results are qualitatively identical, so we kept Figure 3 in the main text to use the mean + std to keep consistency  with the statistical quantification we used for other parts of the results reporting (like the cumulative reward table and the computation/inference time statistics). We also cite the Agarwal et al. 2021 paper you pointed us to in the section of Appendix E.1  when discussing Figure 6.
> >
> > > Why is there a need for this benchmark? HackAtari (https://github.com/k4ntz/HackAtari), for instance, enables environment modifications, and since ATARI 100k is an existing and well-known benchmark, creating it for the purpose of showcasing AXIOM work is not a compelling reason.
> >
> > Thank you for this comment. We agree that existing suites like Atari-100k and tools like HackAtari are valuable for evaluating model-based RL agents under environment modifications. Our motivation for introducing Gameworld-10k is more specific and somewhat orthogonal:
> > 1. Human-solvable in minutes, agent budget of 10k steps.
> >  We deliberately chose games that a human can typically understand and play competently within a few minutes (roughly a few hundred interactions), and then enforced a very small interaction budget (10k steps). This combination creates an extremely data-scarce regime where we can meaningfully ask: “Can a structured Bayesian world model approach human-like sample efficiency?” Many Atari-100k tasks either require substantially more interaction to become intuitive or involve long-horizon strategies that humans usually learn over much longer exposure.
> >
> > 2. Controlled spectrum of difficulty, from Pong to Montezuma-like exploration.
> > Gameworld-10k includes games closely related to classic arcade titles such as Pong-like and Breakout-like tasks (easier, with dense or shaped rewards), but also several less obvious games where the correct objective is not visually obvious and must be inferred from sparse feedback, culminating in a Montezuma-style hard exploration environment. This spectrum is designed so that:
> > - the “easy” games are simple enough that both humans and AXIOM can discover good strategies quickly, and
> > - the “hard” games expose exploration and credit assignment challenges under the same 10k-step constraint.
> >
> > 3. While Atari also spans a range of difficulties, it is less controlled in the sense that difficulty is entangled with many other factors (visual complexity, stochasticity, long horizons, etc.), and the agent interaction budget is typically larger (e.g. 100k).
> >
> > (more reasons continued in next comment)

---

> > > ### Author Response · Authors · 2025-11-20
> > > **"Why this benchmark" continued; reward modelling clarification; which model used for DreamerV3**
> > >
> > > (continued from previous question about "why not HackAtari")
> > >
> > > 4. Object-centric, factorized environment design.
> > > Gameworld games are built from a small number of explicitly parameterized objects (positions, sizes, colors, velocities, interaction rules). This gives us precise control over the generative factors, allowing us to design:
> > > - tasks that directly probe object-centric reasoning,
> > > - structured perturbations of object appearances and dynamics, and
> > > - ablations where particular priors (e.g. local interactions) are systematically matched or violated.
> > >
> > > 5. HackAtari does enable modifications, but they mostly operate at the level of pixel manipulations and reward shaping on top of fixed, complex Atari engines. It is harder there to systematically vary a single physical prior (e.g. interaction range, object identity mapping) while keeping everything else fixed.
> > >
> > > 6. Focusing on a regime where AXIOM’s design is scientifically interesting.
> > > Our goal in this paper is not to propose a new “universal” benchmark for the community, nor to argue that Gameworld-10k should replace Atari-100k. Rather, we use Gameworld-10k as a targeted testbed for a particular architectural question: can a fully probabilistic, object-centric, gradient-free world model compete with large neural agents in a very low-data setting, when its priors are approximately correct? We see this as complementary to Atari-style evaluations, not a replacement.
> > >
> > > > Why is the reward not a function of the next state? And instead predicted from rMM? I'd like the authors to explain their rationale behind this.
> > >
> > > The reward is a random variable of the generative model (the likelihood distribution of the recurrent mixture model – see Equation (7)). The probability of the reward depends on the switch state of the recurrent mixture model. Inference over this switch state itself depends on multiple inputs: the discrete and continuous latent object states, the reward, and the action. It’s this inference of the recurrent mixture model’s switch state, which couples the reward to the state, in the generative model. Of course in the generative process (the actual game), the reward is a function of the game state.
> > >
> > > The reward is predicted from the rMM during model-based rollouts (planning). The reason we couple the reward to the rMM and not directly to one of the continuous or discrete slot states, is to enable high-dimensional correlations to be learned between the reward and other variables (distance to other objects, and other features), all using a mixture model which can be updated using variational Bayes. If we had used some generic neural network (like a value network) that learns a mapping from the state to the reward, we would have to use gradient-based learning to learn this mapping, and the weights of that neural network would be fixed / not subject to continual learning. With the rMM we have introduced a much more flexible way to learn dependencies between the reward and many other variables, all while eschewing the use of backpropagation and making the model parameters amenable to uncertainty quantification (since we have variational posteriors over the parameters) and continual learning.
> > >
> > > > Which model is being used for DreamerV3? Is it the base model?
> > >
> > > We used the 420 million parameter model, based on the results in the paper about scaling and performance/sample efficiency (generally finding that larger models lead to better performance and sample efficiency (Figure 6c of the Dreamer V3 paper). We set the train ratio to 1024 at batch size 16 (effective replay ratio of 64, the highest ratio discussed in the paper – see Figure 6d), and scaled the frames to 96 x 96 (in line with the published implementation). This is detailed in Table 2 and in the Baselines section of the Methods section.

---

### Official Review · Reviewer_ccyd · 2025-11-01

**Soundness:** 2
**Presentation:** 2
**Contribution:** 2
**Rating:** 2
**Confidence:** 4

**Summary:**

The paper develops a model consisting of multiple mixture components for doing reinforcement learning. It develops a formalism for training that large mixture model.

**Strengths:**

The paper defines a large model with many different mixture model components for learning a variational posterior on observing trajectories within an RL problems. It applies variational inference and a mixture component split merge structure to develop an inferential procedure that can be used in planning. Instead of learning to optimize reward from the outset, it uses an approximate Bayesian approach for concurrent world modelling and refinement of parameter distribution training. Arguably this concurrency provides a big win in terms of data-efficiency.

**Weaknesses:**

(Please sort out the references - there is a significant lack of care in the references, capitalisation is all over the places - Gauss is a proper noun etc. This does not reflect well on the work). The paper has an overabundance of gratuitous references to the work of Karl Friston. Bayesian agent architectures have been around for decades prior to Parr et Al. Beliefs are always updated incrementally as new
evidence emerges, it doesn't need another Friston reference to establish that. Nor is mixture adding and merging a Friston idea (SMEM did this in 1998, followed up by many others). And again throughout the paper. One might be forgiven for believing this paper was an exercise in H-index hacking.

Altogether, I do not see any indication as to how the authors relate this to other Bayesian reinforcement learning implementations, or to general model-based reinforcement learning, and world modelling. Despite the copious references to Friston, there seems to be incredibly little reference to most other Bayesian RL work at all? see e.g. the review of Kang, Tobbler and Dayan, and many more. There is decades of work in this area, and this work should really be set in the context of the existing RL literature. The model itself is multiple factorised mixture models, with a heuristic updating scheme for the structure. The exact relationship to Bayesian agents is a little lost in the actual formalism. There is some confusion as to be Bayesian (over model parameters) and having probabilistic state updates (which this shares with most POMDP related formalisms). The planning aspect is a little cryptic, and lacks specifics.

The paper would be improved if it explicitly and precisely defined its problem setting, assumptions, and way the results are assessed. There is no problem statement at all.

I don't believe the priors over parameters has been defined here.

Does this relate strongly to the "RL as inference" angle of Mark Toussaint and later, Surgey Levine?

The recurrent mixture model captures most of the structure of the model. It deals with the reward structure. i.e. the strong dependence of the instantaneous reward on the state, which has to be learnt. Yet it is quite worrying as that seems to be handled in 7 as an _independent_ categorical distribution between state and reward. This can't make sense, yet that is what it says. Reward modelling is critical but is not dealt with explicitly and is hidden away here. That seems quite worrying, and I wonder here is there is actually any real reward modelling happening.

The posterior factorisation assumptions seem to be worrying, in that it decouples things that are strongly coupled. The internal posteriors over the mixture parameters could be discussed further.

However the main weakness of the paper is that it dismisses evaluating the approach on any of the standard settings used in this context, instead defining their own unestablished setting just for them, so we can't tell how good it is as a test. This leads to irrelevant benchmarking, as it is close to impossible to know the performance of the proposed method. At the very least the authors should give results on existing benchmarks, critique those and propose further benchmarks.

**Questions:**

Please could you explain your reward modelling. This seems really unclear at the moment.

---

> ### Author Response · Authors · 2025-11-20
> **Initial comments, reference correction, relationship to Bayesian RL, mixture model growth algorithms**
>
> We thank the reviewer for their detailed and critical assessment of our work. We would like to emphasize upfront that AXIOM is, to our knowledge, the first fully Bayesian, object-centric world model that (i) learns directly from raw pixels in high-dimensional control tasks, (ii) uses only explicit exponential-family distributions with closed-form variational updates (no neural networks, no replay buffers, no gradient-based optimization anywhere in the world model), and (iii) performs model-based planning over this structured generative model to solve a suite of visually rich games within a strict 10k-interaction budget. This combination goes beyond both classical Bayesian model-based RL and active inference models (which typically operate in low-dimensional, hand-specified state spaces) and modern deep world models (which rely on amortized inference and black-box neural architectures). Our primary contribution is to demonstrate that an expanding mixture model architecture with object-centric priors can scale to pixel-based control, remain fully interpretable and probabilistic, and achieve competitive sample efficiency (on an albeit non-standard benchmark) with state-of-the-art deep RL baselines. In the responses below, we clarify how AXIOM relates to prior Bayesian RL work, specify our unique contributions and priors more explicitly, and explain in detail our reward modelling and planning scheme.
>
> > (Please sort out the references - there is a significant lack of care in the references, capitalisation is all over the places - Gauss is a proper noun etc. This does not reflect well on the work). The paper has an overabundance of gratuitous references to the work of Karl Friston.
>
> Thank you for your comments and feedback on our work. We have now taken care to make the citations more consistent and with proper nouns capitalized consistently throughout.
>
> >  Bayesian agent architectures have been around for decades prior to Parr et Al. Beliefs are always updated incrementally as new evidence emerges, it doesn't need another Friston reference to establish that. Nor is mixture adding and merging a Friston idea (SMEM did this in 1998, followed up by many others). And again throughout the paper. One might be forgiven for believing this paper was an exercise in H-index hacking. Despite the copious references to Friston, there seems to be incredibly little reference to most other Bayesian RL work at all? see e.g. the review of Kang, Tobbler and Dayan, and many more. There is decades of work in this area, and this work should really be set in the context of the existing RL literature.
>
> The reason we cite active inference in the introduction section (which, based on your mention of Parr et al., makes us think this is what you’re specifically referring to) as opposed to Bayesian model-based RL more generally, is because in that section we’re specifically situating our work in contrast to earlier active inference models of modern challenges in model-based RL, which have not been able to scale and rely on small-scale, low-dimensional tasks with hand-designed priors and generative models. However, we take the point that we should at least cite more generally Bayesian RL and model-based RL in the introduction, so we have now also added a citation to the paper you mentioned by Kang, Tobler and Dayan (2024) and the survey paper Ghavamzadeh et al. 2015, which covers Bayesian RL as it’s used in the machine learning /RL communities.
>
> > Nor is mixture adding and merging a Friston idea (SMEM did this in 1998, followed up by many others)
>
> We agree, and this is also why we cited Ishwaran and James (2001) several times throughout the Methods section and Rasmussen (1999) (“The infinite Gaussian mixture model”) in the Related Works Section (Appendix F) to pay homage to prior work on growing / merging components in GMMs. That being said, we accept that the prior art on adding/merging components should be cited earlier, so we added a citation to SMEM in the introduction alongside the Bayesian model reduction citation and the fast structure learning citation, to address your concern, in addition to moving and shortening the Related Work section to the main text so that our references to other frameworks are more front-and-center. It’s worth also noting that the reason we specifically cite the ‘fast structure learning’ work by Friston (2024) is because, unlike other mixture-model component growth algorithms (which often require operating on full-batches of data like SMEM), the approach we use here is fully streaming, and operates one datapoint at a time. There is some other work in this area which we previously neglected (like Song and Wang 2005), which we now cite along with the SMEM paper (Ueda et al. 1998), the fast structure learning paper Friston (2024) and seminal work on infinite GMMs by Rasmussen (1999) when discussing growing mixture models.
>
> (Due to character limits we continue our responses in the next comment)

---

> > ### Comment · Reviewer_ccyd · 2025-11-20
> > **Response**
> >
> > (ii) uses only explicit exponential-family distributions with closed-form variational updates (no neural networks, no replay buffers, no gradient-based optimization anywhere in the world model),
> >
> > The fact that you choose not to use tools that are available to you is not itself a benefit, and does not mean you can restrict comparison to ignore work that does have these features. The problem definition and evaluation are (or should be) agnostic of any approach that is chosen to solve the problem.
> >
> > "we’re specifically situating our work in contrast to earlier active inference models of modern challenges in model-based RL, which have not been able to scale and rely on small-scale, low-dimensional tasks with hand-designed priors and generative models."
> >
> > active-inference is one description of a more general body of work, and does not really differ from the majority of other variational inference work for sequential learning. Hence, again restricting your supposed domain to a single literature stream is not helpful to the reader.
> >
> > Furthermore I think the claim that model-based RL, has not been able to scale is not really true: there are a plethora of pixel-level world models out there now, and people doing online learning within them, that in my mind scales well beyond what we see in this paper. This feels assertive to me.
> >
> > I am absolutely not adverse to proper inferential models that respect the underlying POMDP structure assumed of the environment. However I am not so happy with an arbitrary constraining of the tool that one can use, and more importantly a tailoring of the exact test environment to be tailored to your particular method. This strikes me as a case of Dataset Selection (https://rakaposhi.eas.asu.edu/f02-cse494-mailarchive/pdf00004.pdf)

---

> > > ### Author Response · Authors · 2025-11-26
> > > **Initial comments / clarification of what claims were made**
> > >
> > > Thank you for continuing to engage with us on this work, your comments are very helpful and helping us make the paper a stronger contribution. We leave some responses to your latest set of comments below.
> > >
> > > > The fact that you choose not to use tools that are available to you is not itself a benefit.
> > >
> > > We contend that we showed explicitly in the paper how and why it is a benefit, in terms of reduced computational expense (Table 2) and sample efficiency (Figure 3). One of the core contributions of the paper is about the benefits of using explicit core priors without gradient-based optimization, not *just* the fact that we don't gradient-based optimization, replay-buffers, amortization, etc.
> > >
> > > > and does not mean you can restrict comparison to ignore work that does have these features.
> > >
> > > We do not restrict comparison to ignore work that does have these features. We chose to compare AXIOM to two architectures that do have these features: DreamerV3 and BBF. In what way, in your opinion, are we ignoring work that has these feature, and how could the paper be changed in a way that doesn't ignore this work?
> > >
> > > >  The problem definition and evaluation are (or should be) agnostic of any approach that is chosen to solve the problem.
> > >
> > > It is worth re-emphasizing that the goal of this paper was not to "achieve higher cumulative reward than any other architecture, whatever it takes".  If we were in that game, we would probably use amortization / model-scaling to achieve this. The problem statement here is more a scientific one. We are asking the question – what benefits are there, relative to amortized, gradient based models when using a fully structured, explicit generative world model learned with streaming variational Bayesian inference everywhere? And in our opinion, we have both (partially) answered this question through the method and results, and identified benefits in terms of sample efficiency, parameter efficiency, computational efficiency, and interpretability. These benefits are the explicit focus of the Results section and the figures.
> > >
> > > > active-inference is one description of a more general body of work, and does not really differ from the majority of other variational inference work for sequential learning. Hence, again restricting your supposed domain to a single literature stream is not helpful to the reader.
> > >
> > > We largely agree with the statement "active-inference is one description of a more general body of work, and does not really differ from the majority of other variational inference work for sequential learning". The main distinctive feature of active inference, compared to Bayesian model-based RL, lies in how it defines a unified variational objective used for planning, which leads to both an expected-utility-like term and an info-gain term, and the way rewards factor into this object, i.e., by using a 'biased prior’ over future outcomes in lieu of an explicit reward function. In our view, we did not restrict our domain to a single literature stream -- that is why we cite plenty of  other literature in the Bayesian inference / sequential Bayesian inference literature, and the model-based RL literature. We hope that our rebuttal, as well as our inclusion of Related Works in the main text, now help convince you that we have done due diligence / proper citation of other work.
> > >
> > > > Furthermore I think the claim that model-based RL, has not been able to scale is not really true: there are a plethora of pixel-level world models out there now, and people doing online learning within them, that in my mind scales well beyond what we see in this paper. This feels assertive to me.
> > >
> > > Respectfully: we never made the claim “model-based RL has not been able to scale"; indeed, it's precisely _because_ model-based RL has been able to scale to high-dimensional, pixel-based environments (e.g., DreamerV3, MuZero, EfficientZero, etc.), that we attempted to pursue this research in the first place. The claim we are making, is that it has _only ever been able to scale_ with the use of amortized inference. And as we said, this amortized inference comes with drawbacks (computational expense of gradient-based optimization via maintaining a replay buffer, lack of uncertainty quantification over the parameters of the amortization, sample inefficiency, parameter inefficiency, etc.).

---

> > > > ### Author Response · Authors · 2025-11-26
> > > > **Benchmark limitation, Slot Mixture Model Expressiveness, Amortized vs Explicit Inference**
> > > >
> > > > > I am absolutely not adverse to proper inferential models that respect the underlying POMDP structure assumed of the environment. However I am not so happy with an arbitrary constraining of the tool that one can use, and more importantly a tailoring of the exact test environment to be tailored to your particular method. This strikes me as a case of Dataset Selection (https://rakaposhi.eas.asu.edu/f02-cse494-mailarchive/pdf00004.pdf)
> > > >
> > > > We continue to acknowledge this as a limitation of the work (namely, the choice of the benchmark). However, we do still believe that the novelty of the architecture and the uniqueness of this approach relative to what's happening more largely in the ML and RL fields, makes it an important contribution, important enough in our mind to warrant publication at ICLR. That being said, this is just our opinion in the end so if you think that the benchmark/dataset limitation is truly condemning of the work as a whole, we respect that and have nothing really further to say on this point.
> > > >
> > > > > … about a decade ago, variational methods were extended to more amortised variational inference because the typical strong factorisation assumptions of old-style variational methods were so poor they led to ridiculous lower bounds to the marginal log likelihood, and inhibited approaches. They also restricted models to exponential families that were poor models for the highly non-linear setting people cared about. For example in your approach the choice of a slot Mixture Model for the pixel level modelling is really restrictive, and is probably why this required a benchmark with distinctive pixel distributions as opposed to normal image benchmarks where objects are associated with textures, shape etc that neural networks cope with very well. Given we care about realistic settings not toy game settings, the ability to cope with e.g. 3D environments is important, but the base pixel model here seems to throw away much of that. Arguably one could map pixels into a modern video-representation space and work in that, but even then the slotMM seems lacking power for the task.
> > > >
> > > > Thanks for the point; yes, this  is a good point because as you noted, the Slot Mixture Model is restrictive in its expressiveness in the sense it can only deal with monochromatic objects. We acknowledge this specifically in the paper (in the introduction), that we were not concerned with doing complex object segmentation of objects with more complex textures and multiple colors, which is why we restricted the types of sprites in Gameworld to ones with single colors. We still think the automatic growth of the sMM to identify the number, colors, shapes, and presence of objects in the scene is quite interesting and useful, since it makes the algorithm able to learn and grow its representation of the number, shapes, and colors of objects in the scene simultaneously. We can imagine hierarchical extensions to the sMM that can deal with more complex, multi-colored and -textured objects  without abandoning the use of conditionally-conjugate likelihoods, but believe it to be beyond the scope of the current contribution. You could also imagine fitting the sMM on features already extracted from a pretrained object segmentation model, to be able to circumvent the need to write a generative model directly over pixels. Although then a lot of the work of 'scaling to high dimension' would be handled by an amortized inference layer rather than a "pure" inferential layer.
> > > >
> > > > > I guess the other benefit of explicit factorisation is the reduced computation associated with the need to train a neural approximator, versus the extra computation associated with the iterations of the variational approximation scheme. I would love to see a more detailed like by like comparison of these. If the case here is that we don't need amortised inference, or we can do fairly well at less computational costs, it would be good to see that as the explicit target of the paper: have two exactly identical state: have two exactly identical state-models with a state-of-the art amortisation scheme and the variational approximation scheme compare. The problem is that even here, I believe the direct variational inference scheme hinders model power. It still seems as restrictive as it was back in the day.
> > > >
> > > > We agree this is certainly and interesting direction but in our opinion that is a separate undertaking with a different scope. Rather, we decided to select representative,  highly-sample-efficient SOTA architectures from the RL literature to generally understand what the "core priors, Bayesian-all-the-way” choices would get us, relative to gradient-based optimization and amortized inference. We do agree that this sort of comparison would be very useful and interesting, even for a fixed (non dynamic) model like a Gaussian mixture model or mixture of linear-Gaussians, where in one case the model is fit with amortized inference and in the other with CAVI / EM.

---

> > > > > ### Author Response · Authors · 2025-11-26
> > > > > **Stated advantages of not using gradients/backpropagation, Related Work point, Addressing the Methods suggestions**
> > > > >
> > > > > > Again, simply choosing not to use tools that are available is not itself a selling point for a method. Rather there needs to be some additional claim that establishes the point of the paper.
> > > > >
> > > > > As we stated before, we are not asserting that not using gradients is a selling point for its own sake -- we explicitly discuss _why_ not using gradients / replay buffers / statically-learned encoders with no parameter uncertainty, is useful. These additional benefits are expounded upon, in our view, throughout the results and figures, via claims about sample efficiency, parameter efficiency, computational expense, and interpretability.
> > > > >
> > > > > >I would like strongly discourage pushing such things to appendices. You are building your work on the research of others, and you are situating your work within the field for the benefit of the understanding of the reader. Denigrating "Related Work" to an Appendix shows disrespect to that previous work and feels dismissive. Something else should go. In particular the current work felt somewhat narrow in what it believed was the scene for this, and potentially perpetuates a narrow viewpoint on the field. That is not helpful to our dear readers, and further discourages scholarly work by others.
> > > > >
> > > > > That is a fair criticism, and is exactly why, in response to your first set of reviews, we moved the Related Work to the main text, which we believe gives a fair overview of the related work in world models, object-centric representations and the relevant modelling paradigms in Bayesian inference (rSLDS/SLDS, structure learning, variational Bayes)..
> > > > >
> > > > > >Again, choosing not to use tools available is not itself a feature. Why is not having replay buffers or gradient computations (both of which are very powerful) a strength? Instead it requires other, less efficient computations that do not have the convergence properties of gradient optimizers.
> > > > >
> > > > > We explained in the paper why it's a strength and not eschewing those tools for the sake of doing so. We show that AXIOM endows  _more_, not less, efficient computations that use less data and less memory (due to the fact that it's streaming variational inference, so we don't hold the full context of observation / latent history in memory), and leads to more sample efficient models due to the regularizing effect of priors.
> > > > >
> > > > > > There are a few difficulties in the writing, and it could be clearer, and I am sure this is fixable. First you define your latents as {O,z} but then you split (?) the z into z and s, overloading z a bit, which is slightly confusing. Also you say "object-centric latent variables Ot = {O(1)_t , . . . , O(K)_t }, associating each pixel to one of K slots using the assignment variable z_t,smm. But if this description is accurate z_t and O_t are the same thing. Rather O_t are instead features of the objects and z_t is the pixel to object assignment, but that is not clearly described until after. It is worth making clear what O are when they is introduced. It would be better to introduce the latent space more systematically: i.e. start with the bits (z_t, s_t, renamed Z_t,smm), defining what each is, and then collect into compositions where helpful: i.e. collect z_t,s_t into O_t. Likewise the parameters.
> > > > >
> > > > > This is very useful feedback, thank you, and apologies for the earlier confusion of the methods. In response to this, we have now rewritten the methods section (Section 3, lines 167-202)  to first explain the individual variables (the "bits” as you put it) and only afterwards to collect those individual variables into compositions thereof like $\mathcal{O}_t$, $\mathcal{Z}$, etc.. We believe that the Methods now reads more cleanly / easily - thanks again for this suggestion.

---

> ### Author Response · Authors · 2025-11-20
> **Situating work with Bayesian RL, MBRL, and world modelling; answer to whether model is Bayesian over states vs parameters**
>
> > Altogether, I do not see any indication as to how the authors relate this to other Bayesian reinforcement learning implementations, or to general model-based reinforcement learning, and world modelling…and this work should really be set in the context of the existing RL literature.
>
> We do have a related works section in the Appendices (Appendix F) where these relationships are made clear. To summarize: our approach stands in contrast to other model-based RL approaches for the following important reason: when doing with high-dimensional input streams like pixels, standard MBRL always uses some sort of amortized inference scheme (e.g., in the learned encoder or transition model), whereas we use explicit probability distributions and likelihoods throughout. In the introduction and conclusion, we emphasized as a key strength of AXIOM, that the method doesn’t use gradient-based optimization of parameters of black-box function approximators, since all the updates are fixed point updates (variational E- or M-steps) to conditionally conjugate posterior distributions. We have also emphasized this point in the conclusion section (original submission, unchanged), lines 525-526: “Importantly, it does so without relying on neural networks, gradient-based optimization, or replay buffers.”
>
> Unfortunately due to space constraints we did not have room to put the Related Works section in the main text, but rather had to place it in the Supplementary Appendices (Appendix F). However, since we may expand the main text to 10 pages as part of the rebuttals, now we have moved a shorter and tighter version of the Related Work section (now organized into three sections — World Models, Object-Centric Factorization, and Bayesian Inference). In the new Related Work section, we compare and relate our architecture to both world models and Bayesian model-based RL more explicitly, as you requested. However, to make this novelty and relationship between our approach and other MBRL approaches appear more centrally in the text, we have added citations to similar MBRL architectures (which are also Bayesian in essence, yet still rely on modules learned with backpropagation) when making these statements.
>
> Specifically, to make these relationships clear, we have made the following changes:
> - in the introduction, line 83-86, we have supplemented this claim with an additional, literature-relevant comment (new addition in bold):  “This eliminates the need for replay buffers or gradient computations, and enables online adaptation to changes in the data distribution, **a feature that has not been demonstrated in previous (even Bayesian) model-based RL approaches \citep{karl2016deep,zhang2019solar, hafner2025mastering}.**”
> - in the new Related Works section in lines 137-142: “AXIOM is also an example of a model-based RL architecture, but notably its world model is explicitly probabilistic and interpretable. Unlike previous world models (even probabilistic ones like \cite{karl2016deep, zhang2019solar}), AXIOM models the latent world state using explicit object-centric representations that have directly interpretable features like position, velocity, color, and shape, and does so using only closed form variational updating.”
>
> > The exact relationship to Bayesian agents is a little lost in the actual formalism. There is some confusion as to be Bayesian (over model parameters) and having probabilistic state updates (which this shares with most POMDP related formalisms)....I don't believe the priors over parameters has been defined here.
>
> AXIOM treats both model parameters and states probabilistically. In our opinion we have been explicit about this in the Methods section (Section 2) by doing the following: we write out the generative model as a joint distribution (Equation 1), and then proceed to describe the functional forms of the priors and likelihoods in each of its constituent distributions in the following sections: the sMM (slot mixture model), iMM (identity mixture model), tMM (transition mixture model) and rMM (recurrent mixture model). Priors for each of the subsets of parameters of those models are described in the main text and written out explicitly in the Appendix. Finally, we conclude (right before the beginning of Section 3.1, the section titled Variational Inference) with a description of the variational posteriors we use and the fact we use CAVI (coordinate ascent variational inference) to update posteriors over both the latent states (so we have state uncertainty) and parameters (e.g., the parameters of each mixture model).
>
> (further responses continued below)

---

> ### Author Response · Authors · 2025-11-20
> **Clarification of planning algorithm; problem statement; and relationship to planning-as-inference and control-as-inference**
>
> >The planning aspect is a little cryptic, and lacks specifics.
>
> Thank you for pointing this out. Due to space constraints the main text only provides a high-level description, but the planning algorithm is fully specified in Appendix A.11 (including Algorithm 3). Let us briefly summarize the procedure here.
> Conceptually, we use active inference: each candidate policy $\pi = a_{0:H-1}$​ is scored by its expected free energy $G(\pi)$, which combines predicted reward and information gain about the recurrent mixture state. Enumerating all policies is infeasible, so we approximate the policy posterior with a sampling-based MPC scheme inspired by CEM/MPPI.
> Concretely, at each time step we:
> - Maintain a horizon-wise categorical proposal $p(a_\tau)$ over discrete actions for each future time $\tau=0,…,H−1$
> - Sample $P$ candidate policies of length $H$, where $P - R$ trajectories are drawn i.i.d. from $\text{Cat}(p(a_\tau))$ and $R$ trajectories are “exploratory” random walks.In addition we inject the previous best plan, plus a small set of constant-action sequences, to ensure coverage of simple control primitives and persistence of previously good plans.
> - Evaluate each policy by rolling it forward S times through AXIOM’s world model, obtaining per-step predictions of reward ​ and information gain. We then compute a temporally discounted sum of (reward + info-gain) over the horizon time H and average it across S independent samples.
> - Refit the proposal by taking the top-K policies with lowest expected free energy, forming for each $\tau$ in the planning horizon an empirical action histogram over their actions, converting this to probabilities with a tempered softmax, and updating via an exponential moving average:
>  $p(a_\tau)^{\text{new}} = \alpha_{\text{smooth}}\;\hat p(a_\tau) + (1-\alpha_{\text{smooth}})\;p(a_\tau)^{\text{old}}$
> - Execute the first action of the current best plan, then repeat at the next time step with the updated proposal and best plan (i.e., warm-started planning).
>
> Relative to standard CEM/MPPI, the main differences are: (i) we work in discrete action space and fit categorical proposals instead of Gaussians (relative to Okada and Taniguchi 2019); (ii) elite samples are softly weighted via $exp⁡(−G)$ when refitting; (iii) we inject structured trajectories (previous best plan, constant-action sequences, randomized walks) into the candidate set; and (iv) we perform one CEM-style update per environment step, carrying the refined proposal forward in time instead of running multiple inner iterations. These are practical modifications rather than a separate algorithmic contribution, but they make the planner well-suited to AXIOM’s discrete-action, model-predictive, expected-free-energy setting.
>
> > The paper would be improved if it explicitly and precisely defined its problem setting, assumptions, and way the results are assessed. There is no problem statement at all.
>
> Thank you for this comment. While we did not label a separate subsection as “Problem statement,” our intention was to make the setting, assumptions, and evaluation protocol clear through Section 2 (Methods) and Section 3 (Results).
> Concretely, the problem setting we address is:
> - an online, episodic RL problem on the Gameworld-10k suite,
> - where an agent interacts with a Markovian 2D game environment from raw pixel observations, chooses discrete actions, and receives scalar rewards
> - under a strict interaction budget of 10,000 environment steps per environment, with episodes reset as usual when terminal states are reached, and
> - the objective is to maximize undiscounted cumulative return within this budget using a learned world model and model-based planning.
>
> These elements are currently introduced in Section 1 and in the Gameworld-10k description in Section 3, and the probabilistic assumptions are made explicit in Section 2, where we write down the joint generative model, specify the functional forms of the likelihoods and priors for each mixture component (sMM, iMM, tMM, rMM), and then describe the variational posteriors and CAVI updates used for inference and learning. The way results are assessed is defined in Section 3: for each game we report performance curves over interaction steps, aggregating rewards across multiple random seeds
>
> > Does this relate strongly to the "RL as inference" ...?
>
> Yes it shares deep similarities with control-as-inference and planning-as-inference, but it’s importantly different because in control-as-inference, you obtain planning-as-inference by conditioning on a future sequence of observed ‘optimality’ variables (reward-like discrete states), and then framing planning as posterior inference on latent control variables, given this fixed set of future conditioned variables. In active inference-based planning, you instead have a prior over observations in the future (we can have a prior over them because they are unobserved from the perspective of the present).
> (responses continued below)

---

> ### Author Response · Authors · 2025-11-20
> **Continuation on planning-as-inference relationship; posterior factorisation assumptions; benchmark suitability**
>
> (response on planning-as-inference continued from before)
> This prior over observations is used in a generative model of future (action-conditioned) states, and factors into the model evidence or negative log marginal likelihood of that generative model of the future. Action selection is then converged into an inference or model-averaging problem in the following way: action or policy’s probability is proportional to its action-conditional Bayesian model evidence, where this model evidence scores how ‘likely’ the future trajectory conditioned on that action is. Importantly, this generative model of the future includes this ‘biased prior’ over future observations, which allows actions to be more probable in proportion to this reward-like quantity (the log of the observation prior) - and has implications for how information-gain factors into action selection (see .On the Relationship Between Active Inference and Control as Inference).
>
>  > The posterior factorisation assumptions seem to be worrying, in that it decouples things that are strongly coupled. The internal posteriors over the mixture parameters could be discussed further.
>
> This is a fair point, in the sense that mean-field posterior factorization definitely discards some of the information relative to having full joint distributional posteriors. That being said, the mean-field factorization in VB is quite a common assumption made, often for tractability reasons (to be able to obtain closed-form updates to the natural parameters of the mean-field distributions, which take a convenient exponential family form under this factorization). In addition, the independence assumption in the posterior doesn’t mean there is no coupling between factors – it just means the coupling is mean-field. The mean-field likelihood terms are always averaged with respect to the states of the other factors (the complement of the factor currently being updated), meaning there is still coupling between the posteriors in the updates themselves. Specifically, $\log q(\theta_i) \propto  \langle \log p(y | \theta) \rangle_{q(\theta_{j \neq i})} + \text{other terms} $. Any coupling in the likelihood between the $\theta_i$ <-> $\theta_j$’s will still affect either one of them via these mean-field expected log likelihood terms.
>
>
> > However the main weakness of the paper is that it dismisses evaluating the approach on any of the standard settings used in this context, instead defining their own unestablished setting just for them, so we can't tell how good it is as a test...
>
> We agree overall that this is a weakness of the paper, but we don’t think it’s one worth rejecting the paper wholesale for, based on the overall goal and scope of this paper’s contribution. We would like to emphasize that the intention of this paper is not to simply propose a new architecture that is more sample efficient and more performant than other architectures (not to diminish performance benchmarking; this also plays an incredibly important role in ML). We use the performance comparisons with BBF and DreamerV3 to make an overall point: that one can create model-based RL architectures for control tasks on high-dimensional input spaces, that eschew all use of gradient descent, replay buffers, and amortized inference via function approximation. AXIOM is the only (to our knowledge) structured world model with explicit distributions everywhere, that can engage in model based planning directly from a high-dimensional input space like pixels.
>
> We agree that ideally this architecture would also handle higher-dimensional scenarios with these same or similar core priors (e.g., MuJoCo, Minecraft), but we didn’t invest time into this because we wanted to spend our efforts emphasizing the strength and novelty of the architecture and the ideas themselves, which were best demonstrated on games where the core priors are met (namely, on Gameworld).
>
> (final response below on the reward modelling)

---

> > ### Author Response · Authors · 2025-11-20
> > **Clarification of the recurrent mixture model and how it models the dependency between state and reward**
> >
> > > The recurrent mixture model captures most of the structure of the model. It deals with the reward structure. i.e. the strong dependence of the instantaneous reward on the state, which has to be learnt. Yet it is quite worrying as that seems to be handled in 7 as an independent categorical distribution between state and reward. This can't make sense, yet that is what it says. Reward modelling is critical but is not dealt with explicitly and is hidden away here. That seems quite worrying, and I wonder here is there is actually any real reward modelling happening…. Please could you explain your reward modelling. This seems really unclear at the moment.
> >
> > The categorical distribution over the reward variable is not independent of the state – it is actually conditioned on the state via the mixture likelihood. In Equation (7) it is written as a mixture of categorical distributions, where the discrete mixing variable $s_{t,m,rmm}$ acts as an indicator variable and selects one of the component mixture distributions. Perhaps you didn’t notice the exponentiation by $s_{t, m, rmm}$ on the RHS of the equation?
> > We hope this explanation establishes that the reward variable explicitly depends on the recurrent mixture model discrete state $\mathbf{s}_{t,rmm}$. The reward’s correlation with other variables (like the continuous latent states, slot features, action, etc.) is mediated by the common recurrent mixture switch state, which selects mixture likelihoods over the joint states of (reward, action, discrete slot states, etc.). Correlations/dependencies between different variables are thus captured by learning the mixture components’ parameters via variational Bayesian updates. These updates are described in more detail in Appendix A.9 (Equations (52) - (56)).

---

> ### Comment · Reviewer_ccyd · 2025-11-21
> **Continued Response**
>
> "when doing with high-dimensional input streams like pixels, standard MBRL always uses some sort of amortized inference scheme (e.g., in the learned encoder or transition model), whereas we use explicit probability distributions and likelihoods throughout. "
>
> That is quite an assertion. Typically MBRL does use an amortised inference scheme, but all that means is that it leverages more powerful models and more direct computation to do the approximate inference: about a decade ago, variational methods were extended to more amortised variational inference because the typical strong factorisation assumptions of old-style variational methods were so poor they led to ridiculous lower bounds to the marginal log likelihood, and inhibited approaches. They also restricted models to exponential families that were poor models for the highly non-linear setting people cared about. For example in your approach the choice of a slot Mixture Model for the pixel level modelling is really restrictive, and is probably why this required a benchmark with distinctive pixel distributions as opposed to normal image benchmarks where objects are associated with textures, shape etc that neural networks cope with very well. Given we care about realistic settings not toy game settings, the ability to cope with e.g. 3D environments is important, but the base pixel model here seems to throw away much of that. Arguably one could map pixels into a modern video-representation space and work in that, but even then the slotMM seems lacking power for the task.
>
> Finally in an online setting such as MBRL the loss associated with online repeated factorisation projection is also problematic.
>
> That this paper returns to explicit factorisation for variational inference is indeed interesting, and were we to be able to infer strong performance in benchmarks in established benchmarks, that might be really helpful (a kind of "factorisation is all you need"). Unfortunately at the moment the fact that the paper built its own evaluation test somewhat implies it was tried on standard benchmarks and didn't do so well, otherwise such results would be reported.
>
> I guess the other benefit of explicit factorisation is the reduced computation associated with the need to train a neural approximator, versus the extra computation associated with the iterations of the variational approximation scheme. I would love to see a more detailed like by like comparison of these. If the case here is that we don't need amortised inference, or we can do fairly well at less computational costs, it would be good to see that as the explicit target of the paper: have two exactly identical state: have two exactly identical state-models with a state-of-the art amortisation scheme and the variational approximation scheme-of the art models with a state-of-the art amortisation scheme and the variational approximation scheme and compare. The problem is that even here, I believe the direct variational inference scheme hinders model power. It still seems as restrictive as it was back in the day.
>
> Again, simply choosing not to use tools that are available is not itself a selling point for a method. Rather there needs to be some additional claim that establishes the point of the paper.

---

> > ### Comment · Reviewer_ccyd · 2025-11-21
> > **Continued Response**
> >
> > Unfortunately due to space constraints we did not have room to put the Related Works section in the main text, but rather had to place it in the Supplementary Appendices (Appendix F).
> >
> > I would like strongly discourage pushing such things to appendices. You are building your work on the research of others, and you are situating your work within the field for the benefit of the understanding of the reader. Denigrating "Related Work" to an Appendix shows disrespect to that previous work and feels dismissive. Something else should go. In particular the current work felt somewhat narrow in what it believed was the scene for this, and potentially perpetuates a narrow viewpoint on the field. That is not helpful to our dear readers, and further discourages scholarly work by others.
> >
> > " “This eliminates the need for replay buffers or gradient computations, and enables online adaptation to changes in the data distribution, a feature that has not been demonstrated in previous (even Bayesian) model-based RL approaches \citep{karl2016deep,zhang2019solar, hafner2025mastering}.”
> >
> > Again, choosing not to use tools available is not itself a feature. Why is not having replay buffers or gradient computations (both of which are very powerful) a strength? Instead it requires other, less efficient computations that do not have the convergence properties of gradient optimizers.

---

> > > ### Comment · Reviewer_ccyd · 2025-11-21
> > > **Continue Response**
> > >
> > > "AXIOM treats both model parameters and states probabilistically. In our opinion we have been explicit about this in the Methods section (Section 2) by doing the following: we write out the generative model as a joint distribution (Equation 1), and then proceed to describe the functional forms of the priors and likelihoods in each of its constituent distributions in the following sections: the sMM (slot mixture model), iMM (identity mixture model), tMM (transition mixture model) and rMM (recurrent mixture model). Priors for each of the subsets of parameters of those models are described in the main text and written out explicitly in the Appendix. Finally, we conclude (right before the beginning of Section 3.1, the section titled Variational Inference) with a description of the variational posteriors we use and the fact we use CAVI (coordinate ascent variational inference) to update posteriors over both the latent states (so we have state uncertainty) and parameters (e.g., the parameters of each mixture model)."
> > >
> > > There are a few difficulties in the writing, and it could be clearer, and I am sure this is fixable. First you define your latents as {O,z} but then you split (?) the z into z and s, overloading z a bit, which is slightly confusing. Also you say "object-centric latent variables Ot = {O(1)_t , . . . , O(K)_t }, associating each pixel to one of K slots using the assignment variable z_t,smm. But if this description is accurate z_t and O_t are the same thing. Rather O_t are instead features of the objects and z_t is the pixel to object assignment, but that is not clearly described until after. It is worth making clear what O are when they is introduced. It would be better to introduce the latent space more systematically: i.e. start with the bits (z_t, s_t, renamed Z_t,smm), defining what each is, and then  collect into compositions where helpful: i.e. collect z_t,s_t into O_t. Likewise the parameters.
> > >
> > > Your full model is Markovian (in the latent space), and hence to cover POMDP settings it relies on integrating the unobserved inferential elements in the latents, so the capability depends on whether the latents can expand to include the unobserved elements. It would be interesting to give an example where you expect that to be possible in your framework, and a relevant example where it would not be possible (as it relies on an object-centric representation, and so non-object elements (such as global change factors like cyclic lighting changes) feel hard to capture.
> > >
> > > I thank you for writing out the full model in (1). the recurrent mixture model captures many of the critical model components all in one.
> > >
> > > When you go through each element of this full joint, it would be most helpful if you give the component by equation, not just by name. There are lots of moving parts here. Expecting the reader to remember which part you mean by each term is asking a lot, whereas the equation for the term is more recognisable.
> > >
> > > I still do not see where (in the main body) you define what the priors are for parameters? Maybe I am being hopeless, but if I am others will be too. Perhaps put this in or make it more prominent. This has to be in the main text, as you are relying on tractability to make this whole thing work, so you have to define it to demonstrate it is indeed tractable to compute the variational posterior.
> > >
> > > You rely on (variational) EM to update the Q distributions. Perhaps state the typical number of iterations needed. EM in these sort of models has quite unpleasant local minima and local modal collapse which is why everyone stopped using such methods for sequential inference. Some comment on that would help. I assume this is mitigated somewhat by the split-and-merge heuristic, but it would be great if you could say something.

---

> > > > ### Comment · Reviewer_ccyd · 2025-11-21
> > > > **Continued response**
> > > >
> > > > I agree you are clear at training that you are fully Bayesian over parameters and latents under a variational approximation. However I do not find that clear what that means. Under the EM as you describe it, this involves variational updates of the model on all the trajectories before a single step on the parameters can be made (i.e. batch-EM). If that is really the case this seems incredibly infeasible for anything of meaningful size. Otherwise I assume you do some minibatch update scheme, but it is not obvious how to do that within your framework, and it is not described AFAICS, even in the appendix. Furthermore at planning time, how do you handle the uncertainty over the extrinsic and intrinsic variables.
> > > >
> > > > Also in planning you say "In active inference, policies π = a0:H are selected that minimize expected Free Energy (Friston et al.,
> > > > 2017)" - this is not actually correct. If this were the case then only the certainty of the reward would matter. You also state that  log p(r_τ |O_τ , π)| z ) is the Utility. It is not: it is the distribution of rewards, not any Utility function of the rewards (as defined in 3). furthermore you do not integrate out r, but neither do you state what it is. This description is at variance with Algorithm 3.
> > > >
> > > > (The reward is also discrete, which is somewhat constraining).
> > > >
> > > > Basically, the planning description cannot be right as it is missing a specification of r, and in no way tries to maximize any function of reward.
> > > >
> > > > Hence I suggest that the overall approach (which has a lot of bits!) is not really fully clear.

---

> > > > > ### Comment · Reviewer_ccyd · 2025-11-21
> > > > > **Final comments**
> > > > >
> > > > > "This is a fair point, in the sense that mean-field posterior factorization definitely discards some of the information relative to having full joint distributional posteriors. That being said, the mean-field factorization in VB is quite a common assumption made, often for tractability reasons (to be able to obtain closed-form updates to the natural parameters of the mean-field distributions, which take a convenient exponential family form under this factorization). In addition, the independence assumption in the posterior doesn’t mean there is no coupling between factors – it just means the coupling is mean-field. The mean-field likelihood terms are always averaged with respect to the states of the other factors (the complement of the factor currently being updated), meaning there is still coupling between the posteriors in the updates themselves. Specifically, . Any coupling in the likelihood between the <-> ’s will still affect either one of them via these mean-field expected log likelihood terms."
> > > > >
> > > > > It is true this was a common assumption in VB, mostly because there was little choice to maintain tractability. But it is exactly why VB has gone out of favour for flexible inference and planning in favour of more amortized inference approaches, because this failure was so problematic (along with the implied constraint on the model form). If this paper was persuasive that we threw out the baby with the bathwater in making that transition, I would be very interested, but in experimental terms, it does not convince me.
> > > > >
> > > > > In terms of this paper and getting it published. I think it needs a clear delineation of the exact _realistic_ domain (if there is one) where this is going to be the best thing to do in a fully fair comparison. Maybe there are computational benefits? In which case focus on that. But at the moment the paper is wonderful in that it builds together a lot of bits of variational inference as building blocks beyond what others have done. It is beautiful that we can do full approximate inference over all the bits. But, it is (in a rudely put analogy, I apologise) less wonderful in establishing that the result is little more than a Rube-Goldberg machine equivalent for planning: a complicated but inflexible and wobbly way of cooking an egg in a very specific situation...
> > > > >
> > > > > In summary +positive+ once the bits around planning are fixed, I think this paper will hold together. There are a lot of bits, but they appear correct, and establish a towering example of compositional variational inference. -negatively- I am unconvinced it has a place on a modern stage, and the experiments do little to alleviate my concerns about the constraints on both the model and the mean field approximations for inference make it too inflexible for modern problem sets. I'd love it to be otherwise, but sadly my desire is not the same as reality.

---

> ### Author Response · Authors · 2025-11-26
> **Incorporating non-object information into the POMDP; explicitness about priors;**
>
> > Your full model is Markovian (in the latent space), and hence to cover POMDP settings it relies on integrating the unobserved inferential elements in the latents, so the capability depends on whether the latents can expand to include the unobserved elements. It would be interesting to give an example where you expect that to be possible in your framework, and a relevant example where it would not be possible (as it relies on an object-centric representation, and so non-object elements (such as global change factors like cyclic lighting changes) feel hard to capture.
>
> This is an interesting idea! We have some notion of conditioning on non-object elements (action, reward), but these are observed, not latent, variables. The natural way to accommodate unobserved, but non-object-tied, latent variables, would be through additional discrete variables in e.g., the recurrent mixture model or another mixture model. One could imagine introducing another latent variable in the recurrent mixture model, which is not object-factorized (and hence shared across objects), which nonetheless conditions or selects among the object-factorized likelihoods. In this case, the updates to the posterior assignments of that global switch variable(s) would simply become the expected log likelihood assigned to each (continuous, discrete) tuple of object data for the rMM, summed over the object-specific likelihoods. Indeed, in an earlier iteration of the rMM we did have it be a single shared model across objects (as the tMM currently stands), rather than object-factorized. In this case, the rMM switch states did encode more 'global’ switch properties of the environment, rather than object-specific ones. We found for Gameworld 10k that the object-factorized edition was more performant in practice, probably because of the object-specific dynamical rules. Another way to do this and bypass objects entirely, is to attach these global latents to a likelihood (e.g. a simple GMM or mixture of linear-Gaussian models) directly over the pixel-level data or other data (rewards, actions), with the thought that the latents of this model would encode global changes in the reward or pixel distribution.
>
> > I still do not see where (in the main body) you define what the priors are for parameters? Maybe I am being hopeless, but if I am others will be too. Perhaps put this in or make it more prominent. This has to be in the main text, as you are relying on tractability to make this whole thing work, so you have to define it to demonstrate it is indeed tractable to compute the variational posterior.
>
> We would really love to put the prior equations in the main text, but unfortunately it's not feasible given the space constraints (started with 9 pages, now can increase to 10 pages during the rebuttal phase) and all the other things that are important to put in the main text (main likelihood equations, the results figures and tables, and the newly re-incorporated related work). This is why we chose to describe them with words (like NIW or truncated stick-breaking prior) for most of AXIOM's components. This would be even harder now with the changes to the latent state notation which have added extra space. So  it's sadly not tenable to put in the priors as well. We have attempted to be very explicit with the priors in each section of the Appendix (we have made them even clearer now than before, even adding Dirac delta priors for the 'frozen’ variables like the fixed matrices in the sMM and tMM), but just to be very specific, we'll list out here the equations of the appendix where you can find the parameter priors and their exact functional forms:
>
> - Slot Mixture Model: A.2, Equations (14) - (18)
> - Identity Mixture Model: A.6, Equations (27) and (28)
> - Transition Mixture Model: A.7, Equations (32) - (36)
> - Recurrent Mixture Model: A.8, Equations (39) - (41)

---

> > ### Author Response · Authors · 2025-11-26
> > **Number of iterations used for VBEM; explicitness about minibatches of data; uncertainty in planning**
> >
> > > You rely on (variational) EM to update the Q distributions. Perhaps state the typical number of iterations needed. EM in these sort of models has quite unpleasant local minima and local modal collapse which is why everyone stopped using such methods for sequential inference. Some comment on that would help. I assume this is mitigated somewhat by the split-and-merge heuristic, but it would be great if you could say something.
> >
> > Thanks for noticing this. We have now added the following sentence in lines 331-334 of the main text, to be clear about the number of iterations we use at each timestep (1 E-step, and 1 M-step):  “**We use using a single iteration of CAVI for both the latent updates (E-step) and parameter updates (M-step) per timestep, resulting in a fast, streaming form of inference and learning \citep{wainwright2008graphical}.**” Note that during the mixture model expansion, we do one M step per new component that we add, so although it's one "total M step”(when all components are considered together), this in practice is broken down into several M steps, one for each new component that is added per the growing-heuristic. Also just to be clear about our structure learning algorithm:  we are not doing “split and merge” as described in SMEM 1998 -- we are doing model expansion (adding mixture model components, for datapoints that are not adequately explained by existing components, which is determined based on the CRP heuristic) and then periodically merging components with Bayesian model reduction (only considering merges that sufficiently increase the ELBO). Algorithms 1 and 2 of Appendix A explain this.
> >
> > > I agree you are clear at training that you are fully Bayesian over parameters and latents under a variational approximation. However I do not find that clear what that means. Under the EM as you describe it, this involves variational updates of the model on all the trajectories before a single step on the parameters can be made (i.e. batch-EM). If that is really the case this seems incredibly infeasible for anything of meaningful size. Otherwise I assume you do some minibatch update scheme, but it is not obvious how to do that within your framework, and it is not described AFAICS, even in the appendix.
> >
> > We explain this in the first section of the methods on the Slot Mixture Model (line 212): “AXIOM processes sequences of RBG images one frame at a time”. However, to make this more explicit, we have now also added this line in the variational inference section of the main text (line 330-332, new text in bold): “We update the posterior over latent states $q(\mathcal{Z}_{0:T})$ (i.e., the variational E-step) using a simple form of forward-only filtering and update parameters using coordinate ascent variational inference **(CAVI) that operates on single image frames (i.e., minibatches of $N=H\times W$ pixels).**”
> >
> > >Furthermore at planning time, how do you handle the uncertainty over the extrinsic and intrinsic variables.
> >
> > This is a good question, thanks for raising it. In the planning objective itself (the utility + info gain), uncertainty over parameters is accounted for only in the parameter information gain over the recurrent mixture model's Dirichlet parameters. We have not discussed how we handle uncertainty in the other latents during the rollouts, so we have added the following section to Appendix A.11 where we detail this in lines 1585 - 1598 (leaving ellipses for mathjax that wasn't rendering):
> >
> > "In practice, to generate the predictive posterior $q(\mathcal{O}_{\tau} | \pi)$,we do not analytically compute the expectation shown in Equation (67) , but rather sample from the posterior over the recurrent mixture model switch variable [...] and then use the sampled rMM switch state to sample a transition mixture model switch state [...]. This is done using the rMM discrete likelihood distribution which describes the probabilities of different tMM switches given rMM switch states. The selected transition switch state is then used to propagate the continuous object states (positions, velocities) through the corresponding transition mixture component analytically (with its learned linear-Gaussian parameters). We then use the next predicted continuous and discrete states (clamping the action input to that entailed by the policy, clamping the identity inputs to the posterior values [...] at the beginning of the rollout, and omitting the reward input since we are trying to predict it) to generate an expectation over the rMM switch state – a “fictive inference step”. This then serves as the rMM posterior over switch states, for the next timestep of the rollout. Finally, to predict the reward at a given timestep, we use the expected reward distribution under the predicted rMM switch state and compute the expected utility by taking the dot-product of the predicted reward distribution with the biased reward prior with logits = $[-1, 0, +1]$.

---

> ### Author Response · Authors · 2025-11-26
> **continued explanation of planning; clarification of EFE and biased reward model**
>
> (continued from previous comment) So in short: we don't propagate uncertainty analytically for the transition mixture and recurrent mixture switch states, but only do so through samples, but we generate analytic uncertainty for the reward distribution (simply a categorical distribution over the different reward observations) and use the uncertainty in the identity latent (albeit fixed throughout the rollout) to condition inference of the rMM switch state, which is then sampled.
>
> > Also in planning you say "In active inference, policies π = a0:H are selected that minimize expected Free Energy (Friston et al., 2017)" - this is not actually correct. If this were the case then only the certainty of the reward would matter. You also state that log p(r_τ |O_τ , π)| z ) is the Utility. It is not: it is the distribution of rewards, not any Utility function of the rewards (as defined in 3). furthermore you do not integrate out r, but neither do you state what it is. This description is at variance with Algorithm 3.
> …Basically, the planning description cannot be right as it is missing a specification of r, and in no way tries to maximize any function of reward.
>
> Apologies for the lack of clarity – this is actually very helpful that you noticed this, because it made us realize we did not correctly specify the EFE objective – specifically, in the original version we failed to indicate that the reward model used in the planning objective is different than that used in inference and learning. Under active inference, the expected utility is not computed only using the reward distribution learned through the rMM, but also using a ‘biased’ distribution over rewards, a fixed prior distribution. This is the way active inference models encode goals or reward functions. It has some advantages over using standard reward functions in certain contexts (e.g. when your preferences themselves are random variables which must be learned or inferred), but for all intents and purposes here, it plays an identical role to a utility function. We have now clarified/corrected the planning objective to use this biased prior over future observations, denoted $\tilde{p}(r_\tau)$to clarify this.
> Specifically, we have corrected Equation (10) to use the “biased prior over reward observations” $\tilde{p}(r_\tau)$, and added the following new text to Section 3.2 on Planning (new text in bold, lines 376-401, with ellipses indicating unrendered mathjax): “The expected per-timestep utility **[...] is computed using the predictive distribution over object-centric states under a policy $\pi$, which is calculated using rollouts from the learned model, [...]. The tilde notation $\tilde{p}(r_{\tau})$ indicates an `optimistically-biased' prior over for reward observations; this importantly is a fixed prior which differs from the reward contingencies learned by updating the rMM and is the critical ingredient that enables goal-directed behavior under active inference \citep{parr2019generalised}. In this context, the biased prior can be directly analogized to a utility function used in traditional model-based RL.\footnote{We compute the expected utility by averaging the (unbiased) reward expected under the policy-conditioned predictive distribution with the log probabilities of the biased prior over the reward outcome: [...].}** We have also adjusted Appendix A.11 on Planning to include more details about how these rollouts are generated in practice.
>
> Just as an aside: there is debate in the active inference literature about the mathematical status of this 'biased prior over observations': whether it's the prior of a separate, non-veridical generative model just used for planning (see "Whence the Expected Free Energy?” by Millidge, Tschantz, and Buckley 2021) or whether it can be absorbed into a single, unified generative model used for both planning and inference, as in the “generalised free energy” objective of Parr and Friston (2022). However, we do not want to distract this work with the nuances of this debate, so we have decided to simply declare it a "biased prior over observations” and say that it only appears in the objective used for planning, and emphasize that it is not used anywhere in the rMM inference / learning phase, so it is effectively equivalent to the utility function/reward function common in the MBRL literature.

---

> ### Author Response · Authors · 2025-11-26
> **Final remarks**
>
> > In terms of this paper and getting it published. I think it needs a clear delineation of the exact realistic domain (if there is one) where this is going to be the best thing to do in a fully fair comparison. Maybe there are computational benefits? In which case focus on that. But at the moment the paper is wonderful in that it builds together a lot of bits of variational inference as building blocks beyond what others have done. It is beautiful that we can do full approximate inference over all the bits. But, it is (in a rudely put analogy, I apologise) less wonderful in establishing that the result is little more than a Rube-Goldberg machine equivalent for planning: a complicated but inflexible and wobbly way of cooking an egg in a very specific situation…
>
> We understand your reservations, but we do still maintain that there's inherent value in the contribution due to its novelty and the demonstrated benefits (sample efficiency in gameplay, interpretability, computational efficiency, parameter efficiency – recall Table 2). Whatever reservations or doubts one may have about the benchmark, there are also genuine mathematical innovations that allowed us to scale explicitly Bayesian models to this context and use it in a streaming, one-frame-at-a-time fashion, like online mixture model growing and reduction; a novel variant of the rSLDS which uses a mixed discrete/continuous likelihood to infer the switch, instead of a softmax discriminative likelihood (c.f., Linderman et al. 2016). Despite the (acknowledged) scale limitations, we still maintain this is the only known scaled-up scenario in the literature (high-scale meaning – going directly from pixels) of an _explicitly Bayesian_ model-based RL agent. And again, by explicitly Bayesian we mean a fully structured world model, specified with named, exponential-family probability distributions – no amortization, no backpropagation.
>
> >In summary +positive+ once the bits around planning are fixed, I think this paper will hold together. There are a lot of bits, but they appear correct, and establish a towering example of compositional variational inference. -negatively- I am unconvinced it has a place on a modern stage, and the experiments do little to alleviate my concerns about the constraints on both the model and the mean field approximations for inference make it too inflexible for modern problem sets. I'd love it to be otherwise, but sadly my desire is not the same as reality.
>
> Thanks for the summary of your critique. We hope to have satisfied your requests with respect to planning and more clarity on the methods section (thanks again for the suggestions; we believe it positively impacted the paper), hopefully enough to warrant an increase in our score. Besides that though, we also do acknowledge and understand your concerns about the dataset selection and benchmark limitations, but for all the reasons stated above, we maintain that this is a worthwhile contribution for a venue like ICLR.

---

### Meta-Review · Area_Chair_JPoo · 2025-12-29

**Summary:**

The reviewers agreed that the method was technically correct and conceptually consistent. The reviewers’ concerns collectively informed a borderline or a slightly positive final assessment. The main reservations stemmed from
- the narrow and possibly self-serving benchmark,
- overstatement of claims and limited engagement with broader literature
- unclear early exposition of planning and inference details.

My recommendation, therefore, leaned toward weak rejection: acknowledging the paper’s novelty, interpretability, and intellectual contribution, while cautioning that its empirical scope and generalization evidence remain limited.

In short, AXIOM was viewed as a rigorous demonstration of compositional variational inference, whose importance is conceptual rather than empirical. It is not yet a solution fit for large-scale RL practice.

**Reviewer Concerns:**

Concerns Successfully Addressed
- Reviewers ccyd and omuG had questioned the mathematical clarity of AXIOM’s design. The rebuttal directly addressed this. The authors explained that rewards are indeed modeled as conditioned on the latent “mode” variable within the recurrent mixture structure. They also detailed the use of sampling-based MPC under EFE. Additionally, pseudocode provided during the rebuttal connected the inference steps with the planning process.

- The original submission suffered from unclear variable definitions and inconsistently explained priors. The authors responded by revising variable names, defining priors explicitly in text, and reorganizing the Methods section to follow a more modular structure. These changes were well received and eliminated early confusion. While the notation-heavy exposition was still viewed as complex, critical notational inconsistencies were fixed, and the reviewers acknowledged that the model was now properly specified.

- Reviewers criticized the authors for positioning AXIOM as if it represented a general blueprint for reinforcement learning. The rebuttal shifted this framing considerably. The authors explicitly called AXIOM a “proof of concept” rather than a system competing with large-scale deep RL agents.

- The most visible improvement after rebuttal was in the treatment of prior work. Originally, Reviewer ccyd took issue with the Related Work section being confined to the appendix and with an overemphasis on the active-inference lineage. In response, the authors expanded citations to both classical and modern Bayesian RL literature and clarified that AXIOM integrates active-inference principles rather than wholly redefining them.


Concerns That Remain Outstanding
- Despite framing Gameworld‑10k as a controlled, diagnostic benchmark, the rebuttal did not change the empirical landscape of the paper. Reviewers continued to view this as the largest unresolved issue. The benchmark is tightly tailored to AXIOM’s inductive biases. This raised legitimate doubts about overfitting the experimental setup to the model. Several reviewers explicitly asked for at least limited testing on standard benchmarks to assess whether AXIOM’s principles would hold in less tailored environments. The authors’ response is not persuasive.

- Another major outstanding concern relates to scalability. Reviewers, including ccyd, questioned whether the streaming variational updates and explicit mean-field inference could scale to tasks with higher-dimensional visual input or longer-term dependencies. The rebuttal offered conceptual justification—claiming that AXIOM is designed for efficiency through online updates—but no empirical measurement or complexity analysis was provided to support this. As such, questions about computational limitations and long-term stability of the inference process remained unanswered.

- The reviewers remained uncertain how AXIOM’s inference-planning cycle differs materially from prior work in model-based or Bayesian RL. In particular, they wanted a more explicit comparison to control-as-inference, Bayes-adaptive control, and active inference formalisms.

- The authors defended their decision to use explicit, non-neural factorized inference instead of more expressive amortized methods, arguing that interpretability and theoretical transparency were more important than raw performance. The rebuttal lacked experiments demonstrating that explicit inference confers stability or interpretability benefits that might outweigh the loss in scalability! This concern thus remained conceptually open.

**Reviewer Scores:**

Reviewer ccyd was the most critical in the reviewing process. After the discussion, Reviewer ccyd still found the evaluation narrow, offering only a limited contribution. The confidence scores for the negative reviewers are much higher than the positive ones (in average. 4 v.s.2.5)

The other three reviewer might slightly increase their rating, but the main unsatisfied issue, the scope of empirical testing, is not resolved during the rebuttal.

---

### Decision · Program_Chairs · 2026-01-26

Reject